# Characterising hillslope-stream connectivity with a joint event analysis of stream and groundwater levels

Daniel Beiter[1], Markus Weiler[2], and Theresa Blume[1]

[1]GFZ German Research Centre for Geosciences, 4.4 Hydrology, Potsdam, Germany
[2]University of Freiburg, Chair of Hydrology, Freiburg, Germany

**Correspondence:** Daniel Beiter (daniel.beiter@gfz-potsdam.de)

**Abstract.** Hillslope-stream connectivity controls runoff generation, both during events and baseflow conditions. However, assessing subsurface connectivity is a challenging task, as it occurs in the hidden subsurface domain where water flow cannot be easily observed. We therefore investigated if the results of a joint analysis of rainfall event responses of near-stream groundwater levels and stream water levels could serve as a viable proxy for hillslope-stream connectivity. The analysis focuses on

the extent of response, correlations, lag times and synchronicity. As a first step a new data analysis scheme was developed, separating the aspects of a) response timing and b) extent of water level change. This provides new perspectives on the relationship between groundwater and stream responses. In a second step we investigated if this analysis can give an indication of hillslope-stream connectivity at the catchment scale.

Stream- and groundwater levels were measured at five different hillslopes over 5 to 6 years. Using a new detection algorithm

we extracted 706 rainfall response events for subsequent analysis. Carrying out this analysis in two different geological regions (schist and marls) allowed us to test the usefulness of the proxy under different hydrological settings while also providing insight into the geologically-driven differences in response behaviour.

For rainfall events with low initial groundwater level, groundwater level responses often lag behind the stream with respect to the start of rise and the time of peak. This lag disappears at high antecedent groundwater levels. At low groundwater levels the

relationship between groundwater and stream water level responses to rainfall are highly variable, while at high groundwater levels, above a certain threshold, this relationship tends to become more uniform. The same threshold was able to predict increased likelihood for high runoff coefficients, indicating a strong increase in connectivity once the groundwater level threshold was surpassed.

The joint analysis of shallow near-stream groundwater and stream water levels provided information on the presence or absence

and to a certain extent also on the degree of subsurface hillslope-stream connectivity. The underlying threshold processes were interpreted as transmissivity feedback in the marls and fill-and-spill in the schist. The value of these measurements is high, however, time series of several years and a large number of events are necessary to produce representative results. We also find that locally measured thresholds in groundwater levels can provide insight into the connectivity and event response of the corresponding headwater catchments. If the location of the well is chosen wisely, a single time series of shallow groundwater

can indicate if the catchment is in a state of high or low connectivity.

# 1 Introduction

Hillslope-stream connectivity controls both runoff generation (Detty and McGuire, 2010; Jencso et al., 2010; Penna et al., 2015; Scaife and Band, 2017) and export of solutes, pesticides (Ocampo et al., 2006; Jackson and Pringle, 2010) and particulate matter (Thompson et al., 2013). Understanding patterns, controls and dynamics of hillslope-stream connectivity is therefore of interest not only for flood prediction but also for water quality management and policy making. Ali and Roy (2009) collected various definitions of hydrologic connectivity used in previous studies, which differ in spatial scale (hillslope vs watershed) and observed features (e.g. water cycle or landscape). In this study, we considered hydrologic connectivity as "The condition by which disparate regions on a hillslope are linked via lateral subsurface water flow (Hornberger et al., 1994; Creed and Band, 1998)" Unfortunately, the investigation of this connectivity is notoriously difficult, for a number of reasons: it is variable in space and time (much more than our catchment models generally account for) and it is often controlled by thresholds, either in wetness state or in forcing (rainfall amounts and intensity) (Detty and McGuire, 2010b; McGuire and McDonnell, 2010; Scaife and Band, 2017; Oswald et al., 2011; Graham et al., 2010). Full connectivity is usually established only during brief periods of time (Freer et al., 2002; Ocampo et al., 2006; Haught and Meerveld, 2011; van Meerveld et al., 2015). Identifying and measuring hillslope-stream connectivity becomes even more challenging as we are dealing with extensive along-stream interfaces which makes identification/pinpointing of hot spots difficult. While surface connectivity at least often leaves visible traces, subsurface connectivity is usually invisible and therefore hard to localise and measure (Blume and van Meerveld, 2015).

Standard approaches for the investigation of hillslope-stream connectivity include hillslope trench studies (often combined with piezometers) (Bachmair and Weiler, 2014; van Meerveld and McDonnell, 2006b) and tracer-based analyses (McGuire and McDonnell, 2010; McGlynn and McDonnell, 2003; Anderson et al., 1997). While the first approach gives detailed information about (usually) a single hillslope (Graham et al., 2010) it requires considerable effort in the field (both with respect to time and finances), the second approach provides an integral assessment at the catchment scale, but offers little information on spatial patterns or spatial extent of connectivity. At the stream bed interface distributed temperature sensing (DTS) can provide spatially highly resolved information of stream bed temperatures and under favourable conditions information about groundwater inflow points (Krause et al., 2012). While these datasets can be very informative, DTS systems are expensive, require continuous power supply and are time-intensive in installation. All of these methods are often employed on a short-term basis only: a few events, a season, possibly a year. As a result, one is left with the question how representative these snapshots are.

Even though state variables such as soil moisture or groundwater level do not provide actual water fluxes they are often used to assess hydrologic subsurface connectivity (Detty and McGuire, 2010; Haught and Meerveld, 2011; Freer et al., 2002; van Meerveld and McDonnell, 2006b; Ali et al., 2011; Anderson et al., 2010), and using many repeated snap-shots allows to at least infer flow processes (Bracken et al., 2013). Shallow groundwater levels can provide information about catchment state and a joint analysis of groundwater and streamflow dynamics in response to rainfall events offers basic information on runoff generation processes and hillslope-stream connectivity. The relationship of pre-event groundwater levels and streamflow response is often governed by a threshold in groundwater level above which streamflow responds much more strongly than below (Anderson et al., 2010; Detty and McGuire, 2010b; van Meerveld and McDonnell, 2006b). Bedrock topography can cause non-linear

threshold behaviour in cases where the bedrock is highly impermeable or creates reservoirs that need to be filled before spilling over (Freer et al., 2002; Graham et al., 2010; van Meerveld and McDonnell, 2006b). This threshold indicates a sudden increase in contributing area which directly translates to an increase in hillslope-stream connectivity (Anderson et al., 2010; Detty and McGuire, 2010b; van Meerveld and McDonnell, 2006b).

In this study we went for a targeted as well as pragmatic approach: we targeted specifically the footslope and the riparian zone as the essential interface between hillslope and stream. Monitoring shallow groundwater tables in the riparian zone over longer periods of time allowed us to capture a large number of events. We hypothesised that the analysis of these events will provide not full, but representative information on hillslope-stream connectivity. Previous use of piezometers for this purpose often extended over the entire hillslope (Bachmair and Weiler, 2014; van Meerveld and McDonnell, 2006b) which increased financial
and maintenance efforts. While this can be very informative, we suggested that our pragmatic approach focusing only on the footslope and a joint analysis of shallow groundwater and streamflow response to rainfall events would still allow us to develop a general picture of when connectivity is established, how often this occurs and if there is a difference between the sites. Analysing the relationship between responses in near-stream shallow groundwater and stream thus permitted us to determine the dominant processes. We investigated the potential and limitations of this approach by comparing 5 footslopes covering two
distinct geologies. A newly developed data analysis scheme which separates the aspects of response timing and extent of water level change opened up new perspectives on these interactions. With this study we targeted the following hypotheses:

- Hypothesis 1: hillslopes remain disconnected from the stream for most of the time and connect only during short periods of time.

- Hypothesis 2: the selected study sites differ in geologies (schist and marls), topography and soil characteristics. As a
result, their hillslope-stream systems will show differing connectivity patterns.

- Hypothesis 3: monitoring at the footslope can provide information on hillslope-stream connectivity at this location and can indicate connectivity at the headwater catchment scale

## 2  Methods

### 2.1  Study catchment

This investigation targets the $244km^2$ Attert catchment in western Luxembourg, with altitudes between 243 and 549m.a.s.l. (Figure 1, top left). It is driven by a runoff regime with generally low discharge in summer and high discharge in winter. Despite the seasonal differences in runoff, precipitation events are distributed over the entire year, with a mean annual precipitation of 760mm.

The catchment can be divided into three main geologies – marls, schist and sandstone – and two geologies of lower significance
(alluvials and buntsandstein), shown in Figure 1 (top right). Most of the catchment is characterised by marls and Stagnasols with high clay content (20-60%), an undulating landscape and mostly agricultural land use (Sprenger et al. (2016)). The high

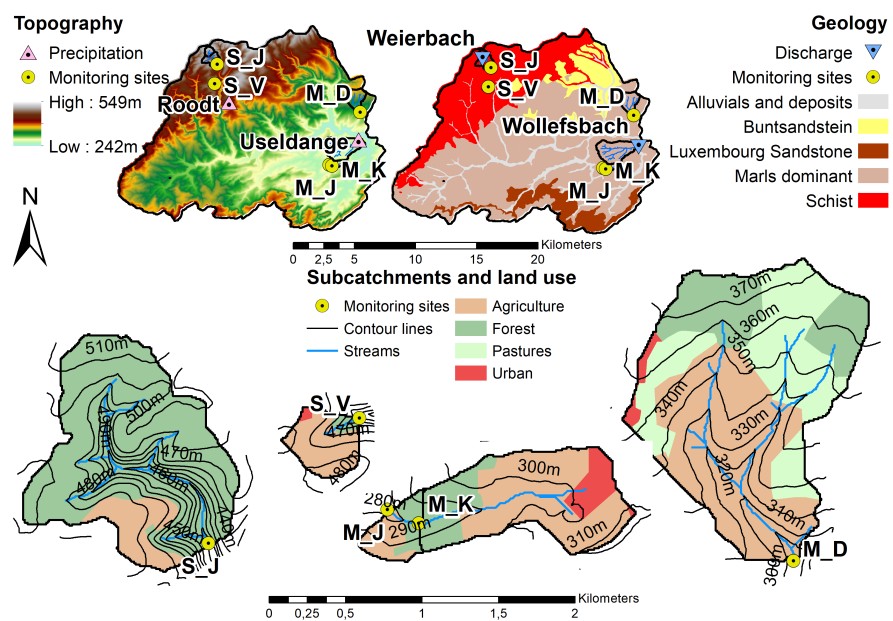

**Figure 1.** The Attert catchment in western Luxembourg and the five monitoring sites: M_D, M_J, M_K (marls), S_J and S_V (schist). Top left: catchment topography, top right: geology, bottom: the five subcatchments including land use.

contents of clay lead to low hydraulic conductivities and a limited drainage capacity. The north-western area (Figure 1) consists of schist bedrock and Cambisols with a texture between loam, silty loam and clayey loam which can drain freely until the soil-bedrock interface (Sprenger et al. (2016)). The landscape is here governed by elevated plateaus with mostly agricultural land use and steep forested hillslopes leading to perennial headwater streams.

A monitoring network with 45 stations was installed in the Attert catchment, recording environmental data such as climate data, soil moisture, groundwater and stream level, amongst others (Zehe et al., 2014; Demand et al., 2019). For the investigation of hillslope-stream connectivity we selected those monitoring sites which were situated at a stream and thus allow a comparison between near-stream shallow groundwater level and the associated stream water level. Unfortunately no such site was available in the sandstone due to its very low drainage density, so the investigation focused on the two geologies marls and schist (Table

1 and Figure 1, bottom). The five selected stations were put into operation between June 2012 and July 2013 and the time span until end of July 2017 was used in the analysis. The spatial arrangement of the piezometers at each site can be seen in Figure 2 and the corresponding elevations and distances from the stream are provided in Figure 3. The prefixes M and S in the site names indicate the two geological regions. The following letter is part of the overall naming-scheme of the monitoring network. A full list of the sites can be found in Appendix A of Demand et al. (2019). M_D is located on a wide meadow

with gentle inclination and Piezometers 1-3 have a distance to the stream between 2m and 10m, while Piezometer 4 is on the steep opposite hillslope directly below a road cut (subsurface probably disturbed during road construction). Piezometer depths extend to about a meter below the stream bed. The other two marls sites – M_J and M_K – are located in a forested plain

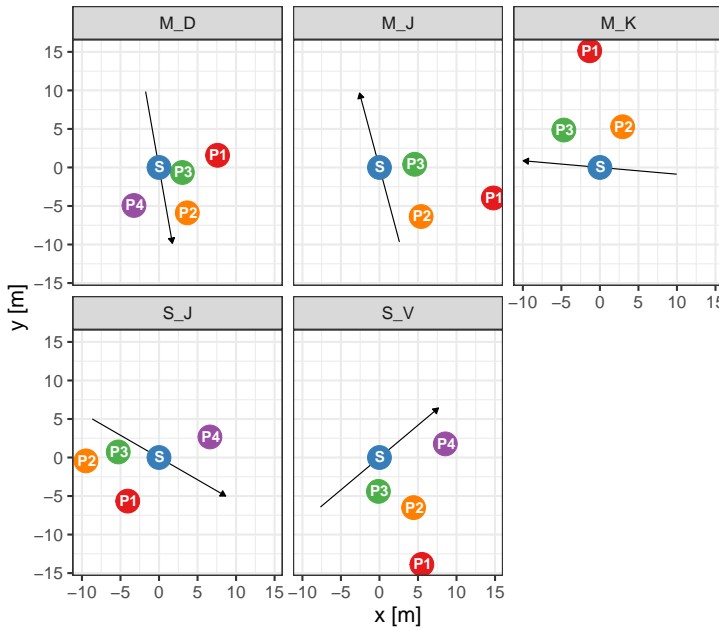

**Figure 2.** Schematic maps of the five sites. "P" stands for piezometer with the corresponding number while "S" stands for stream and is located at (0,0). The arrows point into the direction of stream flow. The coordinates are relative distances to the stream water level sensor (positive y-axis points north).

surrounded by pasture, with the stream incised to about 2.5m and piezometer depths of around 2m. The horizontal distances between stream and piezometers are between 4m and 13m for both sites. S_J is located on a small meadow flood plain, flanked by steep forested hillslopes on both sides of the stream. Piezometer depths are here around 1.5m and reach below the stream bed. Piezometers 1-3 are situated on one side of the stream with distances of about 4 - 8m, while Piezometer 4 is located on the other side at a distance of 6m. S_V is located at a steep forested hillslope in a headwater catchment dominated by pasture on the higher plateau. The distance to the stream is between 2m (Piezometer 4) and 15m (Piezometer 1) and only the lower piezometers (3 and 4) extend to depths below the stream bed. Average hydraulic conductivities for the two soil types range from 293 to 675 cm/day (stagnosols) and from 360 to 648 cm/day (cambisols) (Sprenger et al. 2016)

## 2.2 Monitoring data

Each of the five sites described in section 2.1 was equipped with three to four piezometers to measure shallow groundwater level and one sensor for stream water level. Vertical boreholes were drilled until refusal using the Cobra TT jackhammer with a hollow boring head of 75mm diameter. Refusal was either defined as bedrock (in schist) or when a very dense layer of clay soil was reached (marls), which could not be further penetrated by the cobra. Perforated PVC tubings of 50mm diameter were wrapped into non-woven fabric, installed and packed with filter gravel between 4 and 8mm diameter. The uppermost 30cm below ground level were packed with sealing clay to prevent infiltration bypassing the soil. Depth of refusal was in most cases

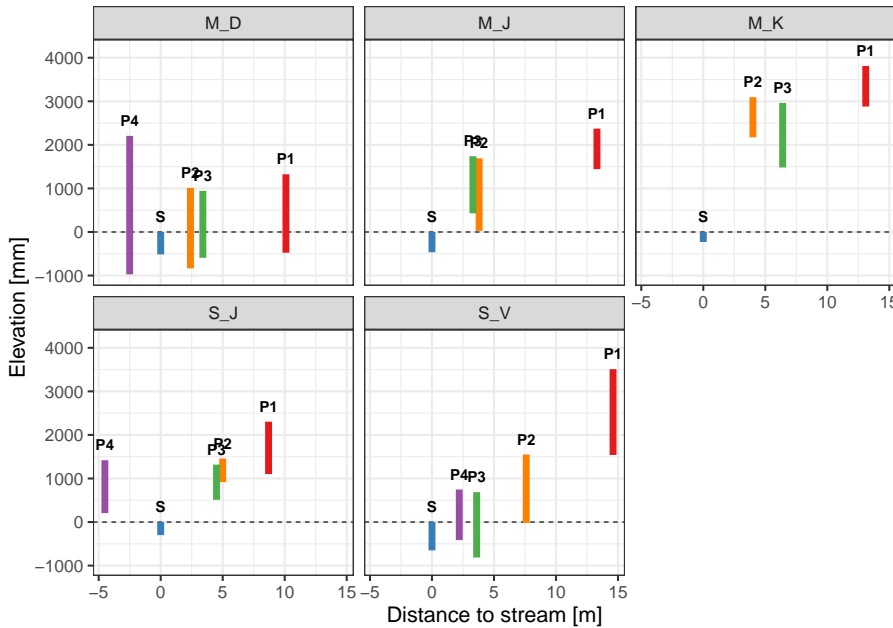

**Figure 3.** Elevations of ground level (upper end end of the bar) and sensor level (lower end) relative to the stream bed. Stream sensors were installed slightly below the stream bed (negative lower end). Distance to the stream is shown on the x-axis. Colour coding is the same as in Figure 2.

**Table 1.** The basic attributes of the monitoring sites.

| Site name | Geology | Soil | Land use | Drainage[1] | Slope quartiles[2] | Upstream area | # of Piezometers |
|-----------|---------|------|----------|-------------|--------------------|--------------|------------------|
| [-] | [-] | [-] | [-] | [-] | [°] | [ha] | [-] |
| M_D | Marls | Stagnosol | Pasture | Limited | 2.3 / 3.3 / 4.5 | 200 | 4 |
| M_J | Marls | Stagnosol | Forest | Limited | 1.3 / 2.3 / 3.6 | 80 | 3 |
| M_K | Marls | Stagnosol | Forest | Limited | 1.3 / 2.2 / 3.3 | 68 | 3 |
| S_J | Schist | Cambisol | Pasture | Free | 2.3 / 4.8 / 12.2 | 154 | 4 |
| S_V | Schist | Cambisol | Forest | Free | 2.7 / 5.2 / 8.0 | 17 | 4 |

[1] According to Sprenger et al. (2016)

[2] Slope quartiles refer to the individual subcatchments (see Figure 1).

below 2m and the water level sensors were installed around 2cm above the bottom.

The sensors used were CTD temperature corrected pressure transducers by METER (formerly Decagon), measuring electric conductivity, temperature and water depth. Full scale is 10m, with a resolution of 2mm and an accuracy of $\pm0.05\%$ of full scale. Connection cables provide ventilation to the transducer and compensate for air pressure. Automated data loggers (CR1000

by Campbell Scientific) logged the data with a temporal resolution of 5min. Hourly precipitation data from the Roodt and Useldange weather stations were obtained from AgriMeteo Luxembourg. Both stations are located within the Attert catchment, the Roodt station close to schist and the Useldange station being close to marls sites (Figure 1, upper left). Discharge data with 15 min temporal resolution were provided from the Luxembourg Institute for Science and Technology (LIST) for the Weierbach 5   station (for schist) and the Wollefsbach station (for marls) (Figure 1, upper right).

## 2.3   Event definition

Automatic event detection is essential when working with long time series and a large number of events. To this end, it is necessary to define a generic response pattern (Figure 4). The general response pattern begins with a pre-event minimum ($h_{preMin}$). When a precipitation event starts, the water level increases until it reaches its peak ($h_{maximum}$). After that peak, 10   water level decreases and the event ends with a post-event minimum ($h_{postMin}$) that might differ from the pre-event minimum. These three points are used to describe water level changes during the event. However, the time period between the two minima (pre- and post-event) is not a robust measure for the event duration. Before or after events water levels are often not stable but subject to small but misleading trends (e.g. wetting-up phase or recession). While searching for the two minima a minimal decline has almost no effect on the water level but inappropriately increases the extracted event duration. To compensate 15   for that, two threshold points ($h_{riseThreshold}$ and $h_{fallThreshold}$) were introduced – one on each limb – that allow for a better temporal representation of each event. Both are defined as a certain percentage of $h_{preAmplitude}$ and $h_{postAmplitude}$. In the case of the rising limb the time where the water table exceeds $h_{riseThreshold}$ is called $t_{rise}$ (see Figure 4). Analogously, the moment the water level falls below $h_{fallThreshold}$ is defined as $t_{fall}$. The distances to $t_{maximum}$ are described as the $t_{riseInterval}$ and $t_{fallInterval}$, respectively. So for time-related analyses these two intervals are used as they are not prone to pre- and post-event 20   trends, but capture the actual event response dynamics. A percentage of 10% of $h_{preAmplitude}$ and $h_{postAmplitude}$ was found to be suitable for that task.

## 2.4   Event detection

The purpose of the event detection is to parse the entire water level time series and extract those intervals during which the water level shows a response to rainfall. Algorithm 1 specifies the necessary steps for the event detection. At first, minimum 25   amplitudes and search intervals need to be defined. Both parameters are subject to a compromise: The minimum amplitudes are used to prevent measurement noise from being mistakenly detected as events, with the drawback of possibly excluding actual low-amplitude events from detection. Search intervals are used to discriminate between subsequent events, which involves the risk of not completely capturing a very long event. In a second step, all local maxima of the stream water level are located. Thirdly, for each maximum the pre-defined search intervals are used in order to determine the global minima in the rising 30   and falling limb. Defining these search intervals depends on the catchment size. Generally speaking, the search interval for the rising limb should be approximately equal to the concentration time of the subcatchment to guarantee that the complete rising limb is covered. Therefore, shorter search intervals are suited for headwaters (several hours to a day) and longer ones for lowland basins (several days). Also, the rise interval is shorter than the fall interval as such events are generally right-skewed

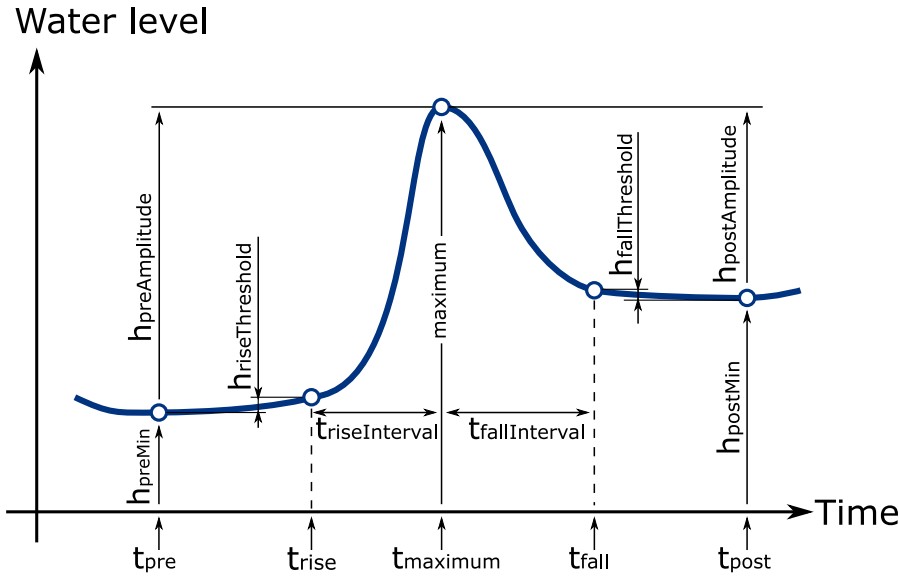

**Figure 4.** Event definition and characteristic variables for event response analysis.

---

**Algorithm 1** Event detection algorithm

---

DEFINE: amplitude thresholds for rising and falling limb ($h_{preAmplitudeMin} = 20$mm, $h_{postAmplitudeMin} = 10$mm)

DEFINE: : two fixed search intervals from peak along the rising and the falling limb ($t_{preSearchInterval} = 24$h, $t_{postSearchInterval} = 48$h)

FIND: all local maxima in time series

**for** each maximum **do**

    FIND: absolute minima on rising and falling limb within defined search intervals

    CALCULATE: $h_{preAmplitude}$ and $h_{postAmplitude}$

    **if** $h_{preAmplitude} < h_{preAmplitudeMin}$ or $h_{postAmplitude} < h_{postAmplitudeMin}$ **then**

        DISCARD: current maximum

    **end if**

**end for**

MERGE: overlapping events

RETURN: $t_{pre}$ and $t_{post}$ of each merged event

---

due to retention behaviour. If two or more events overlap they are merged into one single longer event (Figure 5) and the highest peak is determined as the event maximum. From there on it is handled as a simple event according to Figure 4.

The event detection was first applied to the stream water level time series which returns $t_{pre}$ and $t_{post}$ for each detected stream event. For each of these stream events a subsequent event detection is performed on the shallow groundwater level time series.

Thus, we only include events in the analysis where stream water levels showed a response. Using each stream event for the detection of a possible groundwater event implies that the maximum temporal extent of the groundwater event is equal to the stream event. This is a shortcoming of this method, as a time lag between shallow groundwater and stream or drawn-out groundwater recession might lead to the predefined search window clipping the drawn-out event in the shallow groundwater.

However, in the case of multiple subsequent events a clear definition must exist in order to keep a one-to-one relation between stream and ground water events. If no temporal boundaries were applied for subsequent event detections, an event in the shallow groundwater might overlap with two or more stream events which would drastically increase the complexity of the analysis. Because of the relatively small distances of less than 15m between stream and piezometers, and the small headwater catchments, response delays between stream and piezometer are presumed to be rather short, reducing the risk of clipping. Also, taking $t_{pre}$ and $t_{post}$ as the temporal extent for subsequent detections in groundwater provides a buffer for potential lag times. This one-to-one approach is considered most appropriate as it is a trade-off between good operability of the detection algorithm and a high coverage of stream and groundwater events.

Amplitude thresholds were chosen via trial and error to prevent diurnal stream water fluctuations caused by root water uptake from provoking (erroneous) events. The threshold for the rising limb (20mm) is greater than for the falling limb (10mm) because during the wetting-up phase (in autumn) post-event water levels are very often higher than the pre-event water levels as the catchment becomes more saturated. However, on shorter time scales wetting-up can also occur in other seasons. Using the same threshold for rising and falling limb would lead to the rejection of small events with such a behaviour.

Search intervals were estimated by testing a range of values. A fixed time of 24h for the rising limb performed satisfactorily in our catchments even for long precipitation events and did not merge several subsequent events into one bulk event. With 48h for the falling limb the retention behaviour of the catchment was taken into account allowing for a long-tailed recession in comparison to the rise. The detection algorithm was run for each site individually, as a result the number and selection of detected events is site-specific.

## 2.5 Event type

Introducing event type descriptors allows to infer specific characteristics of a site and its experimental setup. The total number of events for a certain site is defined by its stream response, regardless of whether or not the shallow groundwater responds during the stream events. Event types are: *Complete* detections arise when the water level sensor was initially submerged and the occurring event fulfils the stated detection criteria. For *Partial* detections the criteria are met but the piezometer is initially dry, so it is unknown how far below the sensor level the event started. *Dry* events are events where the piezometer is dry during the stream event and does not record any response. If no local maximum could be found in the groundwater during a stream event, the type was set to *noLocalMaximum*. *lowAmplitudes* means that the rise and/or fall amplitude thresholds are not surpassed. This might be due to a very low-amplitude response but can also cover events with a high rise amplitude but low fall amplitude, in particular when the peak is very close to the $t_{post}$ boundary, which signals a long time lag between stream and groundwater. *allNA* indicates technical sensor problems in the piezometers during the detected streamflow event. While only *complete* events contain valid state and timing variables that can be put into relation with the stream (and are subsequently

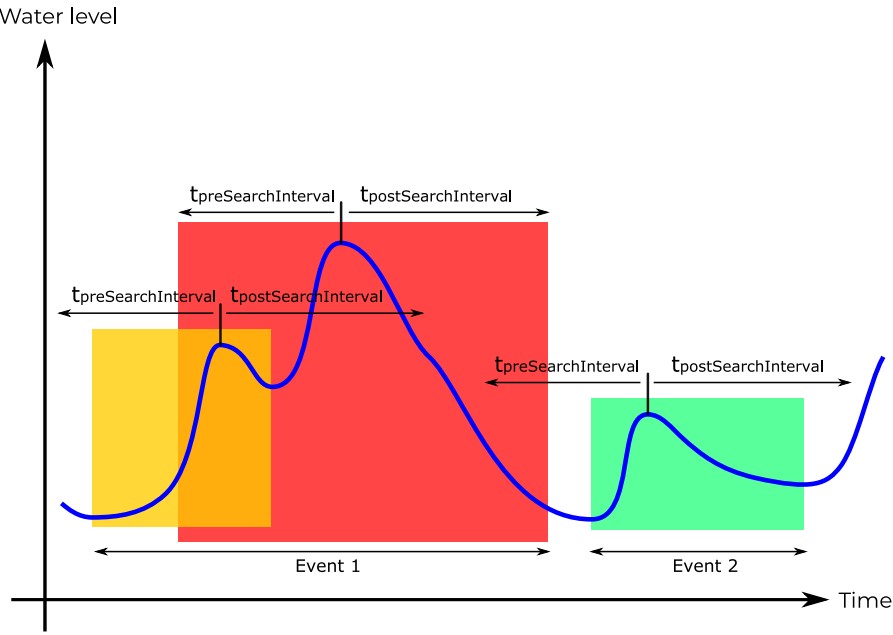

**Figure 5.** Merging conditions of consecutive events. Time series of water level showing three local maxima and the corresponding search windows for the minima. The coloured boxes mark the independently detected events as the interval between the two absolute minima around each peak within the respective search interval. The yellow and red events overlap and are merged into one. The green event is an independent second event.

used for the detailed analyses), all non-complete events also contain relevant information. Knowing about the frequency of occurrence of these other event types helps to characterise each piezometer and site.

## 2.6  Event analysis

The event analysis aims for a better understanding of how and under which conditions the shallow groundwater connects to the
5   stream or disconnects from it. Observing the relation of water table dynamics between stream and shallow groundwater, can reveal connectivity patterns which in turn give insight in the underlying processes. This simultaneous view on groundwater and stream is what is defined as the groundwater-stream (response) relation. A many-event approach ensures that a high variability of catchment conditions and response behaviours is incorporated into the analysis to cover the entire bandwidth of hydrologic system behaviour. Analyses covering single or a low number of events lack the ability of estimating variability and do not allow
10   to deduce how "typical" or "extreme" the event is and if it is representative.

Because the problem is multi-dimensional and considerably complex, a strategy was chosen that allowed us to examine various aspects of the hydrologic responses independently. Combining the information of these different aspects should then give a deeper insight into the occurring processes that control the various hillslope-stream-systems.

The hillslope-stream connectivity can be investigated for periods before an event starts where underlying hydrologic processes

take place on more long term (seasonal) time scales and are represented by the baseflow. As a measure for this connectivity during baseflow conditions (between events) the rank correlation of all pre-event minima ($h_{preMin}$) between each piezometer and the corresponding stream was used. To visually compare before-event relations across piezometers and sites, each sensor's water level was normalised by its minimum and maximum $h_{preMin}$ value.

$$5 \quad h'_{preMin} = \frac{h_{preMin} - \min(h_{preMin})}{\max(h_{preMin}) - \min(h_{preMin})} - \begin{cases} 0 & \text{for stream} \\ 1 & \text{for groundwater} \end{cases} \tag{1}$$

Equation 1 describes the normalisation and results in a values for $h'_{preMin}$ between 0 and 1. To indicate whether the normalised water level is above (stream) or below ground level (piezometers), the value 1 was subtracted when groundwater levels were normalised. This results in values for $h'_{preMin}$ between -1 and 0 for groundwater.

In hillslope-stream systems infiltration and runoff generation processes are highly dynamic during events on a time scale of hours and days. To gain additional insight into what happens during these periods we chose to handle the water level changes and timing as two separate aspects. This provides us with a view of the temporal behaviour on the one hand and changes in the state variables (water levels) of the hydrologic system on the other.

Relative timing and lags between groundwater and stream responses extracted from a large number of events hint at causal relationships. To investigate the variability of this relative timing across all events, piezometers and sites the response behaviour was reduced to timing effects only. A very similar normalisation approach as in Equation 1 was used to compare timings of groundwater responses with those of the stream. Equation 2 uses the time at which the stream exceeds the 10% threshold $t_{rise\ stream}$ and the time where it reaches its peak $t_{max\ stream}$ to normalise groundwater and stream event timing.

$$t' = \frac{t - t_{rise\ stream}}{t_{max\ stream} - t_{rise\ stream}} \tag{2}$$

This stream-based normalisation leads to a value of 0 for the $t_{rise}$ in the stream and 1 for the $t_{maximum}$. A corresponding groundwater event that starts at 0 and reaches its maximum at 1 has the exact same timing as the stream. Values below 0 correspond to a time before the stream responded while values above 1 correspond to a time where the stream already is in recession. By applying this normalisation it is possible to compare relative time lags between stream and groundwater as well as differences in the duration.

The extent of water level increases in stream and groundwater and the relationship between the two can provide useful information on the dominant runoff generation processes. We would expect that a given increase in groundwater level at a given depth would result in a more or less predetermined/deterministic increase of stream water level (assuming the groundwater fluctuations are representative of the catchment). This means that if Events A and B have similar initial conditions and cause similar groundwater level rises we would expect the stream water level rise of Event A to be the same as for Event B. In this case one observation could be used to predict the other. As this also assumes that there is a connection between groundwater and stream and that runoff generation is controlled by shallow groundwater contributions, deviations from deterministic relationships are an indication of other runoff generation processes or flow path variability. Removing the temporal component and only focusing on the extent of the increase between pre-event water level and peak water level enables inspecting this

relationship.

To investigate if shallow groundwater observations at a given hillslope can be used as a proxy for the state of connectivity in the entire catchment we analysed the relationship between event runoff coefficients and the depth. The runoff coefficient describes the ratio of accumulated event discharge at the catchment outlet and accumulated catchment precipitation (Equation 3). Even though each experimental site monitors stream level, no reliable discharge information is available since rating curves are fragmentary and thus uncertain, or do not exist. Therefore, runoff coefficients ($C$) are calculated for nearby subcatchments (Wollefsbach and Weierbach; see Figure 1). The spatial proximity ensures that detected stream water level events coincide with discharge events. The approach to separate baseflow from discharge is based on the constant slope method (Dingman, 2002). Baseflow ($Q_{baseflow}(t)$) was defined as the area below the straight line connecting $t_{rise}$ and $t_{fall}$ and was subtracted from the total discharge ($Q(t)$) to calculate the actual stormflow. Precipitation ($P(t)$) from Roodt station was considered sufficiently representative across the Attert Catchment to be used for all runoff coefficient calculations.

$$C = \frac{\int_{t_{rise}}^{t_{fall}} Q(t) - Q_{baseflow}(t)\, dt}{A \int_{t_{rise}}^{t_{fall}} P(t))\, dt} \tag{3}$$

Relating the shallow groundwater information to the event runoff coefficients can help us to assess how representative the local measurements are for the entire catchment upstream.

## 3   Results

### 3.1   Event detection

Our event detection algorithm identified between 119 and 159 stream runoff events per site and covered a period of five to six years. Not all of these were also detected in all piezometers (Figure 6 and 7). This can be due to data gaps as a result of technical failure of the sensor or data gaps as the piezometer fell dry or because the response in the groundwater was strongly dampened and thus did not fulfil the criteria of the algorithm. In general, the temporal distribution of the detected events shows similar patterns across all sites (Figure 6). It also allows to identify M_D_Piezo4 and M_K_Piezo3 as behaving very differently with many *lowAmplitudes* and *allNA* events. In the case of *lowAmplitudes* we found that many events were clipped by the pre-defined time-window due to very long delays in relation to the stream, which were longer than in the other piezometers at these sites.

As the analysis covers winter and early spring events, the effect of snow fall and snow melt on the event detection was assessed and found to unlikely impact our analysis. Snow fall events are generally quite rare in Luxembourg, so the number of events affected is assumed to be low. A rain on snow event would be captured by its runoff response, but the in this case erroneous estimate of rainfall input would only impact the analysis of event runoff coefficients as our analyses mainly focus on the relationship between streamflow and groundwater responses. Pure snow melt events without a preceding precipitation event are not included in the analysis as precipitation is a necessary identification criterion. Referring to the response type two main patterns can be distinguished (Figure 6). Sites where the sensors remain submerged throughout the observation period thus

producing many *complete* events (M_D and S_J) and sites with piezometers falling dry in summer and autumn (M_K, M_J and S_V). While at the two marls sites these dry periods occur at all piezometers concurrently, at S_V the number of dry events increases in upslope direction (from Piezometer 3 to Piezometer 1). The aggregated values in Figure 7 also reveal two response types with low occurrences – namely *noLocalMaximum* and *partial* events. A total of 68 *partial* events where detected. The *noLocalMaximum* response is very rare with only 11 occurrences. Summary statistics for precipitation, runoff and water level

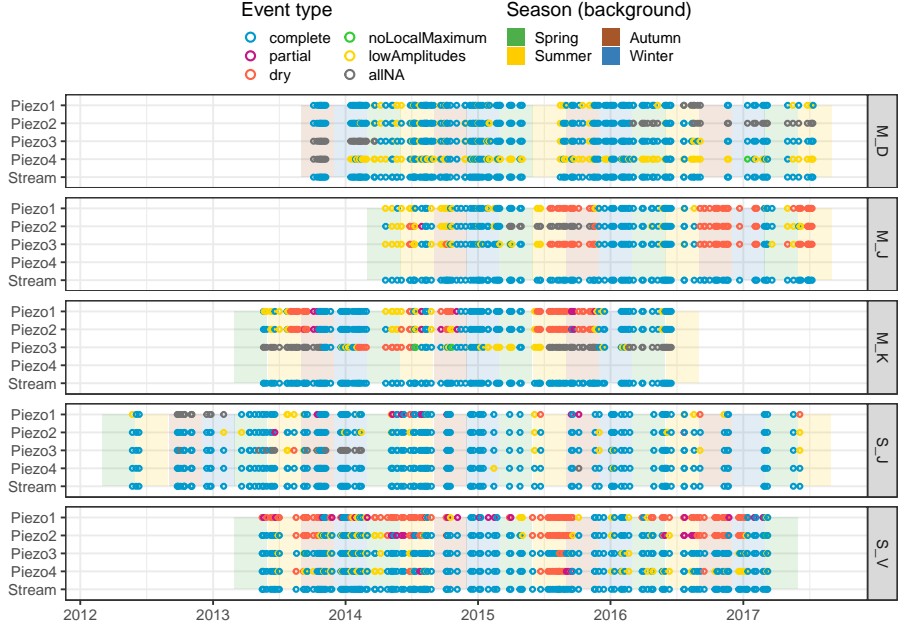

**Figure 6.** Spatiotemporal distribution of detected events for marls (M_) and schist (S_) between June 2012 and July 2017. Seasons are defined as the periods Dec-Feb (Winter), Mar-May (Spring), Jun-Aug (Summer) and Sep-Nov (Autumn)

responses of the detected stream events are shown in table 2. The displayed event runoff describes the total runoff minus baseflow which can lead to a value for the event runoff of 0.0mm.

### 3.2 Before-event hillslope-stream connectivity

The rank correlation coefficients were found to be lower in marls than in schist sites (background colour in Figure 8). In schist
10 only the two upslope piezometers (Piezo1 and Piezo2) of S_V show lower correlation values (0.65 and 0.70), while the others remain above 0.80. For the three marls sites rank correlation coefficients are generally lower (between 0.42 and 0.60) with higher variation. In marls most pre-event groundwater levels cluster in the the shallow depths above $-0.4$ (M_K) and $-0.3$ (M_D and M_J). Schist groundwater levels are more evenly distributed over the entire range (Figure 8). The point colours representing the seasons illustrate that groundwater levels are generally high in winter and spring. Summer events can be found

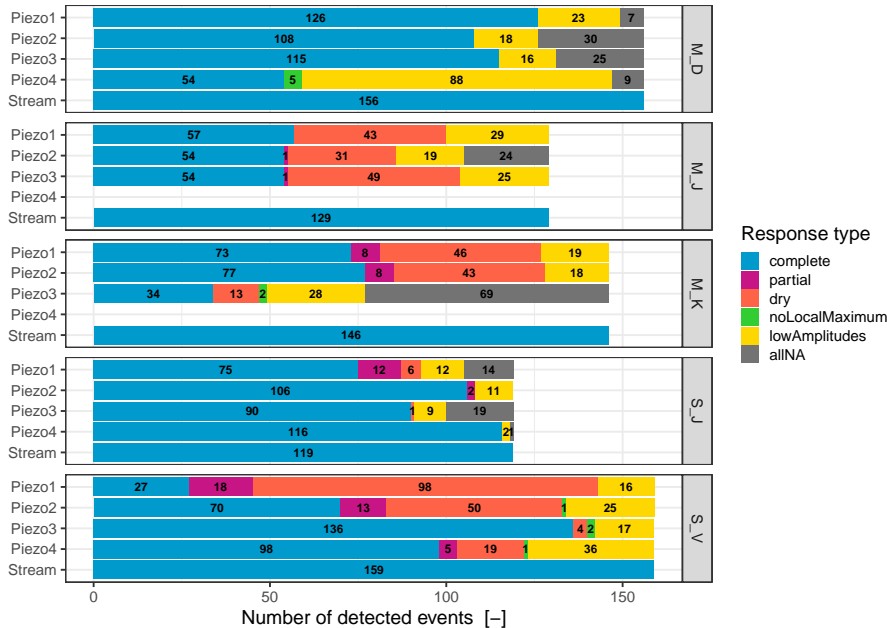

**Figure 7.** Number of detected events in the streams and groundwater at each stream gauge and piezometer, including types of event responses

mostly at the lower end with occasional events at higher groundwater levels. In Autumn the wetting-up phase begins, which produces events over a wider range of groundwater levels.

### 3.3 Comparison of relative response timing between stream and groundwater

The relative timing between groundwater and stream is illustrated in Figure 9. The two black vertical lines represent the timing
of the stream event with $t_{rise}$ at $x = 0$ and $t_{maximum}$ at $x = 1$. Each horizontal bar depicts a groundwater response event with its own $t_{rise}$ at the left end and $t_{maximum}$ at the right end. Groundwater responses that start at 0 and end at 1 have the exact same timing as the stream response. Starting values below 0 reveal a groundwater response before the stream, while an end value above 1 indicates that the stream is already in recession before the groundwater reaches its maximum. The events are sorted on the y-axis by the normalised rise time in the groundwater from delayed groundwater response at the bottom to early
groundwater response at the top. Additionally, the bar colours display the normalised pre-event water levels with high pre-event groundwater level in blue and low pre-event groundwater level in red.

At M_D (Piezo1 to Piezo3), S_J (Piezo1 to Piezo3) and S_V (Piezo3 to Piezo4) a strong relation between pre-event groundwater levels and event timing can be observed. Events occurring at high pre-event groundwater levels (bluish) correspond with a mostly simultaneous rise in groundwater and stream while for events at low groundwater levels (reddish) the groundwater rise
lags behind the stream. Considering the peak, high groundwater events reach their maximum before or simultaneously with the stream while during low groundwater the maximum is reached significantly after the stream. At sites M_J, M_K and S_V

**Table 2.** Characteristics of the stream events summarised for each site. Different values for runoff and precipitation can occur as not all sites cover the same (number of) events. Also, different runoff and precipitation stations were used for marls and schist sites (see Figure 1). An event runoff of 0.0mm can occur by subtracting baseflow from total runoff.

| Variable | Site name | Min | Median | Mean | Max |
|---|---|---|---|---|---|
| Event Runoff [mm] | M_D | 0.0 | 0.6 | 2.7 | 25.0 |
| | M_J | 0.0 | 0.4 | 2.4 | 25.9 |
| | M_K | 0.0 | 0.9 | 2.9 | 21.4 |
| | S_J | 0.0 | 0.2 | 1.6 | 24.3 |
| | S_V | 0.0 | 0.1 | 0.7 | 16.7 |
| Precipitation Intensity [mmh$^{-1}$] | M_D | 0.3 | 2.2 | 3.6 | 21.6 |
| | M_J | 0.6 | 2.4 | 4.0 | 21.5 |
| | M_K | 0.6 | 2.5 | 3.7 | 21.6 |
| | S_J | 0.7 | 3.9 | 5.0 | 17.2 |
| | S_V | 0.4 | 2.9 | 3.9 | 17.2 |
| Precipitation Sum [mm] | M_D | 0.6 | 9.8 | 12.8 | 75.4 |
| | M_J | 1.1 | 10.1 | 13.2 | 62.6 |
| | M_K | 1.0 | 10.4 | 13.0 | 53.3 |
| | S_J | 3.1 | 17.3 | 19.4 | 74.5 |
| | S_V | 1.0 | 13.0 | 14.8 | 58.5 |
| Rise Amplitude [mm] | M_D | 20.0[1] | 64.5 | 88.0 | 378.0 |
| | M_J | 19.3[1] | 46.0 | 64.4 | 282.0 |
| | M_K | 19.0[1] | 48.0 | 61.6 | 227.0 |
| | S_J | 19.2[1] | 46.0 | 55.8 | 241.0 |
| | S_V | 19.0[1] | 43.0 | 51.6 | 137.0 |
| Rise Interval [h] | M_D | 1.4 | 12.2 | 14.7 | 55.2 |
| | M_J | 1.5 | 8.6 | 11.9 | 55.2 |
| | M_K | 1.4 | 8.8 | 10.5 | 55.3 |
| | S_J | 1.4 | 10.2 | 14.3 | 62.5 |
| | S_V | 2.8 | 19.0 | 19.1 | 58.9 |

[1] Threshold of 18mm for event detection algorithm (90% of 20mm).

(Piezo1 to Piezo2) this separation of high (bluish) pre-event groundwater on top and low (reddish) pre-event groundwater at the bottom is visible but not quite as pronounced as for the other sites. In general, groundwater and stream level responses are in sync for about 20-60% of the events, depending on site.

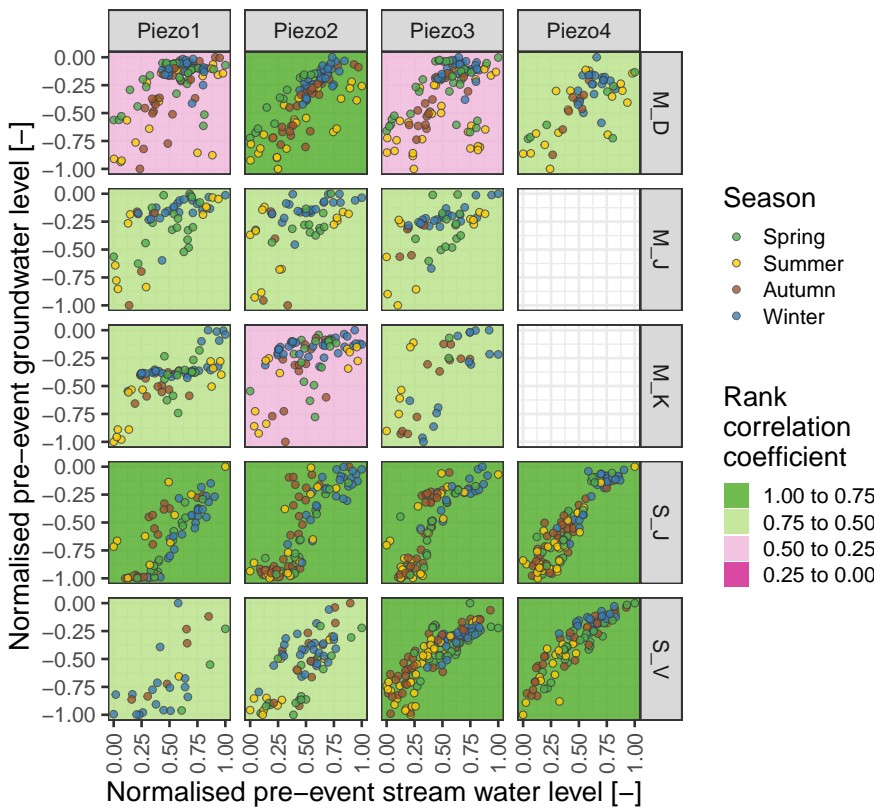

**Figure 8.** Normalised stream and groundwater levels before the investigated precipitation events ($h_{preMin}$). The background colour represents the rank correlation coefficient, the point colours illustrate the season. Both axes are normalised by minimum and maximum $h_{preMin}$ (see equation 1). The negative range on the y-axis indicates depths below ground (groundwater), the positive range on the x-axis depths above ground (stream).

### 3.4 Event-induced increases in stream and groundwater levels

The extent of water level increases in stream and groundwater and the relationship of the two is illustrated in Figure 10. Both pre-event water levels (stream and groundwater) are used as coordinates for the beginning of an event line (lower left point) and the maxima as the coordinates for the end (upper left), with stream water levels on the x-axis and groundwater levels on the y-axis. As we removed the temporal component it is important to keep in mind that peak values did not necessarily occur at the same time. We observe a change in response behaviour between stream and groundwater marked by a threshold which was derived visually (dotted horizontal lines) in Figure 10. The way the patterns changed at the threshold was not identical for all sites. While many piezometers showed an abrupt change in slope (M_D Piezo1-3, M_J Piezo1 and S_J Piezo 2-4) while

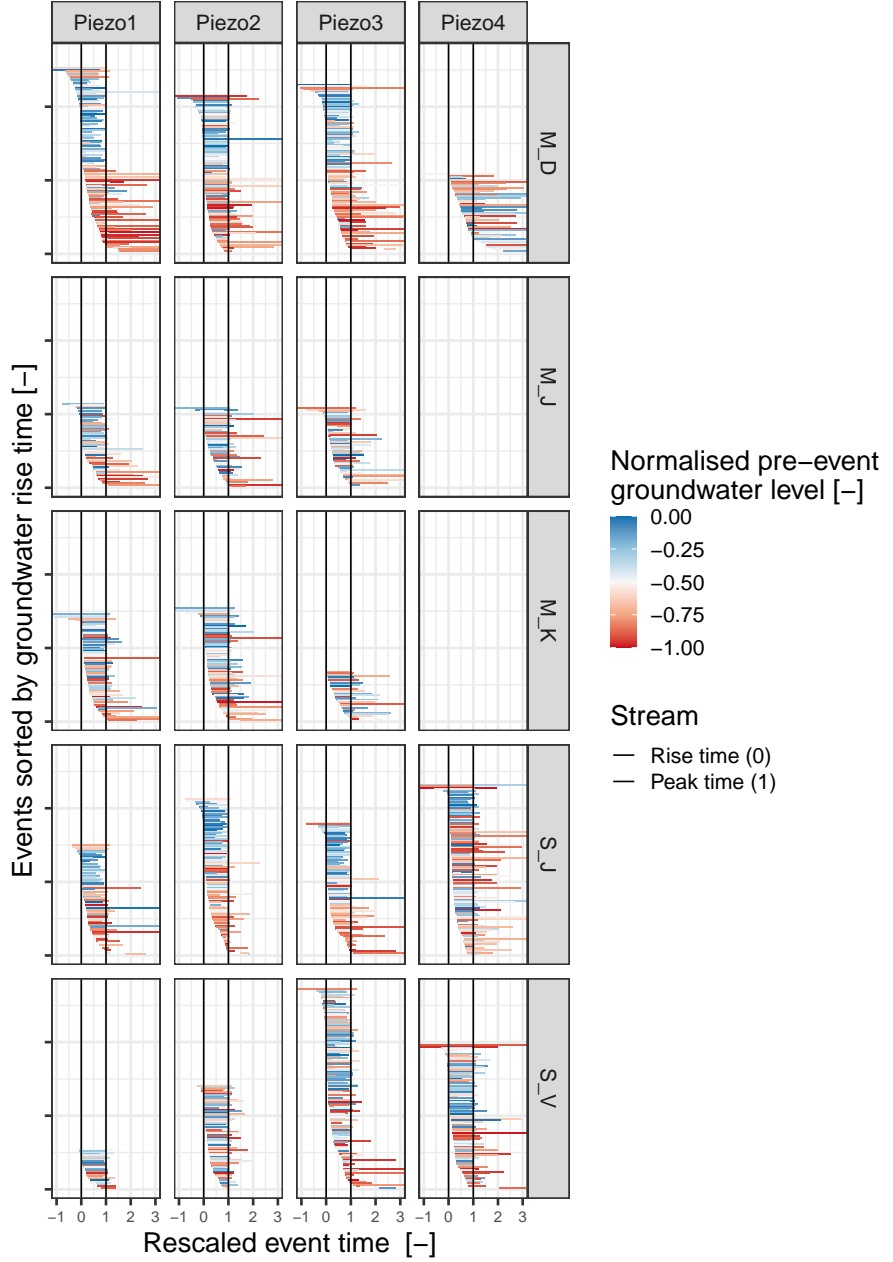

**Figure 9.** Timing of groundwater response relative to stream response. The two vertical lines at 0 and 1 represent the normalised rise and maximum time of each individual stream flow event. Horizontal bars each represent a groundwater event with its individual normalised rise and maximum time. Bluish colours indicate high and reddish colours indicate low pre-event groundwater levels.

others had showed a converging of their envelope functions (encompassing the bundle of slope lines) converging again (S_J Piezo1, S_V Piezo3 and Piezo4). For some piezometers the change in pattern was a sudden clustering of lines (M_K Piezo1-2, S_V Piezo2). All these observed changes in patterns signal that hydrologic processes do change due to different pre-event groundwater levels,when the threshold values are passed. At low groundwater levels amplitudes in the rising limb are large in the groundwater and low in the stream (steep slope of lines), while above the threshold the amplitudes in groundwater are capped at a certain depth below the surface, and stream amplitudes can become large (low slope of lines in Figure 10). Also, the variability of pre-event conditions and event responses is larger below the threshold, while above, the lines are more likely to fall on top of each other and become more deterministic. This is particularly the case for M_D (except Piezo4), M_K (except Piezo3) and S_J. Winter events cluster above the threshold and the other three seasons below the threshold and in the transition zone.

## 3.5 Runoff coefficient

The relation between local pre-event groundwater levels and the event runoff coefficients is displayed in Figure 11. The dotted horizontal lines represent the same individual shallow groundwater thresholds for each piezometer identified in Figure 10 (but here with the normalised pre-event water level on the y-axis). Colours indicate whether the groundwater responded before the stream (red) or after the stream (blue). At M_D, S_J and S_V the pattern is very similar: below the individual pre-event groundwater thresholds runoff coefficients are very small, but increase significantly both in value as well as in variability when pre-event groundwater levels rise above the threshold. For the two forest sites in the marls region – M_J and M_K – the pattern is less clear, with some larger runoff coefficients also occurring below the threshold. A separation with regards to relative response timing (red vs blue) can be observed at M_D and S_J where groundwater responds before the stream for most events above the pre-event water level threshold. At the other three sites M_J, M_K and S_V no clear distinction can be made.

## 3.6 Catchment state

We assume that the threshold (Figure 10) marks a change in catchment state, where conditions above the threshold have the potential for high connectivity while conditions below the threshold indicate lower connectivity. To investigate if the shift in state is synchronous across the sites we plotted the event time series colour-coded by system state (above/below the threshold) (Figure 12). The general pattern clearly shows a common shift in hydrologic connectivity with higher probabilities of catchment states above the threshold from late autumn until early spring. However, below threshold states can occur in winter (see for example the winter of 2016) and above threshold states can also occur in summer (see for example summer of 2014). There is no clear distinction between the geological regions but there are periods where system state varies across the different sites (e.g. fall 2014). However, for most events the below/above threshold state identification is similar in timing across many piezometers.

To study the fraction of events that ended up above the threshold (Table 3), we focused on that piezometer per site that had the largest number of complete events (and excluding M_D_Piezo4 and S_J_Piezo4, which were situated on the opposite slope

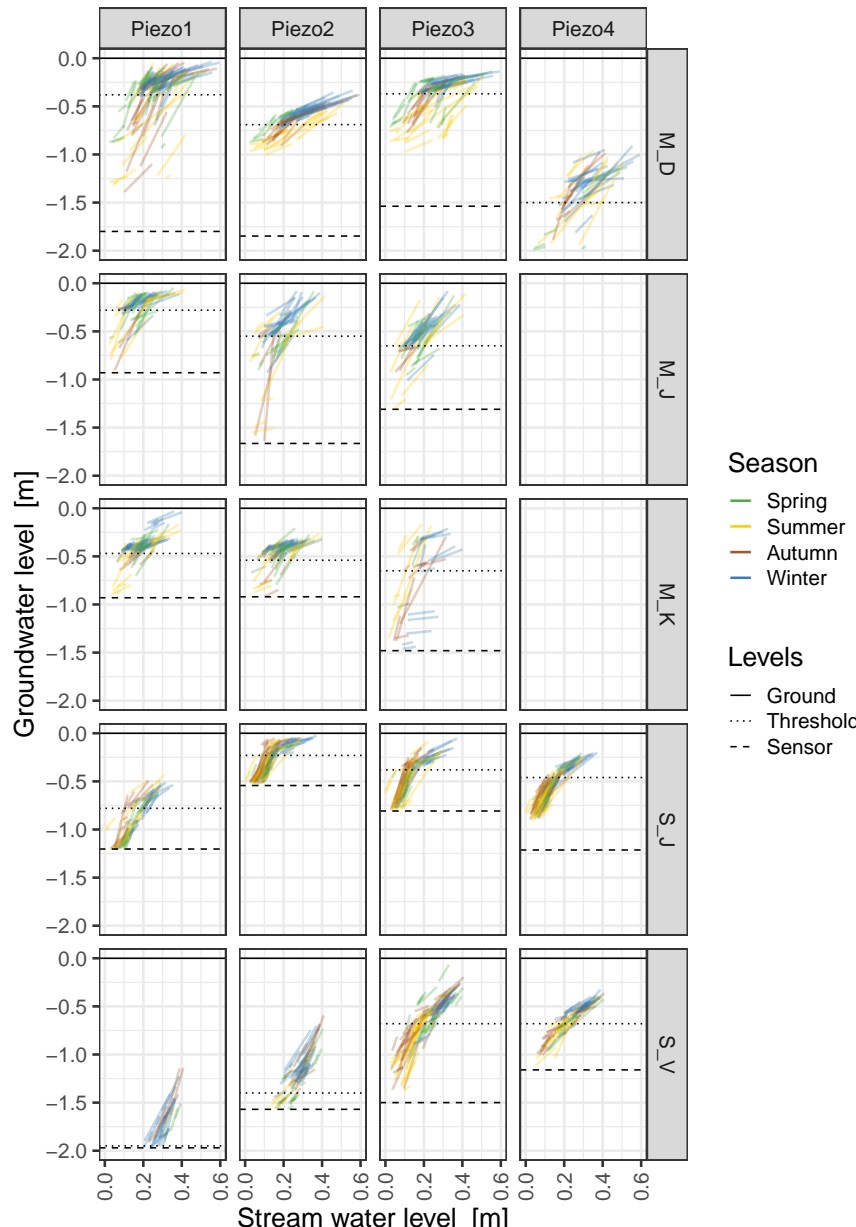

**Figure 10.** Responses in water tables of groundwater (y-axis) and the corresponding stream (x-axis). The lines connect the pre-event minimum with the event maximum. Note that these water levels do not necessarily occur at the same point in time as this visualisation removes the temporal dimension. The y-axis of the plot ends at a depth of 2m for purpose of comparison, thus omitting 2 events occurring below this groundwater level at M_D_Piezo4. Dotted horizontal lines illustrate the threshold between the lower (more variable but mainly steep sloping lines) and upper (less variable with shallow slopes) hydrologic response behaviour.

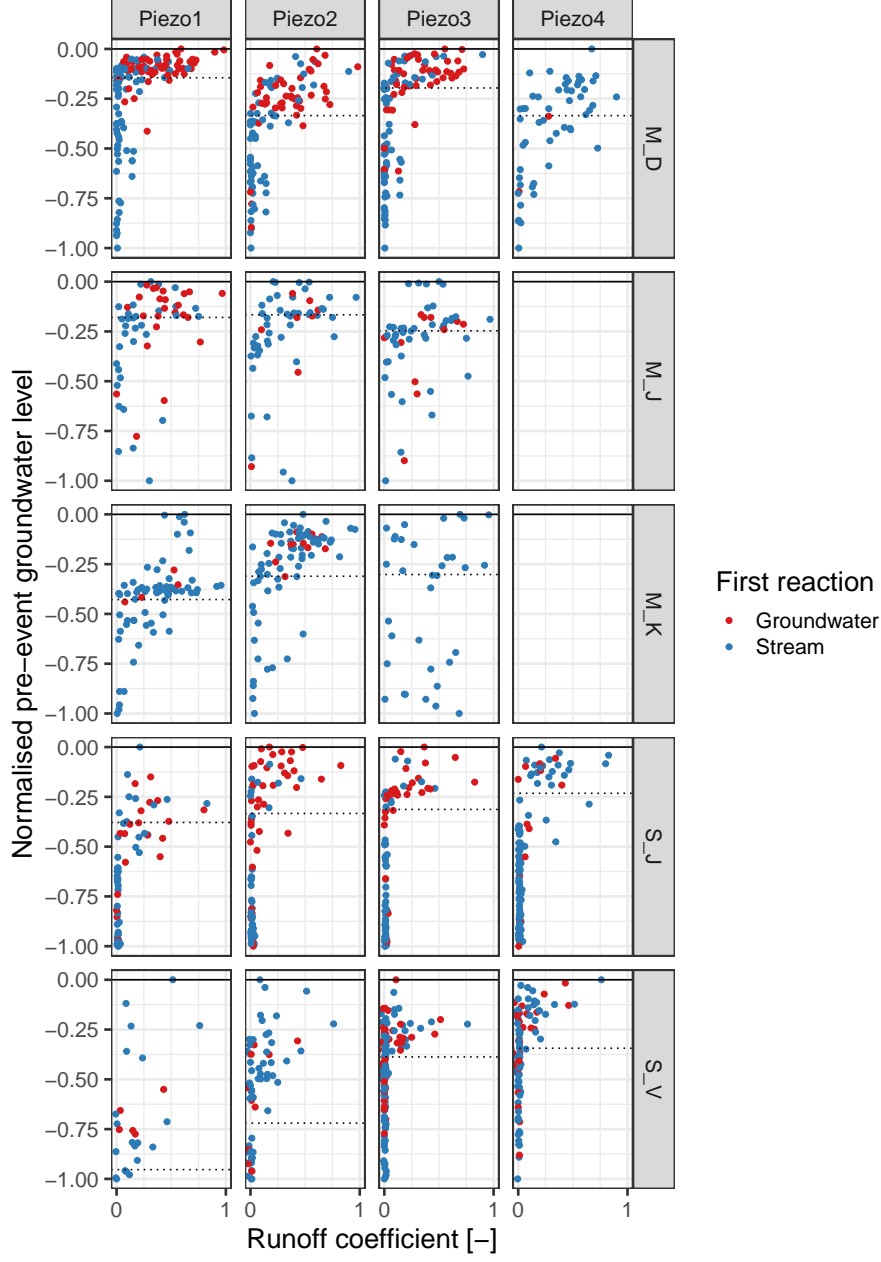

**Figure 11.** Event runoff coefficients versus shallow groundwater levels. Event runoff coefficients were determined for the Weierbach (schist) and Wollefsbach (marls) catchments where discharge data is available. The dotted horizontal lines illustrate the individual thresholds obtained from Figure 10. The point colours indicate whether the groundwater levels responded first (red) or the stream (blue).

compared to the other piezometers at these sites). This selects Piezo1 at site M_D, Piezo2 at sites M_K and S_J and Piezo3 at sites M_J and S_V. The fraction of streamflow events above the threshold ranges between 23% (M_J) and 49% (M_D). There is no relationship between the fraction of events above the threshold and geology, with M_J and S_J having the lowest fractions (<30%) and M_D and S_V the highest fractions (>40%). The low fraction at sites M_J and M_K and S_V_Piezo1 is in part the result of the high number of *partial* and *dry* events (in addition to the *complete* events below the threshold).

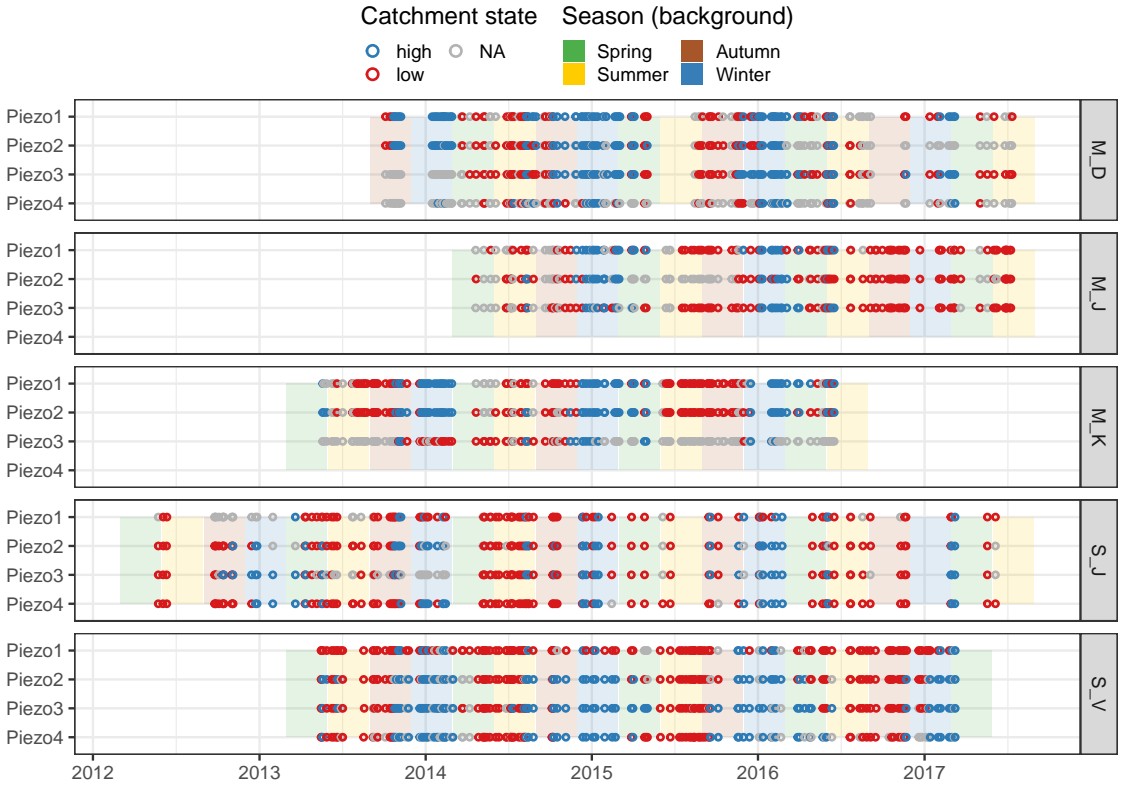

**Figure 12.** Catchment states at the beginning of events. In contrast to Figure 6 it shows whether or not the groundwater levels are above (high/blue) or below (low/red) the locally defined groundwater threshold levels. *Complete*, *partial* and *dry* events are included, all other events are shown in grey (NA).

## 4 Discussion

### 4.1 Event detection

The events summarised in Figures 6 and 7 allow us to identify erratic sensors but also reveal topographic characteristics of the various sites. Topography can explain the occurrence of *dry* events, with a deeply incised stream at M_J and M_K, where

10   we observe the lowest fraction of complete events in the groundwater with 50% or less of the streamflow events, and the

**Table 3.** Fractions of events below (low, including *partial* and *dry* events) and above (high) the threshold. All other event types (*lowAmplitudes*, *noLocalMaximum*, *allNA*) are considered NA.

| Site name | Sensor name | Low | High | NA |
|---|---|---|---|---|
| [-] | [-] | [%] | [%] | [%] |
| M_D | Piezo1 | 32 | 49 | 19 |
| M_D | Piezo2 | 30 | 39 | 31 |
| M_D | Piezo3 | 36 | 37 | 26 |
| M_D | Piezo4 | 17 | 18 | 65 |
| M_J | Piezo1 | 54 | 23 | 22 |
| M_J | Piezo2 | 49 | 18 | 33 |
| M_J | Piezo3 | 60 | 20 | 19 |
| M_K | Piezo1 | 52 | 35 | 13 |
| M_K | Piezo2 | 49 | 38 | 12 |
| M_K | Piezo3 | 21 | 12 | 68 |
| S_J | Piezo1 | 65 | 13 | 22 |
| S_J | Piezo2 | 61 | 29 | 9 |
| S_J | Piezo3 | 49 | 28 | 24 |
| S_J | Piezo4 | 74 | 24 | 2 |
| S_V | Piezo1 | 77 | 13 | 10 |
| S_V | Piezo2 | 52 | 31 | 16 |
| S_V | Piezo3 | 44 | 44 | 12 |
| S_V | Piezo4 | 43 | 33 | 23 |

steep hillslope at S_V leading to a gradient in water level depths and thus differing responses among the piezometers as well as seasonally more strongly fluctuating groundwater levels. The low numbers of *partial* events at sites with high numbers of *complete* and *dry* events (M_J and M_K) signal that the seasonal transition between low and high groundwater levels is very abrupt, skipping intermediate levels. This might be due to pronounced capillarity fringes reaching into the very shallow subsurface. In that case, infiltrating water would reach the upper end of the fringe very quickly and only little water volume would be necessary to lift the groundwater level significantly (e.g. Cloke et al. (2006)). The very low number of only 11 *noLocalMaximum* groundwater events supports the viability of the developed event detection.

## 4.2 Before-event hillslope-stream connectivity

Cross-correlation has been used in previous studies to assess different aspects of hydrologic connectivity, such as lag time analysis between stream and groundwater table (Allen et al., 2010; Bachmair and Weiler, 2014), relating water table connectivity to topographic indices (Jencso et al., 2009) or comparing groundwater levels with runoff coefficients (Seibert et al., 2003). As-

suming well coupled hydrologic systems, high correlation coefficients would be expected, which applies to the two schist sites. Low correlation coefficients indicate a streamflow (baseflow) response that is decoupled from the groundwater. This applies to all three marls sites. However, the within-site variability is not as large as the colour-scheme suggests (for M_D between 0.42 and 0.76 and for M_K between 0.49 and 0.69). Visually comparing the point cloud patterns of the piezometers at each

5 single site (Figure 8) reveals, despite the scatter, site-internal similarity (a site-specific fingerprint) among the piezometers. Two exceptions to this observation are the previously mentioned (see section 3.1) piezometer M_D_Piezo4, which is located in disturbed soil on a steep slope below a road, and M_K_Piezo3, where the anomalous behaviour cannot be explained at first sight. The site-internal similarity in the point-clouds as well as the rank correlation-coefficients suggest that well-placed groundwater observation points can provide information on hillslope-stream connectivity for the given footslope, at least for

pre-event conditions. The observed differences between the geologies suggest that soil texture and bedrock structure might control regional similarities.

## 4.3    Comparison of relative response timing between stream and groundwater

Identical response timing or groundwater rising and peaking just before the stream suggests that hillslope groundwater is driving streamflow response and thus that hillslope-stream connectivity is high (Haught and Meerveld, 2011; Rinderer et al., 2016).

That this occurs under high groundwater levels further supports this conclusion. Groundwater rising and peaking after streamflow indicates that streamflow response is probably not caused by hillslope shallow groundwater and that hillslope-stream subsurface connectivity is low. In a highly heterogeneous catchment, certain "fast" hillslopes with very high hillslope-stream connectivity and high outflows might provoke a stream-response at the stream level gauge before the monitored hillslope responds. In this case the interpretation of low subsurface-connectivity would only hold for the monitored hillslope. During

events with low groundwater levels, precipitation falling onto or very close to the stream might generate a rise in the stream before the groundwater response (McGuire and McDonnell, 2008), as depth of the groundwater level (minus a potential capillary fringe) is the distance a water parcel needs to travel and thus, directly influences the delay in groundwater response. Triggering an early response in stream compared to groundwater can also be the result of infiltration excess overland flow where surface runoff connects faster to the stream than it infiltrates towards the groundwater. However, this can be ruled out for schist as the

high infiltration capacity makes overland flow unlikely, while it can not be ruled out for the clayey soils in the marls region (Wrede et al., 2015). No clear visual differences in timing can be observed between marls and schist. The large variability in response timing confirms the need for monitoring over extended time periods as few or single event analyses run the risk of not being representative. Temporal relationship and water level responses are intertwined in a time series which makes it very intricate focusing on one while looking at both at the same time, e.g. by plotting two time series against each other and

interpreting the resulting hysteresis (Kendall et al., 1999; McGuire and McDonnell, 2010; Zuecco et al., 2016). Choosing to separate the analysis of the temporal response from the water level changes allowed to better reveal the temporal relationship of the hillslope-stream system on the one hand, and water level responses on the other.

## 4.4 Event-induced increase in stream and groundwater levels

Previous studies observed transmissivity feedback as a key mechanism controlling subsurface runoff (Bishop et al., 2011; Detty and McGuire, 2010b). Transmissivity feedback has previously been observed directly via piezometers (Bishop et al., 2011) or indirectly through stable isotope composition in stream runoff (Bishop et al., 2004; Laudon et al., 2004) and tracer transport rates (Laine-Kaulio et al., 2014). In our study the capped response of groundwater events above a certain threshold is a strong indication of transmissivity feedback as one controlling mechanism (M_D and M_K). At low groundwater levels, infiltrating water results in a substantial increase of the groundwater level, suggesting that lateral conductivities are low as water is added more quickly than it can flow away laterally. This changes when the water level reaches a certain level or soil horizon. Now infiltrating water is no longer increasing groundwater level substantially but instead fast lateral transport is likely to be causing the observed pronounced rise in stream water levels. This sudden fast lateral transport of the shallow groundwater is likely due to substantially higher lateral hydraulic conductivity of the upper soil horizons compared to the lower soil horizons. This fits well with the findings by Sprenger et al. (2016) who at site M_K found a strong increase in saturated hydraulic conductivity by a factor of 40 at a depth of 36 cm, while the increase for site M_J (where we did not observe a strong capping of the response) is less than 20% (the other 3 sites were unfortunately not included in the analysis by Sprenger et al. (2016)). A raise in the hydraulic gradient in a more uniform depth profile of hydraulic conductivities, on the other hand, would only lead to a gradual increase in lateral flow. At S_V transmissivity feedback does not seem to occur as the slopes of the lines do not change as abruptly (Figure 10). This is in accordance with the findings of Angermann et al. (2017) at the same hillslope: During sprinkling experiments they observed that relatively high vertical and lateral hydraulic conductivities ($10^{-3}m/s$) lead to fast lateral responses in subsurface. Whether or not a precipitation event can activate certain flow paths depends on the spatial distribution of pre-event water and the characteristics of bedrock topography (van Meerveld et al., 2015). Demand et al. (2019) found that preferential flow is present in particular during dry conditions. When the groundwater level is high, the majority of flow paths are already activated and the degrees of freedom to activate new flow paths are limited. Therefore, the relation between stream and groundwater converges and shifts from variable to more uniform (Figure 10). Investigating rainfall characteristics and their effect on event responses did not help explaining the underlying mechanisms. While rank correlation coefficients between $precipitationSum$ and $h_{preAmplitude}$ reached relatively high values of 0.7 and above, the majority stayed below 0.2 for $t_{riseInterval}$ showing that the precipitation had no clearly identifiable effect on event timing (see Table A1 in Appendix A).

## 4.5 Runoff coefficient

Threshold behaviour is a common observation in runoff generation (Ali et al., 2013), for example Scaife and Band (2017) and Detty and McGuire (2010b) observed a threshold effect of antecedent precipitation and soil moisture on stormflow, and Latron and Gallart (2008) identified a threshold behaviour between groundwater level and runoff coefficient depending on seasonal catchment conditions (dry, wetting-up and wet). In our study the groundwater threshold marking the change in event runoff coefficients (Figure 11) coincides with the regime shift of water table responses (Figure 10). At M_D, S_J and S_V the pattern is

very similar: below the individual pre-event groundwater thresholds runoff coefficients are very small, but increase significantly both in value as well as in variability when pre-event groundwater levels rise above the threshold. For the two forest sites in the marls region – M_J and M_K – the pattern is less clear, with some larger runoff coefficients also occurring below the threshold. A possible explanation could be that Wollefsbach gauge used to determine the runoff coefficients is less representative for these forest sites, as the Wollefsbach Catchment consists almost entirely of pasture and agricultural areas. In addition, the morphology of slopes and stream channel at the two marls forest sites is very distinct (and different to the Wollefsbach), with very low gradients in the slopes but a deeply incised stream bed. As the probability of high runoff coefficients increases above the groundwater threshold it seems that local observations of groundwater levels can give a good indication of catchment state with respect to connectivity and storage and release behaviour. This is true even for neighbouring catchments within the same geological region (M_D and S_V for example are not located in or downstream of the catchments used for the determination of the runoff coefficients). We also find that especially the regime shift and the corresponding threshold can be more clearly identified by groundwater level observations than by antecedent stream water level (Figure 10). This implies that near-stream groundwater observations hold significant predictive power to estimate whether or not an upcoming precipitation event is likely to produce major runoff at the outlet of the subcatchment.

## 4.6 Catchment state

The previously obtained groundwater thresholds allow us to split all events into two groups: Events with catchment states above the threshold are likely to have higher event runoff coefficients (Figure 11) and are thus assumed to generate substantial lateral subsurface stormflow caused by high hillslope-stream connectivity (more connected hillslopes or connectivity extending further upslope, or both). Catchment states below the threshold generate only minor lateral flow. In this case the spatial extent of hillslope-stream connectivity is generally low (few connected hillslopes or connectivity does not extend far up the slopes). Just taking season as a predictor for the expected event response and hillslope-stream connectivity would be too simple: while summer events are likely to be below threshold and winter events above, this is not a general rule and spring and fall events can also not be classified just by their season (Figure 12). However, our study results suggest that it would be sufficient to have the information of one of the piezometers per site to know if pre-event groundwater levels are above or below the threshold. If a rainfall event were to occur when groundwater levels are above the threshold the likelihood of high runoff coefficients is increased. To identify this state (above/below threshold) we do not need all of the piezometers currently installed at a certain hillslope – one would be enough and we could now potentially dismantle the other piezometers. Considering an un-investigated hillslope, one cannot know in advance which location would lead to a 'well-chosen' piezometer and which one to a 'badly-chosen' piezometer. Nonetheless, the analysis showed that local heterogeneity did not influence the piezometers to a degree where no similarity at all could be observed. Therefore, a small number of piezometers (e.g. 3-4) should be enough to identify the characteristic patterns and which piezometers do represent the hillslope and which ones are less suited due to local anomalies. From this point on, one piezometer would be enough to describe the hillslope response and you can remove the other sensors. The well-chosen one would be one that on the one hand is consistent in its response pattern with the majority of the piezometers at this site and on the other hand has the clearest threshold signal among these.

**Table 4.** Observations and corresponding process interpretations

| Observation | Process interpretation |
|---|---|
| Low correlation of pre-event stream and groundwater levels (Figure 8) | Low or only temporary hillslope-stream connectivity (M_D, M_J, M_K) |
| Stream response and peak prior to groundwater response and peak, mainly under dry conditions (Figure 9) | Runoff generated by near-stream overland flow, unsaturated zone preferential flow or direct rainfall (M_D, S_J, S_V) |
| At high groundwater levels: little to no event-induced increase in groundwater levels but high increase in stream water levels (Figure 10) | Transmissivity feedback (M_D, M_K) and fill-and-spill (S_J) |
| Schist: very low runoff coefficients at low groundwater levels. Marls: higher runoff coefficients also occur at low groundwater levels (Figure 11) | Different processes are active in the two geologies at low groundwater levels, surface runoff or preferential flow paths above the shallow groundwater table can produce significant runoff in the marls |
| Marls: groundwater levels cluster at high values, only few data points at low levels, few points in-between (Figure 8) | Groundwater ridging due to capillary fringe effects in the clayey soils |

Even though the temporal dynamics of the switches between above and below threshold conditions are similar across most piezometers and sites, the fraction of stream events ending up above the threshold varies strongly (Table 3). While this only refers to the events and not the continuous time series it still tells us that high connectivity on event basis only occurs for roughly 20-50% of the events. While we saw higher pre-event connectivity at the schist sites (deduced from the rank correlation coefficients of pre-event stream and groundwater levels Figure 8), there was no geological pattern in the fraction of above-threshold events. These two measures describe different aspects of connectivity. While the footslope of the schist sites is well connected during pre-event conditions, this does not necessarily mean that the upslope areas at these sites are more frequently contributing to streamflow than upslope areas where the footslope is less well connected during pre-event conditions.

### 4.7 Synthesis: Process deductions

The joint analysis of shallow near-stream groundwater and stream water levels allows us to identify several runoff generation mechanisms. Observations and the corresponding interpretations are listed in Table 4. The observations described in Table 4 require a large number of events. Only if the number of events is sufficiently high we can capture the variability in responses, the frequency of different response types, the dominant responses and then interpret the underlying processes (Table 5 shows a selection of studies with the number of events analysed).

Events in marls cluster at high pre-event groundwater levels with 60 to 80% of events found in the upper half of the total range

**Table 5.** Selection of studies and the number of events analysed.

| Reference | Temporal extent | Number of events |
|---|---|---|
| Detty and McGuire (2010b) | 3 months | 15 |
| Ali et al. (2011) | 1 year | 50 |
| Penna et al. (2015) | 3 years | 63 |
| Anderson et al. (2010) | 19 months | 99 |
| van Meerveld and McDonnell (2006a) | 2 years 4 months | 147 |
| Scaife and Band (2017) | 15 years | 811 |
| Rinderer et al. (2016) | 2 years and 3 months | 133 |
| Zuecco et al. (2019) | several years | 157 |

and only few events at low levels or in-between. At the same time the piezometers at M_J and M_K experience a considerably high number of dry events but only few partial events (Figure 7). Groundwater transitions fast from very low levels to levels near the surface, with only few events in-between (Figure 8). This fast transition hints towards extended capillary fringes where only low volumes of water are necessary to rise the groundwater table (Cloke et al., 2006). As a result of the transmissivity feedback,

runoff coefficients significantly increase when groundwater levels reach the threshold as the hillslope connects to the stream (Figure 11). This behaviour can be observed in particular at the largest catchment in marls (M_D) with an undulating landscape and mostly pasture and to a lesser degree at smaller catchments with very flat topography and forest. As several characteristics are different between these catchments, this behaviour can not be assigned to one single attribute with confidence. In schist, events are spread over the whole range of pre-event groundwater levels with no clear difference between low, in-between and

high events. Since hydraulic conductivities in schist are generally very high, the sudden increase in runoff coefficient above the threshold can not be explained by transmissivity feedback being the governing process. Nevertheless, capping of groundwater response was observed at S_J. Anderson et al. (2010) found that in watersheds with lateral preferential flow the fill-and-spill mechanism was responsible for capped groundwater responses. This observation can be transferred to the schist site to explain the inhibited groundwater response making its soil-bedrock interface responsible for the threshold relationship.

Studies focusing on the Downslope Travel Distances (Klaus and Jackson, 2018; Gabrielli and McDonnell, 2020) found that only lower regions of a hillslope contribute to the streamflow via interflow, while in upper regions water percolates into the deeper groundwater. In our study, however, we find that there is a threshold in the near-stream groundwater levels above which event runoff coefficients rise strongly to values above 50%, indicating that it is not just the near stream footslope contributing to event runoff.

**5 Conclusions**

In this study we analysed the relation between responses to precipitation of shallow groundwater level and stream level for five different sites in two distinct geologies. An event-based approach was chosen for the analysis of the multi-annual time series

where responses in water level and timing were investigated independently. We found that a multi-event analysis approach including a large number of events is suitable for characterising the hydrologic response behaviour of the hillslope-stream-system and the dynamics of its connectivity. A more selective and exemplary analysis of only a few events would lead to misinterpretation of the results. Nonetheless, the question is not so much about how many events are necessary (in absolute numbers) as more about the necessary time period to cover the temporal variability generated by different hydrological processes. It is therefore necessary to accumulate a large number of events across all seasons. In terms of extreme events (droughts/floods) the covered time period and number of events will need to be even higher, on the one hand to capture these events, and on the other hand to put them into context. Detecting threshold behaviour and identifying the correct threshold would be very unlikely if the above conditions would not be met. Thus, the lack of information on event variability would significantly reduce the confidence of the findings (see Figures in Appendix B).

Revisiting our hypotheses, we now can say the following:

– *Hypothesis 1: hillslopes remain disconnected from the stream for most of the time and connect only during short periods of time.*

We found that the fraction of events above the threshold (with the potential of high runoff coefficients) was roughly 20-50% of the streamflow events, depending on site. Similarly, the relative timing between groundwater and stream level response was very much in sync for 20-60% of the streamflow events, again depending on site. However, as even the events above the threshold do not all produce high runoff coefficients we are unable to falsify the hypothesis. Instead our results indicate that indeed, even though the footslopes might be connected, the hillslopes are often disconnected. Pronounced and continuous footslope-stream connectivity during baseflow conditions is therefore not an indicator of frequently occurring upslope contributions.

– *Hypothesis 2: the two geologies schist and marls differ in topography and soil characteristics. As a result, their hillslope-stream systems will show differing connectivity patterns.*

Differences between the response behaviour of the two geologies were less pronounced than expected for some of the analyses, but the observed results showed that both hydrologic systems are subject to a threshold behaviour where dominating hydrologic processes change. While both geologies show threshold behaviour the underlying processes are likely to be different, with transmissivity feedback occurring in the marls and a more fill-and-spill-like process in the schist. The fact that at low groundwater levels runoff coefficients in the marls tend to be higher than in the schist, in some cases even by an order of magnitude, suggests that also at low groundwater levels different processes are active in the two geological regions. While saturated subsurface connectivity is low at these low groundwater levels, surface runoff or lateral preferential flow above the shallow groundwater must provide sufficient connectivity to enable runoff generation in the marls. Interestingly, the two schist sites showing high pre-event connectivity of stream and footslope had strongly differing fractions of events above the threshold. On the other hand, site M_D had low pre-event connectivity, but a 49% fraction of events above the threshold.

– *Hypothesis 3: monitoring at the footslope can provide information on hillslope-stream connectivity at this location and can indicate connectivity at the headwater catchment scale.*

Our analyses identified patterns that are representative for the site or hillslope, i.e. which were shown by all or most piezometers at these sites. However, piezometers can also be located at points where very local anomalies drastically influence the response behaviour which is why at least three piezometers should be used when first investigating the hillslope-stream relation to secure redundant information and identify the most representative and informative monitoring location for the hillslope or even catchment. Then a single, well-chosen, piezometer can already provide substantial information on catchment state and the potential for high connectivity and thus high runoff events. This conclusion is based on the fact that piezometer water levels above the identified threshold can be related to an increased potential for high event runoff coefficients which in turn indicate increased catchment connectivity. Thus the piezometer water levels above the threshold are indicative of a catchment storage state at which additional rainwater input can easily lead to a strong increase in catchment connectivity and thus runoff production.

The proposed separation of the temporal component and the extent of water level responses for certain aspects of the data analysis proved to be useful in visualising, analysing and interpreting the event response and its variability across a large number of events. Even though the installation and monitoring of piezometers in the near-stream zone is pragmatic and much less cost- and labour-intensive than the installation of hillslope trenches, local near-stream shallow groundwater observations do hold significant predictive power for the potential catchment response. They possibly provide more information than piezometer- or trench observations located further upslope would, as the footslope and riparian zone are both link and gate-keeper, controlling connectivity between hillslopes and streams. Due to the lower cost of piezometer installation and monitoring compared to trenches it is possible to instrument a larger number of sites which in turn makes it possible to systematically investigate subsurface hillslope-stream connectivity in different hydrologic response units instead of focusing on within-slope connectivity on single hillslopes. While we focused on 5 hillslopes in this study it would easily be possible to extend this monitoring design to a larger number of sites thus even better capturing the spatial variability in responses and allowing a thorough investigation into which sites tend to be most representative of the catchment and if these sites can be identified a-priori based on topography or other landscape characteristics. The application of our data analysis to other sites where data is already available might open up new ways of systematic site-intercomparison as our analysis provides a novel way of visualising event responses and thus making the information contained in a large number of events more easily accessible.

# Appendix A: Additional information on rainfall effects and piezometer and profile characteristics

**Table A1.** Spearman rank correlation coefficients between event precipitation and the response variables *riseAmplitude* and *riseInterval*.

| Site Name | Sensor Name | $r_{precipSum,riseAmplitude}$ | $r_{precipSum,riseInterval}$ |
|---|---|---|---|
| [-] | [-] | [-] | [-] |
| M_D | Piezo1 | 0.72 | 0.32 |
| M_D | Piezo2 | 0.79 | 0.24 |
| M_D | Piezo3 | 0.74 | 0.37 |
| M_D | Piezo4 | 0.72 | -0.16 |
| M_D | Stream | 0.55 | 0.14 |
| M_J | Piezo1 | 0.58 | 0.14 |
| M_J | Piezo2 | 0.42 | 0.18 |
| M_J | Piezo3 | 0.71 | 0.16 |
| M_J | Stream | 0.62 | 0.16 |
| M_K | Piezo1 | 0.52 | 0.27 |
| M_K | Piezo2 | 0.59 | 0.20 |
| M_K | Piezo3 | 0.60 | -0.05 |
| M_K | Stream | 0.73 | 0.20 |
| S_J | Piezo1 | 0.72 | 0.19 |
| S_J | Piezo2 | 0.39 | 0.23 |
| S_J | Piezo3 | 0.48 | 0.21 |
| S_J | Piezo4 | 0.60 | 0.29 |
| S_J | Stream | 0.71 | 0.17 |
| S_V | Piezo1 | 0.61 | -0.00 |
| S_V | Piezo2 | 0.68 | 0.06 |
| S_V | Piezo3 | 0.56 | 0.04 |
| S_V | Piezo4 | 0.59 | -0.01 |
| S_V | Stream | 0.73 | -0.09 |

**Table A2.** Spatial information about the piezometers.

| Site Name | Sensor Name | Ground Level | Sensor Level | Distance From Stream |
| --- | --- | --- | --- | --- |
| [-] | [-] | [m] | [m] | [m] |
| M_D | Piezo1 | 1.324 | -0.476 | 10.1 |
| M_D | Piezo2 | 1.012 | -0.836 | 2.4 |
| M_D | Piezo3 | 0.945 | -0.593 | 3.4 |
| M_D | Piezo4 | 2.206 | -0.972 | -2.5 |
| M_D | Stream | 0.000 | -0.514 | 0.0 |
| M_J | Piezo1 | 2.373 | 1.443 | 13.3 |
| M_J | Piezo2 | 1.692 | 0.027 | 3.8 |
| M_J | Piezo3 | 1.740 | 0.430 | 3.3 |
| M_J | Stream | 0.000 | -0.465 | 0.0 |
| M_K | Piezo1 | 3.810 | 2.880 | 13.1 |
| M_K | Piezo2 | 3.095 | 2.175 | 4.0 |
| M_K | Piezo3 | 2.961 | 1.481 | 6.4 |
| M_K | Stream | 0.000 | -0.230 | 0.0 |
| S_J | Piezo1 | 2.304 | 1.101 | 8.7 |
| S_J | Piezo2 | 1.460 | 0.917 | 5.0 |
| S_J | Piezo3 | 1.319 | 0.511 | 4.5 |
| S_J | Piezo4 | 1.419 | 0.206 | -4.5 |
| S_J | Stream | 0.000 | -0.300 | 0.0 |
| S_V | Piezo1 | 3.510 | 1.540 | 14.6 |
| S_V | Piezo2 | 1.551 | -0.019 | 7.6 |
| S_V | Piezo3 | 0.686 | -0.814 | 3.6 |
| S_V | Piezo4 | 0.747 | -0.413 | 2.2 |
| S_V | Stream | 0.000 | -0.650 | 0.0 |

Table A3: Soil horizons of piezometers.

| Cluster | Piezometer | Horizon | Depth |
|---------|------------|---------|-------|
| [-] | [-] | [-] | [$cm$] |
| M_D | Piezo1 | Ap | -30 |
| | | B1 | -65 |
| | | B2 | -100 |
| | | B3 | -122 |
| | | B4 | -178 |
| | | Cv | |
| | Piezo2 | Ah | -5 |
| | | B1 | -30 |
| | | B2 | -50 |
| | | B3 | -110 |
| | | B3 | -125 |
| | | B3 | -155 |
| | | B4 | -165 |
| | Piezo3 | Ah | -13 |
| | | B1 | -35 |
| | | B2 | -55 |
| | | B3 | -162 |
| | | Cv | |
| | Piezo4 | Ah | -4 |
| | | B1 | -121 |
| | | B2 | -186 |
| | | B3 | -246 |
| | | B4 | -313 |
| | | B5 | -335 |
| | | C | |
| M_J | Piezo1 | Ah | -20 |
| | | B1 | -70 |
| | | B2 | -95 |
| | | B2.2 | -112 |
| | | B3 | -142 |
| | | B3.2 | -150 |

| Cluster | Piezometer | Horizon | Depth |
|---------|-----------|---------|-------|
| [-]     | [-]       | [-]     | [cm]  |
|         |           | B4      | -170  |
|         | Piezo2    | Ah      | -9    |
|         |           | B1      | -45   |
|         |           | B2      | -83   |
|         |           | B3      |       |
|         | Piezo3    | Ah      | -10   |
|         |           | B1      | -41   |
|         |           | B2      | -60   |
|         |           | B3      |       |
|         | Piezo4    | B4      | -50   |
| M_K     | Piezo1    | Ah      | -12   |
|         |           | B1      | -30   |
|         |           | B2      | -50   |
|         |           | B3      | -97   |
|         | Piezo2    | Ah      | -15   |
|         |           | B1      | -35   |
|         |           | B2      | -93   |
|         | Piezo3    | Ah      | -13   |
|         |           | B1      | -35   |
|         |           | B2      | -91   |
|         | Piezo4    | Ah      | -8    |
|         |           | B1      | -45   |
|         |           | B2      | -85   |
| S_J     | Piezo1    | Ah      | -7    |
|         |           | B       | -88   |
|         |           | Cv1     | -110  |
|         |           | Cv2     | >114  |
|         | Piezo2    | Ah      | -3    |
|         |           | B       | -34   |
|         |           | B2      | -59   |
|         |           | Cv      | >59   |
|         | Piezo3    | Ah      | -9    |

| Cluster | Piezometer | Horizon | Depth |
|---------|-----------|---------|-------|
| [-]     | [-]       | [-]     | [$cm$] |
|         |           | B       | -35   |
|         |           | B2      | -58   |
|         |           | Cv      | >85   |
|         | Piezo4    | Ah      | -20   |
|         |           | B       | -72   |
|         |           | Cv      | -117  |
|         |           | Cv2     | >117  |
| S_V     | Piezo1    | Ah      | -12   |
|         |           | B1      | -50   |
|         |           | B2      | -80   |
|         |           | B3      | -132  |
|         |           | Cv1     | -160  |
|         |           | Cv2     |       |
|         | Piezo2    | Ah      | -11   |
|         |           | B1      | -58   |
|         |           | Bv      | -86   |
|         |           | B3      |       |
|         | Piezo3    | Ah      | -13   |
|         |           | B1      | -62   |
|         |           | B2      |       |
|         | Piezo4    | Ah      | -14   |
|         |           | Rock    | -24   |
|         |           | B       | -81   |
|         |           | Cv      |       |

**Appendix B:  Visualization of information loss when monitoring only one year instead of multiple years**

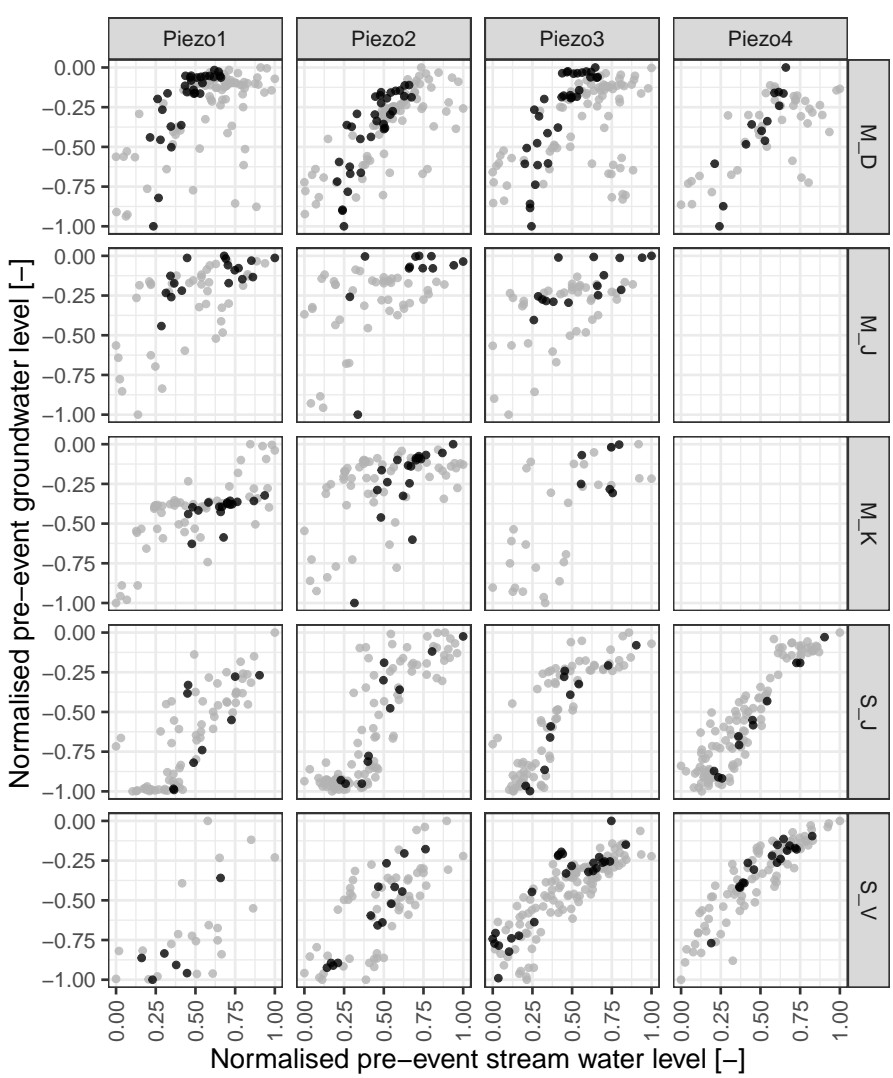

**Figure B1.** Normalised stream and groundwater levels. Black events are from 2015 while grey events are from all other years.

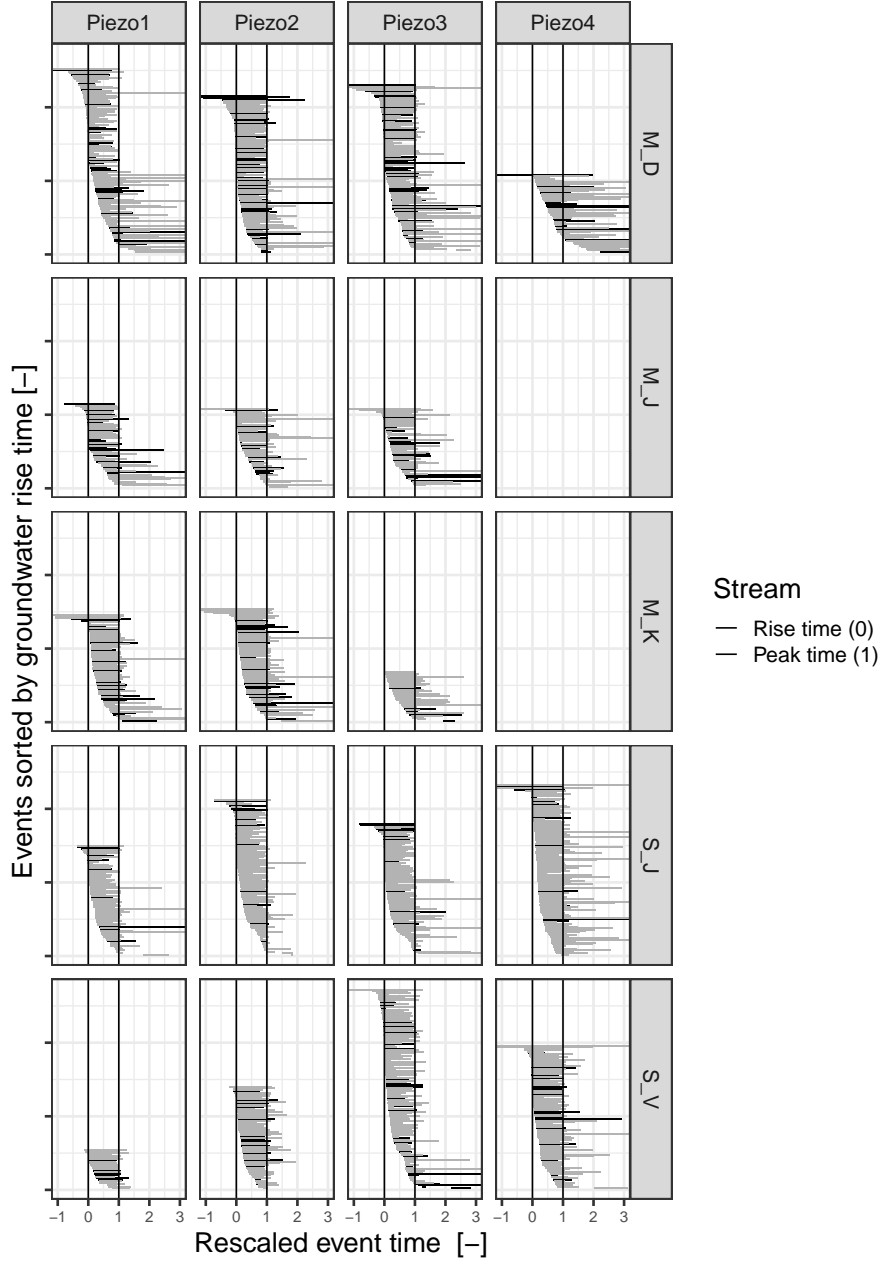

**Figure B2.** Timing of groundwater response relative to stream response. Black events are from 2015 while grey events are from all other years.

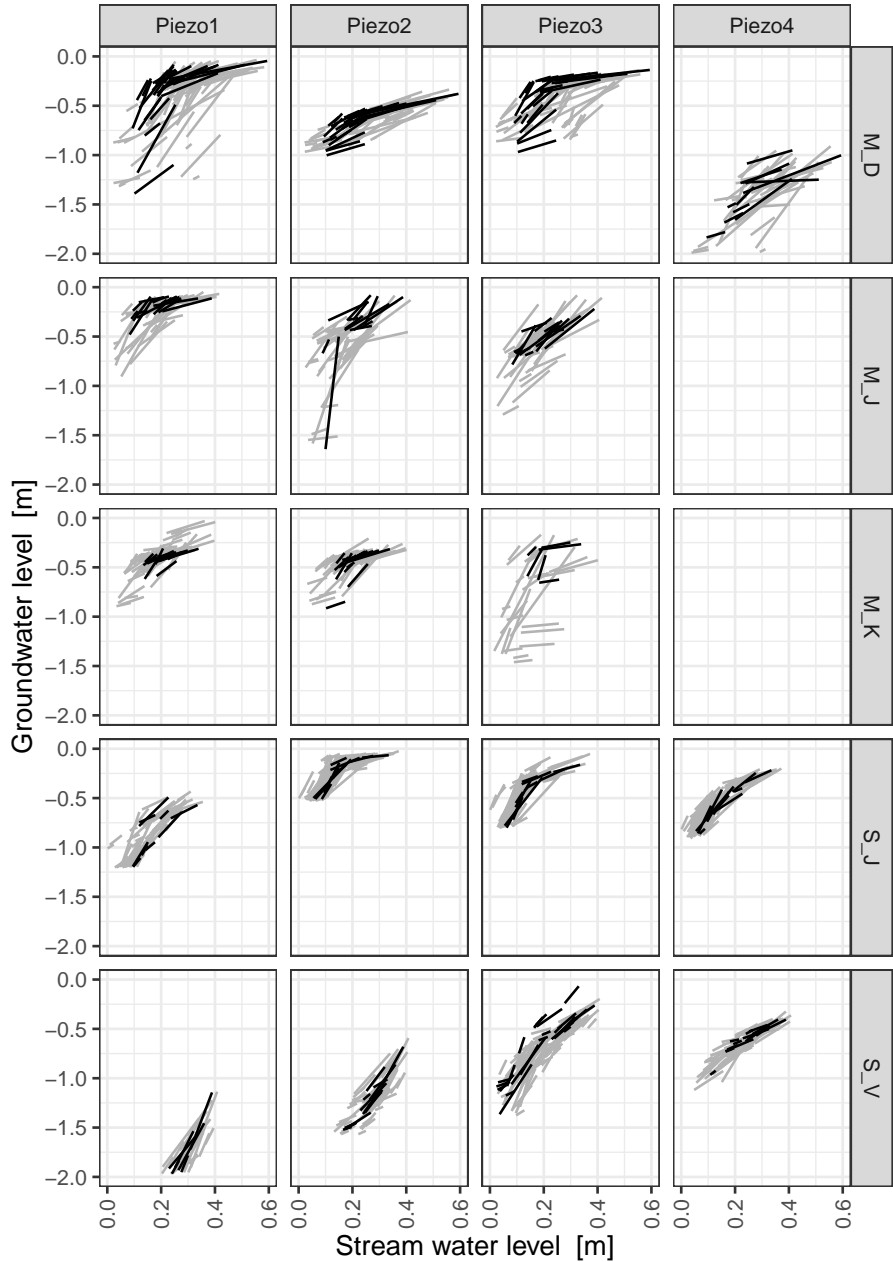

**Figure B3.** Responses in water tables of groundwater (y-axis) and the corresponding stream (x-axis). Black events are from 2015 while grey events are from all other years.

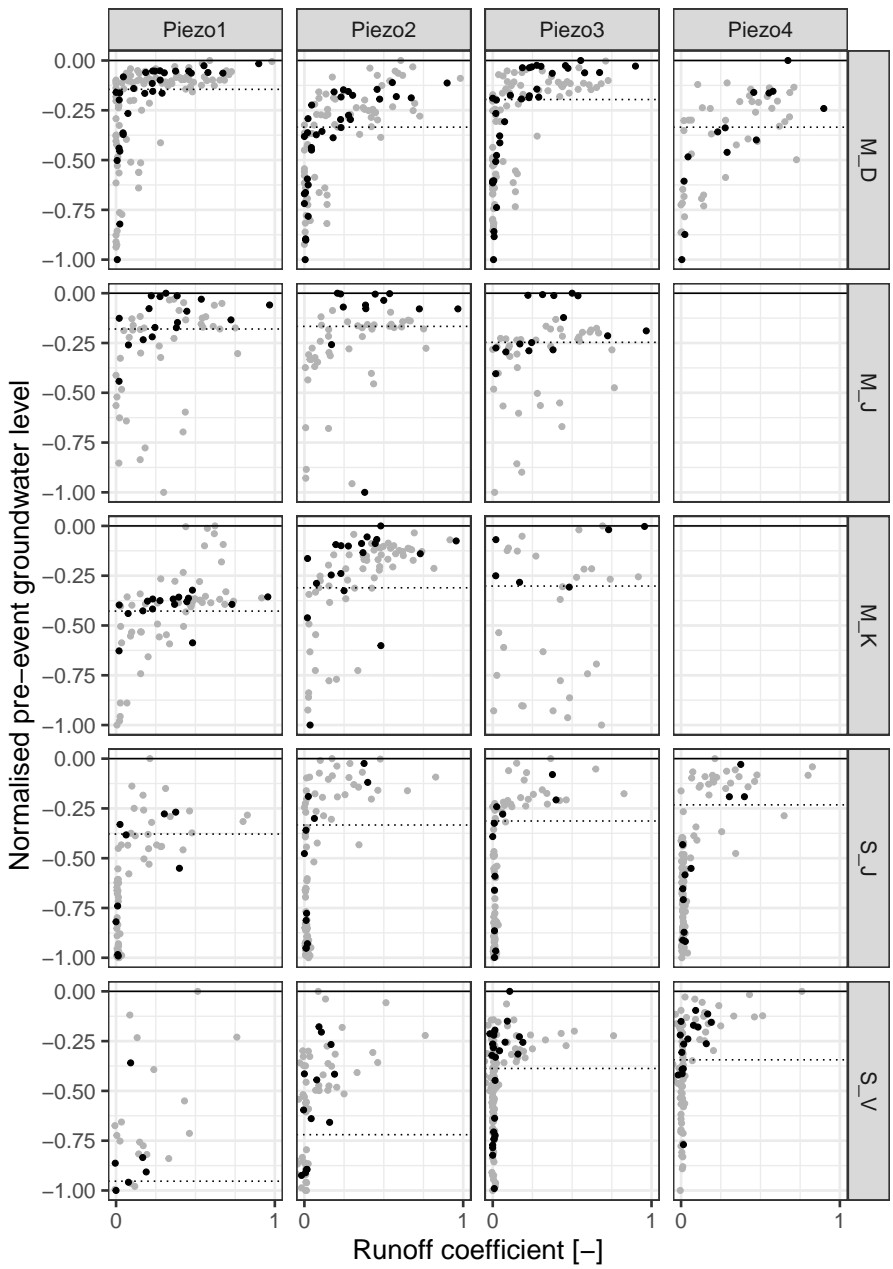

**Figure B4.** Event runoff coefficients versus shallow groundwater levels. Black events are from 2015 while grey events are from all other years.

*Competing interests.* None of the authors have financial competing interests, however, authors Weiler and Blume are members of the editorial board of the journal.

*Data availability.* Data will be made available in a corresponding data publication in ESSD.

*Acknowledgements.* We gratefully acknowledge DFG for the research funding (FOR 1598) and thank the group of Laurent Pfister at the Luxembourg Institut of Technology (LIST) for providing the discharge data, AgriMeteo for the precipitation data, Matthias Sprenger for the soil horizon data, and Tobias Vetter and Britta Kattenstroth for their untiring maintenance of the monitoring sites.

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
