# Peer review of "Characterising hillslope-stream connectivity with a joint event analysis of stream and groundwater levels"

_Hydrology and Earth System Sciences, 2020_

## Referee Comment (RC1) · Anonymous Referee #1 · 6 Mar 2020

**General comment**

This manuscript focuses on the characterization of hillslope-stream connectivity by using a novel joint event analysis of the response of stream and shallow groundwater levels. The authors examined the response timing of 18 groundwater sites located in five different footslopes in Luxembourg for 706 runoff events. The applied methodology included event detection, the quantification of response timing of groundwater compared to stream water level, the analysis of the relations between pre-event groundwater level with pre-event stream water level and runoff coefficient. The authors concluded that the joint analysis of groundwater and stream water levels provided information on the

Creative Commons BY license logo

presence or absence, and on the degree of subsurface hillslope-stream connectivity. The found threshold relations between groundwater and stream water levels were interpreted as transmissivity feedback in the marls study sites, and fill-and-spill in the schist areas. The topic of this manuscript is of interest for the readers of the journal, and overall the paper is well written and structured. The presented analysis for such a large time series of groundwater levels is quite rare, and therefore is particularly important to advance our comprehension of hillslope-stream subsurface connectivity. Nonetheless, I have some specific questions/comments for the authors, and I would like to see integrated in the manuscript some more methodological details.

Specific comments

1) I suggest to the authors to clearly provide in the introduction the definition of subsurface hydrologic connectivity, that they considered (currently such a definition can only be guessed by the readers).

2) The authors mentioned in the abstract that they performed their joint analysis for rainfall-runoff events, but throughout the manuscript there is no description of the rainfall characteristics (e.g., total rainfall, intensities and duration of the selected events) and where they were monitored (are the weather stations located in the study catchments?). I suggest to report such details in the text. Furthermore, I would like to see a table presenting the main summary statistics for rainfall, runoff and groundwater characteristics of the considered events.

3) Since the analysis was carried out for the whole time series (winters and early spring included), I am wondering whether there were snowfalls, and if the authors considered snowmelt-induced runoff events and rain-on-snow events in the analysis. If such events were discarded, I suggest to integrate the description of the methodological approach for event detection. Otherwise, the authors should clearly state that they focused only on rainfall-runoff events.

4) In Table 1 (or in a new table), I suggest to provide the topographic characteristics

of the groundwater sites together with their depth. These details could help to understand whether the topography is very similar (or very different) among the monitored locations, and to support the discussion at page 17, lines 15-19. Moreover, what is the extension of the riparian zone compared to the hillslopes in the study sites?

5) Have the authors considered their analysis of subsurface connectivity in light of recent findings by Klaus and Jackson (2018) and Gabrielli and McDonnell (2020)? Are there bedrock permeability data for the selected study sites?

6) In the section "2.4 Event detection" and Fig. 4, it is not clear which response timings were considered for complex events with multiple peaks (both in stream and groundwater level). Furthermore, which peak in stream water level is considered if there is only one peak for the groundwater level?

7) Page 11, line 5: Please provide a reference for the method used for the stormflow calculation.

8) Page 20, line 5: "No pronounced differences...": could the authors report the results of the applied statistical test?

9) Page 21, lines 3-4: Please provide more details about the investigated relations between rainfall characteristics and event responses.

10) Page 22, lines 8-10: Please remove these details from the available literature, and report them in a table. Please consider that other recent studies examined almost or more than 100 events (e.g., Rinderer et al., 2016; Zuecco et al., 2019).

11) Page 23, line 13-16: The example of considering just two events in the data analysis is a very extreme case, and so far I have never seen it. Therefore, please revise the sentence. The main question is how many events and piezometers do we need to capture the temporal and spatial variability of subsurface connectivity?

Technical corrections

1) Page 2, line 25: "assess".

2) Page 3, line 1: "hillslope" instead of "slope".

3) Page 4, line 2: "July".

4) Page 23, line 4: "these" instead of "this".

5) Figure 11: Based on the caption, the label of the y axis should be "Normalised pre-event groundwater level".

References

Gabrielli C.P., McDonnell J.J., 2020. Modifying the Jackson index to quantify the relationship between geology, landscape structure and water transit time in steep wet headwaters. Hydrological Processes, early view. DOI: 10.1002/hyp.13700

Klaus J., Jackson C.R., 2018. Interflow is not binary: a continuous shallow perched layer does not imply continuous connectivity. Water Resources Research, 54, 5921-5932. DOI: 10.1029/2018WR022920

Rinderer M., van Meerveld H.J., Stähli M., Seibert J., 2016. Is groundwater response timing in a pre-alpine catchment controlled more by topography or by rainfall? Hydrological Processes, 30, 1036-1051. DOI: 10.1002/hyp.10634

Zuecco G., Rinderer M., Penna D., Borga M., van Meerveld H.J., 2019. Quantification of subsurface hydrologic connectivity in four headwater catchments using graph theory. Science of the Total Environment, 646, 1265-1280. DOI: 10.1016/j.scitotenv.2018.07.269

---

## Referee Comment (RC2) · Anonymous Referee #2 · 6 Mar 2020

The manuscript by Beiter et al details a study aimed at understanding hillslope-stream connectivity. They collected 5-6 years of paired near-stream groundwater and stream water levels at five locations within an agricultural catchment in western Luxembourg. At each site, shallow groundwater levels were logged at 5 minute intervals at 3 to 4 piezometers located within 15 m of the stream level site. They extracted about 150 individual rainfall-runoff events from the data record using an approach that interrogates the stream water level time series. For each event identified, they also extracted groundwater response metrics from the corresponding piezometers. They compared stream and groundwater responses to quantify temporal changes in hillslope-stream connectivity. They found a threshold-type response in stream water level linked to an-

tecedent groundwater levels. Low antecedent groundwater levels were associated with variable stream water level responses. In contrast, high antecedent groundwater levels were associated with more consistent stream water level responses. They speculate that the hydrologic processes controlling these patterns were transmissivity feedback at the marls sites and fill and spill at the schist sites.

The topic covered in this manuscript is appropriate for HESS. The study contains an impressive data set and some of the visualizations do a great job of showing these data (e.g., Figure 8 and 9). Overall, the writing is not bad, but some of the grammar is confusing which makes it difficult to understand some of the elements of the paper. Given the amount of data, I'm left feeling a little underwhelmed by the key conclusions. This might reflect the vagueness of the key research questions (page 3, lines 12-15). For example, 'provide information' is a very general statement - try to be more specific about what is learned from this sort of joint analyses. I would encourage the authors to formulate testable hypotheses to help add more structure to the manuscript. This would also help clarify the key findings of this study.

Overall, I agree with reviewer #1's assessment, so I'll try not to repeat things here. I outline one general comment, followed by some more specific comments.

A major strength of this study is looking at the temporal dynamics of hillslope-stream connectivity. In contrast, the study is limited in capturing spatial variability in hillslope-stream connectivity. However, a key question/conclusion of the study concerns whether connectivity can be assessed using a single groundwater piezometer. The authors conclude that 'a single, well chosen, piezometer can already provide substantial information on catchment state...'. How do we know when a location is well chosen? We aren't provided any guidance on this. It is recognized that hillslope-stream connectivity can be spatially variable. How were the locations of the five sites selected? How representative are these locations of subsurface connectivity at other locations within the catchments?

Specific comments:

Abstract: This feels very long for an abstract. With some good editing this could be reduced by half.

P2L6: 'variable in space': Exactly - but this is not well addressed in this study.

P2L9: 'Full connectivity': What is meant here? For the entire catchment or hillslope? Some indication of the spatial scale of interest should be made in this introduction.

Introduction: As reviewer #1 highlights, some of the more recent research on this topic should be discussed here or in the discussion (e.g., Klaus and Jackson 2019 WRR, Gabrielli and McDonnell 2020 HP).

P3L1-3: An example of a run-on sentence that should be avoided.

P3L8: What is meant by a 'rough interpretation'?

Section 2.1: Provide some information about soils and the vegetation cover. Could the sites be given more descriptive names? I realize the 'S' and 'M' represent the dominant geology, but what do 'J', 'V', 'D', and 'K' represent?

Figure 2: Could elevation be added to these plots? Or at least the elevation of the ground surface at the peizometer and the depth of the piezometer relative to the streambed? It might be really helpful to include photographs of the 5 site installations so that the readers can get a better sense of the sites.

Table 1: What are the slope quartiles referring to? The hillslope or catchment?

P6L4: Where is the Roodt station? Any concerns about spatial variability in precipitation inputs? I know it is mentioned that precipitation is assumed to be uniform across the catchments for the runoff ratios; however, it seems like not all stream water level sites respond to precipitation events. This may suggest that the uniform precipitation assumption is not reasonable.

P6L21-22: Looks like this percentage was tested? How sensitive are the results to different values?

Section 2.4: Why not conduct the event detection by using the precipitation record (as is frequently done) instead of the stream level records?

P8L1: Were there times when the piezometers showed a response but not the stream?

P8L15: This would only happen in autumn?

P8L18-22: For the search interval, was this a moving window or fixed interval search?

P10L5-14: Consider re-writing this to improve clarity.

P10L13: What is meant by 'hints'?

P10L26: What is meant by 'a more or less deterministic increase'?

P11L26: For what purpose is this response considered negligible?

Figure 5 (and others): The 'Event type' colour scale is very difficult to interpret for a colour-blind person. Consider using some other way to visualize these data (shapes maybe, although that might be difficult to see as well)?

Figure 7: Perhaps distinguish the Seasons by shape instead of colour.

Figure 8: Very nice graph!

Figure 9: Could the approach used to set those thresholds be discussed a bit more? I realize they were done visually, but there are some sites/piezometers that I would argue don't have a clear threshold (most of S_V, S_J piezo1, most of M_K, etc.).

Section 3.6: Please define 'catchment state' - this seems to appear out of nowhere (unless I missed it earlier).

P17L15: It's not clear to me where the topographic characteristics come from? Is this simply the qualitative discussion in Section 2.1. Are there stream incision data for all

the sites?

Figure 10: Appears that the figure caption for the y axis is incorrect.

P19L1-3: I'm struggling with this logical leap between the results shown in Figure 7 and how they 'indicate that well-placed groundwater observation points can be representative of the given footslope, at least for pre-event conditions'. Given the close proximity of the within-site piezometers, there seems to be a surprising amount of scatter in these plots.

P19L8-10: Or could it be that another portion of the catchment is connected, but not the hillslope with the piezometers?

P21L28-31: How is it known that connectivity 'does not extend far up the slopes' when those observations were not made? The substantial conclusions in this section are based on somewhat subjective placement of a threshold. It could even be argued that no clear threshold exists for some of the sites (see comment regarding Figure 9 above).

P22L1-2: Again, I'm not clear on where the evidence is for this statement.

---

## Referee Comment (RC3) · Anonymous Referee #3 · 8 Mar 2020

The authors studied the connectivity of hillslope to stream water using an impressive amount of data from 5 different catchments. The catchments studied were divided in two different geologies. The methodology they chose to use was focused in groundwater levels in piezometers near the streams and stream levels to try simplifying the hillslope approach that is often used when searching for connectivity with the stream. The study is interesting and I only have a few comments and questions to the authors adding to reviewers 1 and 2. I will now follow with some general comments and later into more specific comments:

- One of the goals of the study was to test if assessing the connectivity between hillslope and stream could be done with a single shallow near stream piezometer. I see your results show the answer is yes, but you miss to discuss or analyze why it worked in some piezometers and why not in others. What could you do different? In the discussions you mention that it works if you use a single well-chosen piezometer, how did you manage to have badly chosen piezometers in your network?

- Related to the previous comment, is there any information about the soil profiles of the piezometers? Or were they installed blindly with the cobra? What do you mean with refusal? Is that refusal as when you reach rocks/bedrock? Or would refusal count as well as when you reach a clayey layer that could divide two aquifers (valid for the sandy soils)? Is there information on the elevation of the piezometers related to the stream/streambed?

Following come some specific comments:

- Page 1 Line 7: I suggest modifying the text here or earlier, "Step two" comes as a surprise since there was never a step one.

- P 4 L 3-12: This paragraph could be friendlier and provide more information if it was shown as well in a set of tables for the piezometers in each catchment. Stating elevation over stream bed, well depth and distance from the stream among other things. Maybe just as supplementary data, but it would help the reader visualize the piezometers better.

- P 4 L 15: I mentioned it before, but does this mean that there is no information on the soil profiles?

- P 5 Figure 2: I like this Figure and the information it provides. But I do agree with reviewer 2, either add data here or on the tables I mentioned two comments ago.

- P 7 Figure 3: I suggest you improve the horizontal lines that come from 'hfallThreshold' and 'hpostAmplitude' because they are hard to see in the current version.

- P 8 L 9: "...are presumed to be rather short,". Is there any data on hydraulic conductivity?

- P 8 L 34: I suggest changing All NA to allNA, as you used the term allNA in all other instances.

- P 10 L 9-10: I would suggest to rephrase this sentence and say directly what you did instead of saying first what you did not do. It would make it easier to read.

- P 11 L 7: why did you consider it sufficiently representative? Could you provide more information to the reader? How far is it located? Similar elevation?

- P 12 L 2: Here you refer to Figure 9 before referring to Figure 8, maybe move Figure 9 to position 8 or change the text.

- P 12 L 3-4: "...low in summer and autumn." I don't see this generalization when I see Figure 7. There are several piezometers were autumn covers the whole spectrum. Or is there median values that we have no knowledge of?

- P 19 L 1-3: Any insights on why you had some exceptions? Because if those were your only wells this study would have completely different conclusions. If you have no insights, then that is valid as well, but it should be stated.

- P 22 Table 2: Use capital letters in each of the boxes, you used in some boxes but not all.

- P 22 L 1: "..single well chosen well..". I agree, but how did you choose well or bad? Assuming you had installed your piezometers.

- P 23 L 1: The closing ) is missing after Figure 10.

---

## Author Comment (AC1) · 28 Apr 2020

Reviewer 1:

General comment

This manuscript focuses on the characterization of hillslope-stream connectivity by using a novel joint event analysis of the response of stream and shallow groundwater levels. The authors examined the response timing of 18 groundwater sites located in five different footslopes in Luxembourg for 706 runoff events. The applied methodology included event detection, the quantification of response timing of groundwater compared

to stream water level, the analysis of the relations between pre-event groundwater level with pre-event stream water level and runoff coefficient. The authors concluded that the joint analysis of groundwater and stream water levels provided information on the presence or absence, and on the degree of subsurface hillslope-stream connectivity. The found threshold relations between groundwater and stream water levels were interpreted as transmissivity feedback in the marls study sites, and fill-and-spill in the schist areas. The topic of this manuscript is of interest for the readers of the journal, and overall the paper is well written and structured. The presented analysis for such a large time series of groundwater levels is quite rare, and therefore is particularly important to advance our comprehension of hillslope-stream subsurface connectivity. Nonetheless, I have some specific questions/comments for the authors, and I would like to see integrated in the manuscript some more methodological details.

Answer: We thank the reviewer for taking the time to review our manuscript and are happy to see this positive assessment.

Specific comments

1. I suggest to the authors to clearly provide in the introduction the definition of subsurface hydrologic connectivity, that they considered (currently such a definition can only be guessed by the readers).

Answer: We agree and will add the following definition to the introduction: Ali and Roy (2009) and Bracken et al. (2013) collected various definitions of hydrologic connectivity used in previous studies, which differ in spatial scale (hillslope vs watershed) and observed features (e.g. water cycle or landscape). The most appropriate definition in the context of our investigation is via flow processes on the hillslope scale, where disparate regions on a hillslope are linked via lateral subsurface water flow (Hornberger et al., 1994, Creed and Band, 1998).

2. The authors mentioned in the abstract that they performed their joint analysis for rainfall-runoff events, but throughout the manuscript there is no description of the rainfall characteristics (e.g., total rainfall, intensities and duration of the selected events)and where they were monitored (are the weather stations located in the study catchments?). I suggest to report such details in the text. Furthermore, I would like to see a table presenting the main summary statistics for rainfall, runoff and groundwater characteristics of the considered events.

Answer: More details about precipitation will be added to the revised manuscript in the form of gauge locations (map), summary statistics (table) and description in the text.

3. Since the analysis was carried out for the whole time series (winters and early spring included), I am wondering whether there were snowfalls, and if the authors considered snowmelt-induced runoff events and rain-on-snow events in the analysis. If such events were discarded, I suggest to integrate the description of the methodological approach for event detection. Otherwise, the authors should clearly state that they focused only on rainfall-runoff events.

Answer: We agree that events influenced by snow fall or snow melt could impact this analysis. However, snow fall events are generally quite rare in Luxembourg, so the number of events affected is assumed to be low. Furthermore, as the analysis is based on streamflow response, snow fall events themselves will not appear in our analysis. A major snow melt event or rain on snow event would be captured by its runoff response, but in this case erroneous estimate of rainfall input would only impact the analysis of runoff coefficients as we otherwise focus our analyses on the relationship between streamflow and groundwater responses. We will add a sentence explaining this to the revised manuscript.

4. In Table 1 (or in a new table), I suggest to provide the topographic characteristics of the groundwater sites together with their depth. These details could help to understand whether the topography is very similar (or very different) among the monitored locations, and to support the discussion at page 17, lines 15-19. Moreover, what is the extension of the riparian zone compared to the hillslopes in the study sites?

Answer: We agree that this information would be helpful. Unfortunately, the available topographic data only has a resolution of 10m, which does not give any additional information on the sites as the piezometer distances are around 10m. However, we will provide average slopes for the measurement locations in addition to the average slopes of the subcatchments. The spatial extent of the riparian zone varies between the five sites but was estimated to be around 2-10m, depending on topography. The adjacent hillslopes derived from the DEM were at least 250m long, except M_D with around 50m.

5. Have the authors considered their analysis of subsurface connectivity in light of recent findings by Klaus and Jackson (2018) and Gabrielli and McDonnell (2020)? Are there bedrock permeability data for the selected study sites?

Answer: Unfortunately, we do not have information on bedrock permeability. Klaus and Jackson (2018) found that according to the Downslope Travel Distances (DTD) only lower regions of a hillslope contribute to the streamflow via interflow, while in upper regions water percolates into the deeper groundwater. We do agree that the presence of a perched groundwater table at the footslope is no proof for a connected hillslope. However, we can observe a threshold behaviour in the hillslope-stream-system that depends on initial groundwater levels. This indicates that soil characteristics (flow path system, layering) start playing a role in how water parcels travel along the hillslope. It is very likely that these heterogeneities allow a hillslope or at least the footslope to connect to the stream via interflow for a short period of time. Gabrielli and McDonnell (2020) built upon the DTD and developed a (gridded) Index which describes the general tendency of a catchment to either shed water laterally to the stream channel or infiltrate water to depth. They found high correlations between their Anisotropy Index (AI) and the assessed Mean Transit Time (MTT) for several catchments. As we are focussing on the hillslope-stream connectivity we did not make any statement towards the water age. We will add a brief discussion of these papers to the revised manuscript.

5. In the section "2.4 Event detection" and Fig. 4, it is not clear which response timings

were considered for complex events with multiple peaks (both in stream and groundwater level). Furthermore, which peak in stream water level is considered if there is only one peak for the groundwater level?

Answer: Thank you for pointing this out. This will be clarified in the text. After merging overlapping event timings, the highest peak is determined as the event maximum. From there on it is handled as a simple event according to Figure 3. This also accounts for the situation where multiple overlapping streams events correspond with a single groundwater level event.

6. Page 11, line 5: Please provide a reference for the method used for the stormflow calculation.

Answer: The approach to separate baseflow from discharge was developed in the style of the constant slope method (Dingman, 2002).

7. Page 20, line 5: "No pronounced differences...": could the authors report the results of the applied statistical test?

Answer: This statement is related to the timing patterns in Figure 8. It illustrates several dimensions of information such as rising and maximum time of a groundwater event, normalised by the according stream event, where each piezometer has a different number of events. This makes it difficult to apply statistical tests to see whether two groups of samples are significantly different from each other. Furthermore, a p-value below a common significance level of alpha = 0.05 would not tell how pronounced differences would be.

8. Page 21, lines 3-4: Please provide more details about the investigated relations between rainfall characteristics and event responses.

Answer: The relationship between rainfall characteristics and event responses is not a major focus of this study and we therefore touched on it only briefly here. However, we agree that this brief glimpse of the analyses might be unsatisfactory and we will add

the results of the correlation analyses in the supplementary data.

9. Page 22, lines 8-10: Please remove these details from the available literature, and report them in a table. Please consider that other recent studies examined almost or more than 100 events (e.g., Rinderer et al., 2016; Zuecco et al., 2019).

Answer: Thank you for pointing these studies out. We will revise the illustration of numbers of events.

10. Page 23, line 13-16: The example of considering just two events in the data analysis is a very extreme case, and so far I have never seen it. Therefore, please revise the sentence. The main question is how many events and piezometers do we need to capture the temporal and spatial variability of subsurface connectivity?

Answer: We agree that considering only two events is a very extreme case. We will remove this statement and will make it clearer, that the question is not so much about how many events are necessary (in absolute numbers) as more about the necessary time period to cover the temporal variability generated by different hydrological processes. It is therefore necessary to accumulate a large number of events across all seasons. In terms of extreme events (droughts/floods) the covered time period and number of events will need to be even higher, on the one hand to capture these events, and on the other hand to put them into context.

Technical corrections 1. Page 2, line 25: "assess". 2. Page 3, line 1: "hillslope" instead of "slope". 3. Page 4, line 2: "July". 4. Page 23, line 4: "these" instead of "this". 5. Figure 11: Based on the caption, the label of the y axis should be "Normalised pre-event groundwater level".

References Gabrielli C.P., McDonnell J.J., 2020. Modifying the Jackson index to quantify the re-lationship between geology, landscape structure and water transit time in steep wet headwaters. Hydrological Processes, early view. DOI: 10.1002/hyp.13700 Klaus J., Jackson C.R., 2018. Interflow is not binary: a continuous shallow perched

layer does not imply continuous connectivity. Water Resources Research, 54, 5921-5932. DOI: 10.1029/2018WR022920 Rinderer M., van Meerveld H.J., Stähli M., Seibert J., 2016. Is groundwater response timing in a pre-alpine catchment controlled more by topography or by rainfall? Hydro-logical Processes, 30, 1036-1051. DOI: 10.1002/hyp.10634 Zuecco G., Rinderer M., Penna D., Borga M., van Meerveld H.J., 2019. Quan-tification of subsurface hydrologic connectivity in four headwater catchments us-ing graph theory. Science of the Total Environment, 646, 1265-1280.DOI:10.1016/j.scitotenv.2018.07.269

Ali G. A., Roy A. G., 2009. Revisiting Hydrologic Sampling Strategies for an Accurate Assessment of Hydrologic Connectivity in Humid Temperate Systems. Geography Compass, 3, 350-374. DOI: 10.1111/j.1749-8198.2008.00180.x Bracken L., Wainwright J., Ali G., Tetzlaff D., Smith M., Reaney S., Roy, A., 2013. Concepts of hydrological connectivity: Research approaches, pathways and future agendas. Earth-Science Reviews, 119, 17 - 34. DOI: 10.1016/j.earscirev.2013.02.001

Hornberger G. M., Bencala K. E., McKnight D. M., 1994. Hydrological controls on the temporal variation of dissolved organic carbon in the snake river near montezuma, Colorado. Biogeochemistry 25, pp. 147–165. DOI: doi.org/10.1007/BF00024390

Creed I. F., Band L. E., 1998. Exploring functional similarity in the export of NitrateN from forested catchments: a mechanistic modeling approach. Water Resources Research 34, 11, pp. 3079–3093. DOI: 10.1029/98WR02102

S L Dingman, 2002. Physical hydrology. Upper Saddle River. ISBN-13: 978-1478611189

---

## Author Comment (AC2) · 28 Apr 2020

Reviewer 2:

The manuscript by Beiter et al details a study aimed at understanding hillslope-stream connectivity. They collected 5-6 years of paired near-stream groundwater and streamwater levels at five locations within an agricultural catchment in western Luxembourg. At each site, shallow groundwater levels were logged at 5 minute intervals at 3 to 4 piezometers located within 15 m of the stream level site. They extracted about 150 individual rainfall-runoff events from the data record using an approach that interrogates the stream water level time series. For each event identified, they also extracted

groundwater response metrics from the corresponding piezometers. They compared stream and groundwater responses to quantify temporal changes in hillslope-stream connectivity. They found a threshold-type response in stream water level linked to antecedent groundwater levels. Low antecedent groundwater levels were associated with variable stream water level responses. In contrast, high antecedent groundwater levels were associated with more consistent stream water level responses. They speculate that the hydrologic processes controlling these patterns were transmissivity feedback at the marls sites and fill and spill at the schist sites. The topic covered in this manuscript is appropriate for HESS. The study contains an impressive data set and some of the visualizations do a great job of showing these data (e.g., Figure 8 and 9). Overall, the writing is not bad, but some of the grammar is confusing which makes it difficult to understand some of the elements of the paper. Given the amount of data, I'm left feeling a little underwhelmed by the key conclusions. This might reflect the vagueness of the key research questions (page 3, lines 12-15).For example, 'provide information' is a very general statement - try to be more specific about what is learned from this sort of joint analyses. I would encourage the authors to formulate testable hypotheses to help add more structure to the manuscript. This would also help clarify the key findings of this study.

Answer: We thank the reviewer for taking the time to review our manuscript and are happy to see that Figures 8 and 9 are appreciated! We will sharpen the research question and conclusions by including the following hypotheses in the revised version of the manuscript: Hypothesis 1: hillslopes remain disconnected from the stream for most of the time and connect only during short periods of time. Hypothesis 2: marls and schist hillslope-stream systems differ in connectivity patterns as their soil properties and topography is quite different. Hypothesis 3: monitoring at the footslope can provide information on hillslope-stream connectivity at this location but also at the catchment scale.

Overall, I agree with reviewer #1's assessment, so I'll try not to repeat things here. I

outline one general comment, followed by some more specific comments.

A major strength of this study is looking at the temporal dynamics of hillslope-stream connectivity. In contrast, the study is limited in capturing spatial variability in hillslope-stream connectivity. However, a key question/conclusion of the study concerns whether connectivity can be assessed using a single groundwater piezometer. The authors conclude that 'a single, well chosen, piezometer can already provide substantial information on catchment state...'. How do we know when a location is well chosen? We aren't provided any guidance on this. It is recognized that hillslope-stream connectivity can be spatially variable. How were the locations of the five sites selected? How representative are these locations of subsurface connectivity at other locations within the catchments?

Answer: Thank you for pointing this out. That 'single, well-chosen piezometer' is related to the plot scale only. It is rather hypothetical in the sense that one cannot be sure without having multiple observations per footslope (e.g. three) first. The analysis showed that despite possible local heterogeneities (e.g. soil texture, tree roots) the distributed piezometers revealed very similar patterns regarding amplitudes and temporal responses and can therefore be considered representative for the hillslope. Also 'bad-chosen' piezometers can be identified by strongly disagreeing with the other piezometers. The selection of the five sites was mainly influenced by the different geologies (marls and schist) and their subsequent influence on soil, topography, hillslope morphology and potential land use. Within these geologies headwater catchments of different sizes were considered to cover the variability of such in the Attert catchment. From this point of view the selected sites can be considered representative for headwater catchments in the Attert catchment.

Specific comments:

Abstract: This feels very long for an abstract. With some good editing this could be reduced by half.
Answer: We will make an effort to shorten the abstract.

P2L6: 'variable in space': Exactly - but this is not well addressed in this study.

Answer: In this sentence of the introduction we were referring to the spatial variability of hillslope-stream connectivity (surface and subsurface) at the catchment scale. Our study covered some of this spatial variability of subsurface connectivity by comparing 5 different sites, each equipped with 3-5 piezometers. Of course, it would be great to have more sites and more piezometers but even with this setup (and the advantage of the long time series measured here) it was possible to see that there are typical response patterns per site and per geology – suggesting that for some purposes we can assess certain aspects of connectivity with "representative" measurements. We will add a more detailed explanation on this in the revised manuscript.

P2L9: 'Full connectivity': What is meant here? For the entire catchment or hillslope? Some indication of the spatial scale of interest should be made in this introduction.

Answer: We agree and we will revise the sentence to: Full connectivity across entire hillslopes or catchments is usually established only during brief periods of time (Freer et al., 2002;Ocampo et al., 2006; Haught and Meerveld, 2011; van Meerveld et al., 2015). We will also add the information on the scale of interest in the introduction in Hypothesis 3: monitoring at the footslope can provide information on hillslope-stream connectivity at this location but also at the catchment scale.

Introduction: As reviewer #1 highlights, some of the more recent research on this topic should be discussed here or in the discussion (e.g., Klaus and Jackson 2019 WRR,Gabrielli and McDonnell 2020 HP).

Answer: Klaus and Jackson (2018) found that according to the Downslope Travel Distances (DTD) only lower regions of a hillslope contribute to the streamflow via interflow, while in upper regions water percolates into the deeper groundwater. We do agree that the presence of a perched groundwater table at the footslope is no proof for a

connected hillslope. However, we can observe a threshold behaviour in the hillslope-stream-system that depends on initial groundwater levels. This indicates that soil characteristics (flow path system, layering) start playing a role in how water parcels travel along the hillslope. It is very likely that these heterogeneities allow a hillslope or at least the footslope to connect to the stream via interflow for a short period of time. Gabrielli and McDonnell (2020) built upon the DTD and developed a (gridded) Index which describes the general tendency of a catchment to either shed water laterally to the stream channel or infiltrate water to depth. They found high correlations between their Anisotropy Index (AI) and the assessed Mean Transit Time (MTT) for several catchments. As we are focussing on the hillslope-stream connectivity we did not make any statement towards the water age. We will add a brief discussion of these papers to the revised manuscript.

P3L1-3: An example of a run-on sentence that should be avoided.

Answer: We agree. The sentence in question was: "We hypothesise that monitoring shallow groundwater tables in the riparian zone over longer periods of time and thus not only a few, but a large number of events will provide not full, but representative information on hillslope-stream connectivity at low cost. " We will change this to: "Monitoring shallow groundwater tables in the riparian zone over longer periods of time will allow us to capture a large number of events. We hypothesize that the analysis of these events will provide not full, but representative information on hillslope-stream connectivity."

P3L8: What is meant by a 'rough interpretation'?

Answer: Analysing the relationship between responses in near-stream shallow groundwater and stream thus permits us to determine the dominant processes. We will clarify this in the revised manuscript.

Section 2.1: Provide some information about soils and the vegetation cover. Could the sites be given more descriptive names? I realize the 'S' and 'M' represent the dominant geology, but what do 'J', 'V', 'D', and 'K' represent?
Answer: Aggregated information about the various sites (soil, land use, etc.) is provided in Table 1. The letters refer to a larger scale reference system of 45 sensor clusters distributed in the Attert Catchment and do not have a meaning per se. A full list of the sites can be found in Appendix A of Demand et al. 2019.

Figure 2: Could elevation be added to these plots? Or at least the elevation of the ground surface at the piezometer and the depth of the piezometer relative to the streambed? It might be really helpful to include photographs of the 5 site installations so that the readers can get a better sense of the sites.

Answer: Photographs of the 5 sites can be added. However they focus on the setup on the hillslope rather than the topography between hillslope and stream. We will take into consideration adding these photos in the supplement for a general idea of the sites. For a better topographic overview at the sites we will add elevation information to Figure 2 in the revised version.

Table 1: What are the slope quartiles referring to? The hillslope or catchment?

Answer: Since a detailed DEM was not available, the slope quartiles were calculated for the subcatchment of each stream level gauge (see little topographic maps in Figure 1) to give aggregated information about the topography for each site. We will clarify this in the revised manuscript

P6L4: Where is the Roodt station? Any concerns about spatial variability in precipitation inputs? I know it is mentioned that precipitation is assumed to be uniform across the catchments for the runoff ratios; however, it seems like not all stream water level sites respond to precipitation events. This may suggest that the uniform precipitation assumption is not reasonable.

Answer: The location of Roodt station will be added to the map. Precipitation may vary to some extent over the entire catchment. However, there are indicators supporting the assumption that it is sufficiently uniform for our analyses. While we see some events

which do not occur at all sites simultaneously, but these are mainly due data gaps for the respective stream gauge. Additionally, from all detected stream events those without a previous precipitation event observed at the Roodt station were removed. The number of removed stream events due to non-existing precipitation events was mostly below 10. Nevertheless, we reran the analysis for the three marls sites with data from the precipitation station at Useldange (<5km distance to marls sites) and the results are almost entirely the same. There is a very small shift of 1-2 events per site which are now (not) detected. Also the runoff coefficients patterns remain the same. We will add this explanation to the revised manuscript.

P6L21-22: Looks like this percentage was tested? How sensitive are the results to different values?

Answer: No formal test was applied. Sensitivity regarding the timing was higher when the percentages were lower due to the extended onset and offset. Higher thresholds were in general less sensitive and even more for the onset compared to the offset.

Section 2.4: Why not conduct the event detection by using the precipitation record (as is frequently done) instead of the stream level records?

Answer: The general idea was to design a stream-centred approach in order to focus on the response and its relation to the hillslope. Starting with the precipitation events would have also required us to formulate a definition for whether or not a stream event was observed. From our point of view, using event detection on the precipitation time series would not have helped investigating the interaction between hillslope and stream responses.

P8L1: Were there times when the piezometers showed a response but not the stream?

Answer: This was not investigated, due to the way the analysis was set up: we first identified the stream flow events and then used these events to check for groundwater responses.

P8L15: This would only happen in autumn?

Answer: The wetting-up phase is generally in autumn since groundwater levels are low in summer and high in winter. In particular during this period one can observe stream/piezometer events where the post-event water level lies above the pre-event water level. However, on shorter time scales wetting-up can also occur in other seasons.

P8L18-22: For the search interval, was this a moving window or fixed interval search?

Answer: The search intervals are of fixed length starting at the peak time with 24h in direction pre-peak and 48h in direction post-peak. We will clarify this in the revised manuscript.

P10L5-14: Consider re-writing this to improve clarity.

Answer: Will be revised.

P10L13: What is meant by 'hints'?

Answer: To prove causality in terms of subsurface flow one would need tracer observations. It is the many snapshots approach that – despite the lack of real proof for causality – enables the assumption of causality. This is what is meant by hint. We will clarify this in the revised manuscript.

P10L26: What is meant by 'a more or less deterministic increase'?

Answer: This refers to the assumption that two very similar groundwater responses would have two stream water responses that are also very similar to each other. This would reflect a deterministic system where the observation of one could be used to predict the other. We will clarify this in the revised manuscript.

P11L26: For what purpose is this response considered negligible?

Answer: Thank you for pointing that out. This refers to the performance of the detection

method. If the number of noLocalMaximum events would be high the whole approach would need to be questioned. This will be re-phrased to make it clearer.

Figure 5 (and others): The 'Event type' colour scale is very difficult to interpret for a colour-blind person. Consider using some other way to visualize these data (shapes maybe, although that might be difficult to see as well)?

Answer: We will make an effort to improve visibility for colour-blind persons.

Figure 7: Perhaps distinguish the Seasons by shape instead of colour.

Answer: We will adjust the Figure as suggested.

Figure 8: Very nice graph! Answer: Thank you!

Figure 9: Could the approach used to set those thresholds be discussed a bit more? I realize they were done visually, but there are some sites/piezometers that I would argue don't have a clear threshold (most of S_V, S_J piezo1, most of M_K, etc.).

Answer: It is true that thresholds were defined visually and for some piezometers this abrupt change in slope does not appear. In such cases a threshold was set to the level where the envelope functions (encompassing the bundle of slope lines) start converging again (S_J P1, S_V P3 and P4). For some piezometers the change in pattern was a sudden clustering of lines (M_K P1 and P2, S_V P2) above a certain value. All these observed changes in patterns signal that hydrologic processes do change due to different pre-event groundwater levels. We will add this explanation to the revised manuscript to clarify the choice of thresholds.

Section 3.6: Please define 'catchment state' - this seems to appear out of nowhere (unless I missed it earlier).

Answer: We will revise the manuscript to properly introduce the definition of catchment state.

P17L15: It's not clear to me where the topographic characteristics come from? Is this

simply the qualitative discussion in Section 2.1. Are there stream incision data for all the sites?

Answer: Indeed it is from the qualitative discussion since there is no detailed DEM. Unfortunately we did not measure stream incision but we will provide the average slopes over the measurement plots in the revised manuscript published by Demand et al. (2019).

Figure 10: Appears that the figure caption for the y axis is incorrect.

Answer: Thank you for pointing this out. The axis shows the "Normalised pre-event (groundwater) levels". We will correct this in the revised manuscript.

P19L1-3: I'm struggling with this logical leap between the results shown in Figure 7 and how they 'indicate that well-placed groundwater observation points can be representative of the given footslope, at least for pre-event conditions'. Given the close proximity of the within-site piezometers, there seems to be a surprising amount of scatter in these plots.

Answer: Despite the scatter, the similarity of the pattern (Figure 7) between piezometers at one site is relatively high compared to the similarity between sites, which leads to the assumption of a fingerprint that represents the functional link between a certain hillslope and the stream. Differences in patterns of piezometers at one site are due to local heterogeneities (e.g. hydraulic conductivities). In reverse this gives insight in how strong heterogeneities effect hillslope-stream responses.

P19L8-10: Or could it be that another portion of the catchment is connected, but not the hillslope with the piezometers?

Answer: This might be the case, yes. The catchment could contain "fast" hillslopes that manage to provoke a stream response at the stream level gauge before the adjacent (monitored) hillslope responds. We will mention this in the revised manuscript.

P21L28-31: How is it known that connectivity 'does not extend far up the slopes' when

those observations were not made? The substantial conclusions in this section are based on somewhat subjective placement of a threshold. It could even be argued that no clear threshold exists for some of the sites (see comment regarding Figure 9 above).

Answer: It is correct that we did not monitor upslope groundwater levels and this statement is an interpretation of the data. We reason that the observable increase in runoff coefficient (Figure 10) signals either more hillslopes being connected or that the connectivity of hillslopes leads further upslope, or both. In return, when groundwater tables are below this threshold, the spatial extent of subsurface connectivity is generally low. This could apply to the zones further upslope as well as other hillslopes within the catchment. We will clarify this in the revised manuscript.

P22L1-2: Again, I'm not clear on where the evidence is for this statement.

Answer: We apologize that we did not make this sufficiently clear and we will improve clarity in the revised manuscript. What we mean here is that looking at Figure 10, it would be sufficient to have the information of one of the piezometers per site to know if pre-event groundwater levels are above or below the threshold. If a rainfall event were to occur when groundwater levels are above the threshold the likelihood of high runoff coefficients is increased. To identify this state (above/below threshold) we do not need all of the piezometers currently installed at a certain hillslope – one would be enough and we could now potentially dismantle the other piezometers. For the one that we would keep monitoring we would pick one that on the one hand is consistent in its response pattern with the majority of the piezometers at this site and on the other hand has the clearest threshold signal among these. Please also see additional explanations to this end in previous answers.

References:

Demand, D.; Blume, T. & Weiler, M., 2019. Relevance and controls of preferential flow at the landscape scale. Hydrology and Earth System Sciences Discussions, 1-37 Gabrielli C.P., McDonnell J.J., 2020. Modifying the Jackson index to quantify the

re-lationship between geology, landscape structure and water transit time in steep wet headwaters. Hydrological Processes, early view. DOI: 10.1002/hyp.13700 Klaus J., Jackson C.R., 2018. Interflow is not binary: a continuous shallow perched layer does not imply continuous connectivity. Water Resources Research, 54, 5921-5932. DOI: 10.1029/2018WR022920 Freer, J., McDonnell, J. J., Beven, K. J., Peters, N. E., Burns, D. A., Hooper, R. P., Aulenbach, B., and Kendall, C.: The role of bedrock to­pography on subsurface storm flow, Water Resources Research, 38, 5–1–5–16, DOI: 10.1029/2001WR000872, 2002. Ocampo, C. J., Sivapalan, M., and Oldham, C.: Hy­drological connectivity of upland-riparian zones in agricultural catchments: Implications for runoff generation and nitrate transport, Journal of Hydrology, 331, 643 – 658, DOI: 10.1016/j.jhydrol.2006.06.010, 2006. Haught, D. R. W. and Meerveld, H. J.: Spatial variation in transient water table responses: differences between an upper and lower hillslope zone, Hydrological Processes, 25, 3866–3877, DOI: 10.1002/hyp.8354, 2011. van Meerveld, H. J., Seibert, J., and Peters, N. E.: Hillslope–riparian-stream connec­tivity and flow directions at the Panola Mountain Research Watershed, Hydrological Processes, 29, 3556–3574, DOI: 10.1002/hyp.10508, 2015.

---

## Author Comment (AC3) · 28 Apr 2020

Reviewer 3:

The authors studied the connectivity of hillslope to stream water using an impressive amount of data from 5 different catchments. The catchments studied were divided in two different geologies. The methodology they chose to use was focused in ground-water levels in piezometers near the streams and stream levels to try simplifying the hillslope approach that is often used when searching for connectivity with the stream. The study is interesting and I only have a few comments and questions to the authors adding to reviewers 1 and 2. I will now follow with some general comments and later

into more specific comments: - One of the goals of the study was to test if assessing the connectivity between hillslope and stream could be done with a single shallow near stream piezometer. I see your results show the answer is yes, but you miss to discuss or analyse why it worked in some piezometers and why not in others. What could you do different? In the discussions you mention that it works if you use a single well-chosen piezometer, how did you manage to have badly chosen piezometers in your network?

Answer: We thank the reviewer for taking the time to review our manuscript and are happy to see this positive résumé. Considering an un-investigated hillslope one cannot know in advance which location would lead to a 'well-chosen' piezometer and which one to a 'badly-chosen' piezometer. Nonetheless, the analysis showed that local heterogeneity did not influence each piezometer to a degree where no similarity at all could be observed. Therefore, a small number of piezometers (e.g. 3-4) should be enough to identify the characteristic patterns and which piezometers do represent the hillslope and which ones are disturbed due to local impacts. From this point on, one piezometer would be enough to describe the hillslope response and you can remove the other sensors. The well-chosen one would be one that on the one hand is consistent in its response pattern with the majority of the piezometers at this site and on the other hand has the clearest threshold signal among these. We will add an additional explanation to elaborate this in the revised manuscript.

- Related to the previous comment, is there any information about the soil profiles of the piezometers? Or were they installed blindly with the cobra? What do you mean with refusal? Is that refusal as when you reach rocks/bedrock? Or would refusal count as well as when you reach a clayey layer that could divide two aquifers (valid for the sandy soils)? Is there information on the elevation of the piezometers related to the stream/streambed?

Answer: For two of the five sites detailed information about soil profiles can be found in Sprenger et al. (2016) who investigated the areas of interest in-depth in terms of

soil profiles. Indeed the piezometers were drilled with the cobra and we have roughly described the profiles based on the cobra cores. This information will be added in the supplement. Refusal was either defined as bedrock (in schist) or when a very dense layer of clay soil was reached (marls), which could not be further penetrated by the cobra. Elevations of piezometers will be added to the revised manuscript.

Following come some specific comments:

- Page 1 Line 7: I suggest modifying the text here or earlier, "Step two" comes as a surprise since there was never a step one.

Answer: We agree and will revise the text according to your suggestion.

- P 4 L 3-12: This paragraph could be friendlier and provide more information if it was shown as well in a set of tables for the piezometers in each catchment. Stating elevation over stream bed, well depth and distance from the stream among other things. Maybe just as supplementary data, but it would help the reader visualize the piezometers better.

Answer: We agree and will add this information as supplementary data.

- P 4 L 15: I mentioned it before, but does this mean that there is no information on the soil profiles?

Answer: The cobra cores were described roughly during the drilling process. We will add this information in the supplementary data.

- P 5 Figure 2: I like this Figure and the information it provides. But I do agree with reviewer 2, either add data here or on the tables I mentioned two comments ago.

Answer: Information will be added in the revised version. For a better topographic overview at the sites we will add elevation information to Figure 2 in the revised version. We will also consider adding a table to the mentioned paragraph.

- P 7 Figure 3: I suggest you improve the horizontal lines that come from 'hfallThreshold' and 'hpostAmplitude' because they are hard to see in the current version.

Answer: Thank you for point this out. This will be adjusted in the revised version.

- P 8 L 9: "...are presumed to be rather short,". Is there any data on hydraulic conductivity?

Answer: Sprenger et al. (2016) obtained hydraulic conductivities for various sites within the catchment by inverse modelling. Average values for the two soil types spread from 293 – 675 cm per day (stagnosols) and 360 – 648 cm per day (cambisols). These information will be added in the revised version of the manuscript.

- P 8 L 34: I suggest changing All NA to allNA, as you used the term allNA in all other instances.

Answer: This will be adjusted in the revised version.

- P 10 L 9-10: I would suggest to rephrase this sentence and say directly what you did instead of saying first what you did not do. It would make it easier to read.

Answer: Thank you for pointing this out. This will be adjusted in the revised version.

- P 11 L 7: why did you consider it sufficiently representative? Could you provide more information to the reader? How far is it located? Similar elevation?

Answer: The location of Roodt station will be added to the map. It is located in the schist region at an elevation of about 400m. The marls sites are around 300m. Precipitation may vary to some extent over the entire catchment. However, there are indicators supporting the assumption for this analysis: Stream events without a previous precipitation event were removed. The number of removed stream events due to non-existing precipitation events was mostly below 10, indicating that rainfall events generally occurred across the entire catchment. As also mentioned in the answer to Reviewer 1, the analysis for the marls sites was rerun with a precipitation station <5km away from the monitoring sites and at the same elevation. No real changes in event detection

results could be observed, as well as no real changes in the runoff coefficient patterns. We therefore prefer to use the single reference rainfall station.

- P 12 L 2: Here you refer to Figure 9 before referring to Figure 8, maybe move Figure9 to position 8 or change the text.

Answer: Will be revised.

- P 12 L 3-4: "...low in summer and autumn." I don't see this generalization when I see Figure 7. There are several piezometers were autumn covers the whole spectrum. Or is there median values that we have no knowledge of?

Answer: We agree, the statement was not very precise. The wetting-up phase where groundwater levels start to rise on the seasonal time scale happens mostly in Autumn-Winter, which is why autumn events can also be found at higher groundwater levels. We will rewrite this statement.

- P 19 L 1-3: Any insights on why you had some exceptions? Because if those were your only wells this study would have completely different conclusions. If you have no insights, then that is valid as well, but it should be stated.

Answer: In case of M_D Piezo4 it is a road cut with a heavily disturbed soil (P 4 L 4). For the forested site (M_K Piezo3) it might be a strong influence of close biopores (tree roots) or an erratic sensor. We will add a discussion of these issues in the revised manuscript.

- P 22 Table 2: Use capital letters in each of the boxes, you used in some boxes but not all.

Answer: Thank you for pointing this out. We will make this consistent in the revised manuscript.

- P 22 L 1: "..single well chosen well..". I agree, but how did you choose well or bad? Assuming you had installed your piezometers.

Answer: Before having installed the piezometers one cannot tell which exact locations are most representative for the hillslope and which ones are influenced by very local heterogeneities. Only in hindsight is it possible to determine which piezometers at a site have a common characteristic response pattern. One of those would then be enough to monitor threshold behaviour. Choosing among them, we would choose the one with the clearest threshold signal.

- P 23 L 1: The closing ) is missing after Figure 10.

Answer: We will correct this.

References

Article (Sprenger_2016) Sprenger, M.; Seeger, S.; Blume, T. & Weiler, M. Travel times in the vadose zone: Variability in space and time Water Resources Research, Wiley-Blackwell, 2016, 52, 5727-5754

---

## Author Response (AR1)

Reviewer 1:

The page and line numbers refer to the track changes document.

General comment

This manuscript focuses on the characterization of hillslope-stream connectivity by using a novel joint event analysis of the response of stream and shallow groundwater levels. The authors examined the response timing of 18 groundwater sites located in five different footslopes in Luxembourg for 706 runoff events. The applied methodology included event detection, the quantification of response timing of groundwater compared to stream water level, the analysis of the relations between pre-event groundwater level with pre-event stream water level and runoff coefficient. The authors concluded that the joint analysis of groundwater and stream water levels provided information on the presence or absence, and on the degree of subsurface hillslope-stream connectivity. The found threshold relations between groundwater and stream water levels were interpreted as transmissivity feedback in the marls study sites, and fill-and-spill in the schist areas. The topic of this manuscript is of interest for the readers of the journal, and overall the paper is well written and structured. The presented analysis for such a large time series of groundwater levels is quite rare, and therefore is particularly important to advance our comprehension of hillslope-stream subsurface connectivity. Nonetheless, I have some specific questions/comments for the authors, and I would like to see integrated in the manuscript some more methodological details.

We thank the reviewer for taking the time to review our manuscript and are happy to see this positive assessment.

Specific comments

1.  I suggest to the authors to clearly provide in the introduction the definition of subsurface hydrologic connectivity, that they considered (currently such a definition can only be guessed by the readers).

    We added the following definition to the introduction (page 2 line 7-11):
    "Ali and Roy (2009) collected various definitions of hydrologic connectivity used in previous studies, which differ in spatial scale (hillslope vs watershed) and observed features (e.g. water cycle or landscape). The most appropriate definition in the context of our investigation is the following: 'The condition by which disparate regions on a hillslope are linked via lateral subsurface water flow (Hornberger et al. 1994; Creed and Band 1998)'"

2.  The authors mentioned in the abstract that they performed their joint analysis for rainfall-runoff events, but throughout the manuscript there is no description of the rainfall characteristics (e.g., total rainfall, intensities and duration of the selected events) and where they were monitored (are the weather stations located in the study catchments?). I suggest to report such details in the text. Furthermore, I would like to see a table presenting the main summary statistics for rainfall, runoff and groundwater characteristics of the considered events.

We added the location of the two precipitation stations (Roodt and Useldange) to the map in Figure 1 (page 4). The text was modified (page 7, line 7-12). Additionally, summary statistics for precipitation, runoff and stream response can be found in Table 2 (page 16).

3. Since the analysis was carried out for the whole time series (winters and early spring included), I am wondering whether there were snowfalls, and if the authors considered snowmelt-induced runoff events and rain-on-snow events in the analysis. If such events were discarded, I suggest to integrate the description of the methodological approach for event detection. Otherwise, the authors should clearly state that they focused only on rainfall-runoff events.

On page 13, line 22-28 we added the following lines:
"As the analysis covers also winter and early spring events, the effect of snow fall and snow melt on the event detection was assessed and found to be unlikely to impact our analysis: Snow fall events are generally quite rare in Luxembourg, so the number of events affected is assumed to be low. A rain on snow event would be captured by its runoff response, but the in this case erroneous estimate of rainfall input would only impact the analysis of event runoff coefficients as our analyses mainly focus on the relationship between streamflow and groundwater responses. Pure snow melt events are not included in the analysis, as in this case there is no directly preceding precipitation event and thus this necessary event identification criterion is not met."

4. In Table 1 (or in a new table), I suggest to provide the topographic characteristics of the groundwater sites together with their depth. These details could help to understand whether the topography is very similar (or very different) among the monitored locations, and to support the discussion at page 17, lines 15-19. Moreover, what is the extension of the riparian zone compared to the hillslopes in the study sites?

Thank you for pointing this out. We decided that the best way to visualise the topography was in a figure, rather than a table. Therefore, new added a new figure (Figure 3, page 6) illustrating the distance to stream together with ground level and piezometer depth relative to the stream bed. Table A2 on page 32 contains the corresponding values.

5. Have the authors considered their analysis of subsurface connectivity in light of recent findings by Klaus and Jackson (2018) and Gabrielli and McDonnell (2020)? Are there bedrock permeability data for the selected study sites?

We added the following (page 28, line 12):
"Studies focusing on the Downslope Travel Distances (Klaus and Jackson, 2018; Gabrielli and McDonnell, 2020) found that only lower regions of a hillslope contribute to the streamflow via interflow, while in upper regions water percolates into the deeper groundwater. In our study, however, we find that there is a threshold in the near-stream groundwater levels above which event runoff

coefficients rise strongly to values above 50%, indicating that it is not just the near stream footslope contributing to event runoff."

6.  In the section "2.4 Event detection" and Fig. 4, it is not clear which response timings were considered for complex events with multiple peaks (both in stream and ground-water level). Furthermore, which peak in stream water level is considered if there is only one peak for the groundwater level?

    We edited the text as follows (page 9, line 3-5):
    "If two or more events overlap they are merged into one single longer event (Figure 5) and the highest peak is determined as the event maximum. From there on it is handled as a simple event according to Figure 4."

7.  Page 11, line 5: Please provide a reference for the method used for the stormflow calculation.

    A reference was added (page 13, line 5-6):
    "The approach to separate baseflow from discharge was developed in the style of the constant slope method (Dingman, 2002)."

8.  Page 20, line 5: "No pronounced differences...": could the authors report the results of the applied statistical test?

    As we cannot do statistical testing in this case, we changed the wording to clarify this. We now say "No clear visual differences in timing can be observed between marls and schist." (page 24, line 25)

9.  Page 21, lines 3-4: Please provide more details about the investigated relations between rainfall characteristics and event responses.

    Ranked cross correlation coefficients between rainfall and response characteristics were added in table A1 (page 31).

10. Page 22, lines 8-10: Please remove these details from the available literature, and report them in a table. Please consider that other recent studies examined almost or more than 100 events (e.g., Rinderer et al., 2016; Zuecco et al., 2019).

    We removed the number of events from the text and created a new table (Table 5, page 29). Unfortunately, we did no have access to Zuecco et al. (2019).

11. Page 23, line 13-16: The example of considering just two events in the data analysis is a very extreme case, and so far, I have never seen it. Therefore, please revise the sentence. The main question is how many events and piezometers do we need to capture the temporal and spatial variability of subsurface connectivity?

    Thank you for stressing this point. In an effort to be more realistic we deleted "(e.g. one for winter and one for summer)" as well as "If these two events represented the two extremes (high and low pre-event water levels) one would need to assume a

functional relation for all the potential events in-between (which is likely to be flawed)" and instead added the following lines (page 29, line 6-10):
"Nonetheless, the question is not so much about how many events are necessary (in absolute numbers) as more about the necessary time period to cover the temporal variability generated by different hydrological processes. It is therefore necessary to accumulate a large number of events across all seasons. In terms of extreme events (droughts/floods) the covered time period and number of events will need to be even higher, on the one hand to capture these events, and on the other hand to put them into context."

Technical corrections
1. Page 2, line 25: "assess". (page 2, line 31)
2. Page 3, line 1: "hillslope" instead of "slope". (page 3, line 8)
3. Page 4, line 2: "July". (page 5, line 5)
4. Page 23, line 4: "these" instead of "this". (page 28, line 5)
5. Figure 10: Based on the caption, the label of the y axis should be "Normalised pre-event groundwater level". (page 21, Figure 11)

Technical corrections can be found at locations in parenthesis.

Reviewer 2:

The page and line numbers refer to the track changes document.

The manuscript by Beiter et al details a study aimed at understanding hillslope-stream connectivity. They collected 5-6 years of paired near-stream groundwater and streamwater levels at five locations within an agricultural catchment in western Luxembourg. At each site, shallow groundwater levels were logged at 5 minute intervals at 3 to 4 piezometers located within 15 m of the stream level site. They extracted about 150 individual rainfall-runoff events from the data record using an approach that interrogates the stream water level time series. For each event identified, they also extracted groundwater response metrics from the corresponding piezometers. They compared stream and groundwater responses to quantify temporal changes in hillslope-stream connectivity. They found a threshold-type response in stream water level linked to antecedent groundwater levels. Low antecedent groundwater levels were associated with variable stream water level responses. In contrast, high antecedent groundwater levels were associated with more consistent stream water level responses. They speculate that the hydrologic processes controlling these patterns were transmissivity feedback at the marls sites and fill and spill at the schist sites. The topic covered in this manuscript is appropriate for HESS. The study contains an impressive data set and some of the visualizations do a great job of showing these data (e.g., Figure 8 and 9). Overall, the writing is not bad, but some of the grammar is confusing which makes it difficult to understand some of the elements of the paper.
Given the amount of data, I'm left feeling a little underwhelmed by the key conclusions. This might reflect the vagueness of the key research questions (page 3, lines 12-15).For example, 'provide information' is a very general statement - try to be more specific about what is learned from this sort of joint analyses. I would encourage the authors to formulate testable hypotheses to help add more structure to the manuscript. This would also help clarify the key findings of this study.

We thank the reviewer for taking the time to review our manuscript and are happy to see that Figures 8 and 9 (now Figures 9 and 10) are appreciated. We sharpened the research questions by including the following hypotheses into the introduction and discussion (page 3, line 19-24; page 29, line 13-page 30, line 18):

"Hypothesis 1: hillslopes remain disconnected from the stream for most of the time and connect only during short periods of time.

Hypothesis 2: the two geologies schist and marls differ in topography and soil characteristics. As a result, their hillslope-stream systems will show differing connectivity patterns.

Hypothesis 3: monitoring at the footslope can provide information on hillslope-stream connectivity at this location but also at the catchment scale."

Introduction: As reviewer #1 highlights, some of the more recent research on this topic should be discussed here or in the discussion (e.g., Klaus and Jackson 2019 WRR,Gabrielli and McDonnell 2020 HP).

We added the following (page 28, line 12):
"Studies focusing on the Downslope Travel Distances (Klaus and Jackson, 2018; Gabrielli and McDonnell, 2020) found that only lower regions of a hillslope contribute to the streamflow via interflow, while in upper regions water percolates into the deeper groundwater. In our study, however, we find that there is a threshold in the near-stream groundwater levels above which event runoff coefficients rise strongly to values above 50%, indicating that it is not just the near stream footslope contributing to event runoff."

P3L1-3: An example of a run-on sentence that should be avoided.

The sentence was changed to (page 3, line 8-10):
"Monitoring shallow groundwater tables in the riparian zone over longer periods of time will allow us to capture a large number of events. We hypothesize that the analysis of these events will provide not full, but representative information on hillslope-stream connectivity."

P3L8: What is meant by a 'rough interpretation'?

The sentence was changed to (page 3, line 14-16):
"Analysing the relationship between responses in near-stream shallow groundwater and stream thus permits us to determine the dominant processes."

Section 2.1: Provide some information about soils and the vegetation cover. Could the sites be given more descriptive names? I realize the 'S' and 'M' represent the dominant geology, but what do 'J', 'V', 'D', and 'K' represent?

Despite the aggregated information about the various sites (soil, land use, etc.) in Table 1 another Table (A3) was added containing the soil horizons. The letters refer to a larger scale reference system of 45 sensor clusters distributed in the Attert Catchment and do not have a meaning per se. A sentence to this effect was added on page 5. A full list of the sites can be found in Appendix A of Demand et al. 2019.

Figure 2: Could elevation be added to these plots? Or at least the elevation of the ground surface at the piezometer and the depth of the piezometer relative to the streambed? It might be really helpful to include photographs of the 5 site installations so that the readers can get a better sense of the sites.

Thank you for pointing this out. We decided that the best way to visualise the topography was in a figure, rather than a table. Therefore, new added a new figure (Figure 3, page 6) illustrating the distance to stream together with ground level and piezometer depth relative to the stream bed. Table A2 on page 32 contains the corresponding values. Unfortunately, the photographs do not help in getting a better image of the sites.

Table 1: What are the slope quartiles referring to? The hillslope or catchment?

A footnote was added to Table 1 (page 6) stating that the slope quartiles refer to the subcatchments.

P6L4: Where is the Roodt station? Any concerns about spatial variability in precipitation inputs? I know it is mentioned that precipitation is assumed to be uniform across the catchments for the runoff ratios; however, it seems like not all stream water level sites respond to precipitation events. This may suggest that the uniform precipitation assumption is not reasonable.

We reran the analysis with Roodt station only for the schist sites and Useldange station only for the marls sites. The results are almost entirely the same. There is a very small shift of 1-2 events per site which are now (not) detected). Also, the runoff coefficient patterns remain the same. However, in the revised manuscript we are now using the two precipitation stations, one per geology to avoid the assumption of uniform distribution within the catchment.
The text was edited accordingly (Page 7, line 6-11):
"Hourly precipitation data from the Roodt and Useldange weather stations were obtained from AgriMeteo Luxembourg. Both stations are located within the Attert catchment, the Roodt station close to schist and the Useldange station being close to marls sites. Discharge data with 15 min temporal resolution were provided from the Luxembourg Institute for Science and Technology (LIST) for the Weierbach station (for schist) and the Wollefsbach station (for marls). Figure 1 shows precipitation stations (upper left plot) and discharge stations (upper right plot)."

P6L21-22: Looks like this percentage was tested? How sensitive are the results to different values?

No formal test was applied. Sensitivity regarding the timing was higher when the percentages were lower due to the extended onset and offset. Higher thresholds were in general less sensitive and even more for the onset compared to the offset.

Section 2.4: Why not conduct the event detection by using the precipitation record (as is frequently done) instead of the stream level records?

The general idea was to design a stream-centred approach in order to focus on the response and its relation to the hillslope. Starting with the precipitation events would have also required us to formulate a definition for whether or not a stream event was observed. From our point of view, using event detection on the precipitation time series would not have helped investigating the interaction between hillslope and stream responses.

P8L1: Were there times when the piezometers showed a response but not the stream?

This was not investigated, due to the way the analysis was set up: we first identified the stream flow events and then used these events to check for groundwater responses. This is explained on page 9, line 8.

P8L15: This would only happen in autumn?

The wetting-up phase is generally in autumn since groundwater levels are low in summer and high in winter. In particular during this period one can observe stream/piezometer events where the post-event water level lies above the pre-event water level. However, on

shorter time scales wetting-up can also occur in other seasons. We added this information on page 10, line 4.

P8L18-22: For the search interval, was this a moving window or fixed interval search?

The following line was edited to clarify this (algorithm 1):
"DEFINE: two fixed search intervals from peak along the rising and the falling limb (tpreSearchInterval = 24h, tpostSearchInterval = 48h)"

P10L5-14: Consider re-writing this to improve clarity.

Thank you for pointing this out. The part in question was re-written (page 12, line 2-5):
"In hillslope-stream systems infiltration and runoff generation processes are highly dynamic during events on a time scale of hours and days. To gain additional insight into what happens during these periods we chose to handle the water level changes and timing as two separate aspects. This provides us with a view of the temporal behaviour on the one hand and changes in the state variables (water levels) of the hydrologic system on the other."

P10L13: What is meant by 'hints'?

The sentence was rephrased for clarification (page 10, line 6):
"Relative timing and lags between groundwater and stream responses extracted from a large number of events hint at causal relationships."

P10L26: What is meant by 'a more or less deterministic increase'?

The sentence in question was rephrased (page 12, line 18-21):
We would expect that a given increase in groundwater level at a given depth would result in a more or less predetermined/deterministic increase of stream water level (assuming the groundwater fluctuations are representative of the catchment). This means that if Events A and B have similar initial conditions and cause similar groundwater level rises we would expect the stream water level rise of Event A to be the same as for Event B.

P11L26: For what purpose is this response considered negligible?

This was made clearer by shortening the mentioned sentence (page 14, line 2-3) and adding another one to the discussion (page 23, line 5-7)

"The noLocalMaximum response is very rare with only 11 occurrences."

"The very low number of only 11 noLocalMaximum groundwater events support the suitability of the developed event detection. A high number would indicate that the chosen approach is not performing properly and would need to be questioned entirely."

Figure 5 (and others): The 'Event type' colour scale is very difficult to interpret for a colour-blind person. Consider using some other way to visualize these data (shapes maybe, although that might be difficult to see as well)?

We put an effort into changing the representation of 'event type'. However, it makes the whole figure very confusing and not readable at all. We apologise for the inconvenience.

Figure 7: Perhaps distinguish the Seasons by shape instead of colour.

Similar to the situation in the previous comment, we tried to change the representation of season but could not come up with a solution that produced a clearer output.

Figure 8: Very nice graph!
Thank you!

Figure 9: Could the approach used to set those thresholds be discussed a bit more? I realize they were done visually, but there are some sites/piezometers that I would argue don't have a clear threshold (most of S_V, S_J piezo1, most of M_K, etc.).

We apologise for the inaccuracy in the description. We added the following explanation to the manuscript (page 7, line17 and following):
"The way the patterns changed at the threshold was not identical for all sites. While many piezometers showed an abrupt change in slope (M_D Piezo1-3, M_J Piezo1 and S_J Piezo 2-4) others showed a converging of their envelope functions (encompassing the bundle of slope lines) (S_J Piezo1, S_V Piezo3 and Piezo4). For some piezometers the change in pattern was a sudden clustering of lines (M_K Piezo1-2, S_V Piezo2). All these observed changes in patterns signal that hydrologic processes change when the threshold values are passed."

Section 3.6: Please define 'catchment state' - this seems to appear out of nowhere (unless I missed it earlier).

We agree and added a bit more clarification (page 19, line 23-26)

"We assume that the threshold (Figure 10) marks a change in catchment state, where conditions above the threshold have the potential for high connectivity while conditions below the threshold indicate lower connectivity. To investigate if the shift in state is synchronous across the sites we plotted the event time series colour-coded by system state (above/below the threshold) (Figure 12)."

P17L15: It's not clear to me where the topographic characteristics come from? Is this simply the qualitative discussion in Section 2.1. Are there stream incision data for all the sites?

Indeed it is from the qualitative discussion since there is no detailed DEM. Unfortunately, we did not measure stream incision but are now providing more topographic information in Figure 3 (page 6).

Figure 10: Appears that the figure caption for the y axis is incorrect.

We corrected the figure caption of Figure 11 (page 21, former Figure 10).

P19L1-3: I'm struggling with this logical leap between the results shown in Figure 7 and how they 'indicate that well-placed groundwater observation points can be representative of the

given footslope, at least for pre-event conditions'. Given the close proximity of the within-site piezometers, there seems to be a surprising amount of scatter in these plots.

Despite the scatter, the similarity of the pattern (Figure 7) between piezometers at one site is relatively high compared to the similarity between sites, which leads to the assumption of a fingerprint that represents the functional link between a certain hillslope and the stream. We have rephrased this section to clarify our interpretation.

P19L8-10: Or could it be that another portion of the catchment is connected, but not the hillslope with the piezometers?

A sentence was added (page 24, line 16-18):
"In a highly heterogeneous catchment, certain 'fast' hillslopes with very high hillslope-stream connectivity and high outflows might provoke a stream-response at the stream level gauge before the monitored hillslope responds. In this case the interpretation of low subsurface-connectivity would only hold for the monitored hillslope."

P21L28-31: How is it known that connectivity 'does not extend far up the slopes' when those observations were not made? The substantial conclusions in this section are based on somewhat subjective placement of a threshold. It could even be argued that no clear threshold exists for some of the sites (see comment regarding Figure 9 above).

A clarification on how the thresholds were obtained can be found in the answer to the comment regarding Figure 9 (now Figure 10). While the threshold was only determined visually and could therefore be called subjective, it is supported by the fact that runoff coefficients for events above this threshold are more likely to be high than for events below it. This is true for nearly all piezometers.
The following lines were modified for clarification (page 26, line 22-27):
"Events with catchment states above the threshold are likely to have higher event runoff coefficients (Figure 11) and are thus assumed to generate substantial lateral subsurface stormflow caused by high hillslope-stream connectivity (more connected hillslopes or connectivity extending further upslope, or both. Catchment states below the threshold generate only minor lateral flow. In this case the spatial extent of hillslope-stream connectivity is generally low (few connected hillslopes or connectivity does not extend far up the slopes)."

P22L1-2: Again, I'm not clear on where the evidence is for this statement.

We apologize that we did not make this sufficiently clear and have improved clarity in the revised manuscript. We added the following lines for clarification (page 25, line 19-25):
"However, our study results suggest that it would be sufficient to have the information of one of the piezometers per site to know if pre-event groundwater levels are above or below the threshold. If a rainfall event were to occur when groundwater levels are above the threshold the likelihood of high runoff coefficients is increased. To identify this state (above/below threshold) we do not need all of the piezometers currently installed at a certain hillslope – one would be enough and we could now potentially dismantle the other piezometers. Considering an un-investigated hillslope, one cannot know in advance which location would lead to a 'well-chosen' piezometer and which one to a 'badly-chosen'

piezometer. Nonetheless, the analysis showed that local heterogeneity did not influence the piezometers to a degree where no similarity at all could be observed. Therefore, a small number of piezometers (e.g. 3-4) should be enough to identify the characteristic patterns and which piezometers do represent the hillslope and which ones are less suited due to local anomalies. From this point on, one piezometer would be enough to describe the hillslope response and you can remove the other sensors. The well-chosen one would be one that on the one hand is consistent in its response pattern with the majority of the piezometers at this site and on the other hand has the clearest threshold signal among these."

Please also see additional explanations to this end in previous answers.

Reviewer 3:

The authors studied the connectivity of hillslope to stream water using an impressive amount of data from 5 different catchments. The catchments studied were divided in two different geologies. The methodology they chose to use was focused in ground-water levels in piezometers near the streams and stream levels to try simplifying the hillslope approach that is often used when searching for connectivity with the stream. The study is interesting and I only have a few comments and questions to the authors adding to reviewers 1 and 2. I will now follow with some general comments and later into more specific comments:
- One of the goals of the study was to test if assessing the connectivity between hillslope and stream could be done with a single shallow near stream piezometer. I see your results show the answer is yes, but you miss to discuss or analyse why it worked in some piezometers and why not in others. What could you do different? In the discussions you mention that it works if you use a single well-chosen piezometer, how did you manage to have badly chosen piezometers in your network?

We thank the reviewer for taking the time to review our manuscript and are happy to see this positive résumé.

This additional explanation was added in section 4.6 in the revised manuscript (page 27, line 5-12):
Considering an un-investigated hillslope, one cannot know in advance which location would lead to a 'well-chosen' piezometer and which one to a 'badly-chosen' piezometer. Nonetheless, the analysis showed that local heterogeneity did not influence the piezometers to a degree where no similarity at all could be observed. Therefore, a small number of piezometers (e.g. 3-4) should be enough to identify the characteristic patterns and which piezometers do represent the hillslope and which ones are disturbed due to local anomalies. From this point on, one piezometer would be enough to describe the hillslope response and you can remove the other sensors. The well-chosen one would be one that on the one hand is consistent in its response pattern with the majority of the piezometers at this site and on the other hand has the clearest threshold signal among these.

Related to the previous comment, is there any information about the soil profiles of the piezometers? Or were they installed blindly with the cobra? What do you mean with refusal? Is that refusal as when you reach rocks/bedrock? Or would refusal count as well as when you reach a clayey layer that could divide two aquifers (valid for the sandy soils)? Is there information on the elevation of the piezometers related to the stream/streambed?

Indeed, the piezometers were drilled with the cobra and we have roughly described the profiles based on the cobra cores (see Table A3, page 33) and elevations of piezometers are now shown in Figure 3 of the revised manuscript.
The following sentence was added to clarify the term 'refusal' (page 6, line 13-14):
"Refusal was either defined as bedrock (in schist) or when a very dense layer of clay soil was reached (marls), which could not be further penetrated by the cobra."
For two of the five sites detailed information about soil profiles can be found in Sprenger et al. (2016) who investigated the areas of interest in-depth in terms of soil profiles.

Following come some specific comments:

- Page 1 Line 7: I suggest modifying the text here or earlier, "Step two" comes as a surprise since there was never a step one.

Thank you for pointing that out. We changed the text to the following (page 1, line 5-8): "As a first step a new data analysis scheme was developed, separating the aspects of a) response timing and b) extent of water level change. This provides new perspectives on the relationship between groundwater and stream responses. In a second step we investigated if this analysis can give an indication of hillslope-stream connectivity at the catchment scale."

- P 4 L 3-12: This paragraph could be friendlier and provide more information if it was shown as well in a set of tables for the piezometers in each catchment. Stating elevation over stream bed, well depth and distance from the stream among other things. Maybe just as supplementary data, but it would help the reader visualize the piezometers better.

We decided that the best way to visualise the topography was in a figure, rather than a table. Therefore, new added a new figure (Figure 3, page 6) illustrating the distance to stream together with ground level and piezometer depth relative to the stream bed. Table A2 on page 32 contains the corresponding values.

- P 4 L 15: I mentioned it before, but does this mean that there is no information on the soil profiles?

The cobra cores were described roughly during the drilling process. This information was added to table A3 on page 33.

- P 5 Figure 2: I like this Figure and the information it provides. But I do agree with reviewer 2, either add data here or on the tables I mentioned two comments ago.

We added a new figure (Figure 3, see response above).

- P 7 Figure 3: I suggest you improve the horizontal lines that come from 'hfallThreshold' and 'hpostAmplitude' because they are hard to see in the current version.

The lines in Figure 4 (former Figure 3) were slightly modified to improve visibility.

- P 8 L 9: "...are presumed to be rather short,". Is there any data on hydraulic conductivity?

A reference for the hydraulic conductivities was added (page 6, line 8-9): "Average hydraulic conductivities for the two soil types range from 293 to 675 cm/day (stagnosols) and from 360 to 648 cm/day (cambisols) (Sprenger et al. 2016)."

- P 8 L 34: I suggest changing All NA to allNA, as you used the term allNA in all other instances.

All NA was changed to allNA (page 11, line 7).

- P 10 L 9-10: I would suggest to rephrase this sentence and say directly what you did instead of saying first what you did not do. It would make it easier to read.

Thank you for the suggestion. The sentence was revised accordingly (page 12, line 3-4): "To gain additional insight into what happens during these periods we chose to handle the water level changes and timing as two separate aspects."

- P 11 L 7: why did you consider it sufficiently representative? Could you provide more information to the reader? How far is it located? Similar elevation?

We reran the analysis with Roodt station only for the schist sites and Useldange station only for the marls sites. The results are almost entirely the same. There is a very small shift of 1-2 events per site which are now (not) detected. Also, the runoff coefficients patterns remain the same. Nevertheless, we maintained the use of one closely located precipitation station per geology to avoid the assumption of uniform distribution within the catchment.
The text was edited accordingly (Page 7, line 6-11):

"Hourly precipitation data from the Roodt and Useldange weather stations were obtained from AgriMeteo Luxembourg. Both stations are located within the Attert catchment, the Roodt station close to schist and the Useldange station being close to marls sites (Figure 1, upper left). Discharge data with 15 min temporal resolution were provided from the Luxembourg Institute for Science and Technology (LIST) for the Weierbach station (for schist) and the Wollefsbach station (for marls) (Figure 1, upper right)."

- P 12 L 2: Here you refer to Figure 9 before referring to Figure 8, maybe move Figure9 to position 8 or change the text.

The early mention of Figure 10 (former Figure 9) was removed.

- P 12 L 3-4: "...low in summer and autumn." I don't see this generalization when I see Figure 7. There are several piezometers were autumn covers the whole spectrum. Or is there median values that we have no knowledge of?

We apologize for not being precise enough. We added the following sentence to specify the behaviour in autumn (page 15, line 1-2):
"In Autumn the wetting-up phase begins which produces events over a wider range of groundwater levels. Summer events can be found mostly at the lower end with occasional events at higher groundwater levels."

- P 19 L 1-3: Any insights on why you had some exceptions? Because if those were your only wells this study would have completely different conclusions. If you have no insights, then that is valid as well, but it should be stated.

In case of M_D Piezo4 it is a road cut with a heavily disturbed soil (page 5, line 10). For the forested site (M_K Piezo3) there is no obvious explanation. We added this information also on page 24, line 5-6.

- P 22 Table 2: Use capital letters in each of the boxes, you used in some boxes but not all.

Corrections were made in the revised manuscript.

- P 22 L 1: "..single well chosen well..". I agree, but how did you choose well or bad? Assuming you had installed your piezometers.

We thank you for making this point. We added an explanation to how a well-chosen piezometer would be identified (page 25, line 19-25):
"However, our study results suggest that it would be sufficient to have the information of one of the piezometers per site to know if pre-event groundwater levels are above or below the threshold. If a rainfall event were to occur when groundwater levels are above the threshold the likelihood of high runoff coefficients is increased. To identify this state (above/below threshold) we do not need all of the piezometers currently installed at a certain hillslope – one would be enough and we could now potentially dismantle the other piezometers. Considering an un-investigated hillslope, one cannot know in advance which location would lead to a 'well-chosen' piezometer and which one to a 'badly-chosen' piezometer. Nonetheless, the analysis showed that local heterogeneity did not influence the piezometers to a degree where no similarity at all could be observed. Therefore, a small number of piezometers (e.g. 3-4) should be enough to identify the characteristic patterns and which piezometers do represent the hillslope and which ones are less suited due to local anomalies. From this point on, one piezometer would be enough to describe the hillslope response and you can remove the other sensors. The well-chosen one would be one that on the one hand is consistent in its response pattern with the majority of the piezometers at this site and on the other hand has the clearest threshold signal among these."

- P 23 L 1: The closing ) is missing after Figure 10.

This was corrected

References

[revised manuscript text omitted]

---

## Author Response (AR2)

Review #1:

I thank the authors for considering the comments of the reviewers and improving the manuscript accordingly. Overall, I think the revised manuscript is almost ready for acceptance after a revision of the following minor comments. Pages and lines refer to the revised manuscript with no tracked changes.

We very much appreciate your evaluation and edited the manuscript according to the comments below.

- Page 2, lines 7-8: The sentence can be shortened as follows: "In this study, we considered hydrologic connectivity as "The condition by which disparate regions on a hillslope…"".

The suggestion was adopted (page 2, line 7)

- Page 3, line 18: I suggest to reformulate as follows: "The selected study sites differ in geologies (schist and marls), topography and soil characteristics.".

The suggestion was adopted (page 3, line 21)

- Figure 3: Please mention in the caption the meaning for the different length of the stream bar in each site.

A sentence for the stream sensors was added. (page 6, Figure 3)

- Page 12, line 24: I suggest to change to: "to unlikely impact our analysis".

The suggestion was adopted (page 13, line 12)

- Page 12, line 28: "there is no directly preceding precipitation event" is a bit unclear. Please consider rephrasing.

The sentence in question was rephrased as follows (see page 13, line 15-16): "Pure snow melt events without a preceding precipitation event are not included in the analysis as precipitation is a necessary identification criterion."

- Table 2: I suggest to explain in the caption (and in the text at the end of section 3.1) why there are events with minimum runoff of 0.0 mm. How is that possible?

The event runoff excludes the baseflow (see formula 3) which is why values of 0.0mm are possible. Explanatory sentences were added in at the end of section 3.1 (site 13, page 23-24) as well as in the caption of Table 2 (page 16).

- Page 22, lines 10-11 and page 23 lines 1-2: The sentence is a bit long. I suggest to break it in two sentences.

The sentence in question was split into two and rephrased as following (page 22, line 11 – page 23, line 1-3): "Visually comparing the point cloud patterns of the piezometers at each single site (Figure 8) reveals, despite the scatter, site-internal similarity (a site-specific fingerprint) among the piezometers. Two exceptions to this observation are the previously mentioned (see section 3.1) piezometer M_D_Piezo4, which is located in disturbed soil on a steep slope below a road, and M_K_Piezo3, where the anomalous behaviour cannot be explained at first sight."

- Page 28, line 5: I suggest to change to: "Despite the high spatial variability, we were unable to detect patterns that could be explained only by the geologies of the areas".

A comment by Reviewer #2 required rewriting of the paragraph containing the sentence in question. Thus, this sentence disappeared entirely.

Reviewer #2:

In this manuscript, the authors analyze a multi-year time series of shallow groundwater levels in 15 wells and stage at 5 stream locations in two catchments. They identified events throughout the study period and compared groundwater responses to stream response in several ways to determine how often hillslopes connected to the streams, whether or not connectivity depended on underlying geology, and if monitoring wells at a footslope can be informative of catchment connectivity. In their conclusions they describe how the mechanisms by which footslopes connect to the stream are likely different between the two geologies in the study (fill and spill vs.transmissivity feedback), while the frequency of connection didn't vary significantly. Furthermore they conclude that a single well in a footslope in a catchment can describe catchment connectivity, and that many years of data are required to adequately describe the processes they investigated.

The data analysis and presentation of the results in the paper were very well done. I think this study is appropriate for HESS and will be of interest to its readers. However, I think some of the conclusions are overstated in their present form (detailed below). I think the reasoning behind a couple of the conclusions must be more clearly stated, or they should probably be taken out. I think there is still plenty of interest even if a couple of the conclusions were either hedged more clearly or taken out.

We thank you for taking the time to review this manuscript and are happy to see that you appreciated our work, in particular the aspects of data analysis and presentation. We also appreciate your suggestions to improve the manuscript further and will respond to them comment by comment below.

The first finding that I think needs work is that this study illustrates that you need many years of data to make the conclusions made in the study. I certainly don't disagree! I think this is probably true. However, I don't see in the paper where this is demonstrated quantitatively. I think the authors would need to show how the conclusions would differ if you showed only one year, compared to the entire time series. This might be accomplished by a version of figure 8, 9, or 10 that highlights one year of the data and how that differs from the total time period of the study. However, I wonder if the authors need to make this conclusion. I think there are plenty of other conclusions in the paper without having to make this one, which might necessitate more figures and analysis in an already figure-heavy paper.

We appreciate the point being made here. Since we consider the coverage of multiple years as very important, we suggest adding a few figures (see below) to the appendix to exemplarily highlight what difference it would make if we had monitored one particular year (2015 in this case). Instead of going into detail for each variable, site and piezometer, this serves to visualise what information is missing, leading to misinterpretations such as linearity where there is none,

very few high or low runoff coefficients where it is more balanced, uniform relation between stream and hillslope where there is much more variability, missing out a threshold where there is one, etc. In these figures the events of 2015 are shown in black and all others in grey. We refer to these figures on page 29, line 5 to strengthen the argument as you suggest.

[Figure]

[Figure]

[Figure]

[Figure]

The second finding that I struggled with is the notion that one piezometer can indicate the connectivity of the entire catchment. I agree with the authors that there are wells in their analysis that are tightly related to streamflow response, but I think the conclusions that result should maybe be a bit more nuanced. To me, the wells that are related to the stream response are likely somehow indicative of the storage state of the catchment. That could be because they are in a position where they are representing hillslope connectivity, as posited in the manuscript, but it could also be explained by other processes. How do we know the well isn't just in a spot that is in the hyporheic zone and that is why it is so well linked to stream behavior? Or the well might response might be governed by deeper flowpaths that are more indicative of overall catchment wetness than the hillslopes, and that is why the relationship is so good? I think it would be good to mention other possibilities, and also to be careful saying that these wells represent hillslope connectivity. The spatial extent of the wells in the study was not that large, and the wells didn't go very far up the hillslopes, so I don't think the authors can say whether or not they were indicative of what was going on up the hillslope. All that being said, I think the authors' analysis shows that some near stream wells CAN be good indicators of catchment wetness and be predictive of how the stream will respond to precipitation, which is definitely an interesting finding!

We think that this is a misunderstanding. Yes, as you say, the wells that have a high predictive power for the response of the catchment (i.e. the event runoff coefficient) are probably indicative of the storage state of the catchment. But for high event runoff coefficients to occur the storage levels need to be at a level where connectivity is easily established by the added event rainfall input. We did not want to state that the well actually "sees" catchment or entire hillslope connectivity (this connectivity is instead indicated by the high event runoff coefficient), but that there is a relationship with the event runoff coefficients that allows us to use the piezometer to predict potential connectivity. We have added two sentences in the conclusions to hypothesis 3 to clarify this. Whether this increased connectivity and runoff production is due to more hillslopes being connected, or to a few hillslopes connecting further uphill, or a mixture of both, cannot be answered. If a piezometer would be located in the hyporheic zone, piezometer responses would be strongly correlated to those of the stream.

Abstract:
Lines 13-17: Is this finding detailed in the conclusions?

Thank you for pointing this out. We included the findings about relative timing into the conclusion for hypothesis 1 (page 29, line 10-11).

Line 23: I think the authors should be careful saying they provided insight into catchment scale connectivity with such a small spatial distribution of wells.

We agree with the reviewer about overstating the link to the entire catchment. We specified this statement as follows (see page 1, line 23-24):
"We also find that locally measured thresholds in groundwater levels can provide insight into the connectivity and event response of the corresponding headwater catchments."

Main text
Should the intro be past tense?

We changed the part of the introduction describing our study to past tense (page 3, line 6-18).

Page 3 line 11: can spatial connectivity be addressed with the study design?

Thank you for pointing that out. It is true that by monitoring the footslope no exact statements can be made about the spatial distribution along that hillslope. Thus we changed the phrase to focus on the spatial distribution between the sites (page 3, line 13-14):
"While this can be very informative, we suggest that our pragmatic approach focusing only on the footslope and a joint analysis of shallow groundwater and streamflow response to rainfall events will still allow us to develop a general picture of when connectivity is established, how often this occurs and if there is a difference between the sites."

Hypothesis 2: can you add some discussion in your lit review about bedrock controls?

The following sentence was added to the discussion, characterising the effects of bedrock topography on threshold behaviour (page 3, line 1-3):
"Bedrock topography can cause non-linear threshold behaviour in cases where the bedrock is highly impermeable, or where hollows in the bedrock topography need to be filled before spilling over (Freer 2002, Graham 2010, Tromp van Meerveld 2006b)."

Hypothesis 3: again, catchment scale connectivity just seems like a stretch. I wonder if a surrogate for this is just catchment wetness or storage state.

The clear thresholds between the water levels in the piezometers and event runoff coefficients show that the piezometers can be indicative of the catchment state at which larger scale connectivity is quickly established by incoming event rainfall. You are right, it isn't the piezometer observations on their own that tell us about the establishment of larger scale connectivity, it is the high event runoff coefficients that indicate that a large part of the catchment is contributing to stream flow and is thus connected. We will rephrase the

"Hypothesis 3: monitoring at the footslope can provide information on hillslope-stream connectivity at this location and can indicate connectivity at the headwater catchment scale."

Figure 1: I know there is a lot going on in this figure already, but some indication of land cover could be good

Information about land use was added to the Figure but only for each subcatchment as it would have been too cramped otherwise (page 4, Figure 1).

Figure 2: and all figures: since the bedrock comparison is important for the study, it could be good to indicate that more clearly in these figures, so maybe a different colour in the tabs where the site labels are for the two bedrock types, or just labelling them with the name of the bedrock types… just a suggestion.

The site names start with a prefix (either "M_" or "S_") indicating whether it is a marls site ("M_") or schist ("S_"). This is stated on page 5, line 6-7.

Page 6 line 3: "were taking" = took

Page 10 line 9: "allow us" = allowed us

Page 12 line 23: "also" could be removed

Page 15 line 5: "values not necessarily" , should this be "values did not necessarily"

The above points were corrected. Thank you for pointing them out.

Figure 9: Very cool plot.

It is very much appreciated, thank you!

Figure 9: For this and figures 10 and 11: I wonder if there is a way to indicate in these, without them getting too cluttered, how far the wells are from the stream. While they are all considered "near stream", some are much further away than others, and seeing that information might aid in interpretation

We tried to add distance-to-stream to these three plots but unfortunately could not come up

with a solution that supports the visualisation of the data. As Figure 3 shows the distance and gives an overview on the topography, it might be more helpful to use this as an auxiliary information when interpreting the results rather than adding more information into the plots.

Page 18 line 18: Are the thresholds in water level all at the same stream stage across wells? That could be interesting to discuss.

This is indeed an interesting question. When obtaining the thresholds from Figure 10 we realised that this is only possible for the groundwater levels but not the stream level. Trying to do so resulted in different stream level thresholds for different piezometers at a single site. Since the groundwater thresholds were consistent to a certain degree, we did not continue investigating stream water level thresholds.

Page 21 line 9: Can the authors comment on whether or not the particle size distribution in the shallow soils is consistent with the ability to have a thick capillary fringe?

The dominant soil texture is clay or silty clay in the marls (Sprenger et al. 2016). On page 4, line 9 we provided information on the clay content: "Stagnosols with high clay content (20-60%),"

Page 23 line 15: I'm not sure what "infiltration distance" means in this context

The sentence in question was rephrased to (page 24, line 10-12):
"… as the depth of groundwater level (minus a potential capillary fringe) is the distance a water parcel needs to travel and thus, directly influences the delay in groundwater response."

Page 23 line 26: "on the one hand" and "on the other" could be removed

We decided to keep the part in question to really emphasise the separation of water level changes and timing.

Page 24 line 13: "variability of possible flow paths compared to conditions of high groundwater levels" - I'm not sure what this is saying

We decided to remove the sentence in question as it is rephrased in a clearer manner on page 25, line 11

Page 24 line 18: is = are
we corrected this, thank you

Page 25 line 14: I don't see how, with this experimental setup, conclusions can be made about how many hillslopes are connected or how far that connection extends up slopes

As mentioned in the manuscript (page 26, line 9-13), we observe increasing runoff coefficients above a threshold but we cannot differentiate whether few hillslopes connect further upslope, or more hillslopes connect, or a combination of both. However, based on the high event runoff coefficients (in part > 50%) we can say that above the threshold there is the potential that more of the headwater catchment gets connected to the stream.

Page 26 lines 6-7: According to what analyses can you say how many events you need to characterize a response?

As we stated in the conclusions (page 28, line 21 – page 29, line 1): "…the question is not so much about how many events are necessary (in absolute numbers) as more about the necessary time period to cover the temporal variability generated by different hydrological processes. It is therefore necessary to accumulate a large number of events across all seasons".

To clarify this point we have added additional figures to the appendix (see above).

Page 26 line 12: It'd be good to see an example of this capillary fringe transition… could this just be explained by higher winter-time water levels? It'd be interesting to see just how fast this transition occurs. (I may be misunderstanding this though)

We are not entirely sure if we understand you correctly here. You are referring to: "Groundwater transitions fast from very low levels to levels near the surface, with only few events in-between. This fast transition hints towards extended capillary fringes where only low volumes of water are necessary to raise the groundwater table (Cloke et al., 2006)". In Figure 8 (page 17) one can observe that events at M_J and M_K cluster at high groundwater levels with only few events for lower groundwater levels. This includes events from around the year. A reference to this Figure was added to the sentence in question.

Page 27 line 9-13: I don't think much can be said about hillslope travel distances/connectivity in the framework described in the Klaus and Jackson paper. Sure, there is a level at which there is a suggestion that a lot of water is moving into the stream, but is that just due to a much higher potential energy gradient? Does that really give any specific indication of from how far up the

hillslope that water is coming? It just doesn't seem that relevant to the rest of the paper.

Two previous reviewers asked to discuss this paper in the context of our manuscript. Under the viewpoint of potential connectivity from upslope regions it could be relevant. However, since we do not make any statement to that regard it remains a bit out of context. We would like to leave it to the Editor to decide whether or not we should include this statement and reference.

Conclusions

First paragraph: First, I think a better first paragraph would be a very succinct summary of what the authors did in the study, just as a lead in and transition. Second, as detailed above, I'm not sure the conclusions made quite strongly in this first paragraph are backed up by the study. I'd consider adding some sort of detailed discussion or analysis of how the findings of the study would change with a more limited time series, or just taking this stuff out.

A very concise intro about what was done was added to the first paragraph. Regarding the second point made in this comment, please see our previous response (second comment) regarding the newly added figures in the appendix.

Hypothesis section: I like the very straightforward presentation of how the hypotheses were addressed. However, I think it would be useful to restate each one. Otherwise you have to flip back and forth in the paper to see which is which. Also, I think it would be good to go through these and be sure that there is only discussion of conclusions directly about each hypotheses in each paragraph. Right now H1 has discussion of geology that doesn't seem related to H1 as stated in the beginning of the paper.

We welcome this point of critique regarding the spill-over of conclusions about geology where it is not part of the particular hypotheses. To address the hypotheses more neatly we rearranged the part in question and restated the hypotheses (page 29, line 7-16).

Page 28 Lines 15-19: I had to read this sentence several times to figure it out, I think it should be split into two or more.

The questioned sentence was rephrased to: "The fact that at low groundwater levels runoff coefficients in the marls tend to be higher than in the schist, in some cases even by an order of magnitude, suggests that also at low groundwater levels different processes are active in the two geological regions. This indicates that in contrast to schist, marls develops surface runoff or lateral preferential flow above the shallow groundwater that must provide sufficient connectivity to enable runoff generation while saturated subsurface connectivity is low."

Page 28 line 32: as mentioned above, the footslope can be gatekeeper, but there are other explanations for why water levels there might be well synced up with streamflow.

please see our response to this question above

Page 29 line 5: "can be identified a-priori" if this is going to be included in the conclusions, I think there should be discussion of at least a proposed method for how this could be done in the discussion. Were the topographic similarities between the wells that worked well for the analysis?

This is an outlook, a suggestion, not something we have tested.

Page 29 line 7: "novel way of visualizing" Definitely! The figures in this manuscript are super well done.

Again, thank you!

[revised manuscript text omitted]

[44] removed: M_D
[45] removed: Piezo1
[46] removed: M_D
[47] removed: Piezo1
[48] removed: M_D
[49] removed: Piezo1
[50] removed: M_D
[51] removed: Piezo1
[52] removed: M_D
[53] removed: Piezo1
[54] removed: M_D
[55] removed: M_D
[56] removed: Piezo2
[57] removed: M_D
[58] removed: Piezo2
[59] removed: M_D
[60] removed: Piezo2
[61] removed: M_D
[62] removed: Piezo2
[63] removed: M_D
[64] removed: Piezo2
[65] removed: M_D
[66] removed: Piezo2
[67] removed: M_D

| Cluster | Piezometer | Horizon | Depth |
|---|---|---|---|
| [-] | [-] | [-] | [cm] |
| [..[68] ] | [..[69] ] | B1 | -35 |
| [..[70] ] | [..[71] ] | B2 | -55 |
| [..[72] ] | [..[73] ] | B3 | -162 |
| [..[74] ] | [..[75] ] | Cv | |
| [..[76] ] | Piezo4 | Ah | -4 |
| [..[77] ] | [..[78] ] | B1 | -121 |
| [..[79] ] | [..[80] ] | B2 | -186 |
| [..[81] ] | [..[82] ] | B3 | -246 |
| [..[83] ] | [..[84] ] | B4 | -313 |
| [..[85] ] | [..[86] ] | B5 | -335 |
| [..[87] ] | [..[88] ] | C | |
| M_J | Piezo1 | Ah | -20 |
| [..[89] ] | [..[90] ] | B1 | -70 |
| [..[91] ] | [..[92] ] | B2 | -95 |
| [..[93] ] | [..[94] ] | B2.2 | -112 |

[68] removed: M_D
[69] removed: Piezo3
[70] removed: M_D
[71] removed: Piezo3
[72] removed: M_D
[73] removed: Piezo3
[74] removed: M_D
[75] removed: Piezo3
[76] removed: M_D
[77] removed: M_D
[78] removed: Piezo4
[79] removed: M_D
[80] removed: Piezo4
[81] removed: M_D
[82] removed: Piezo4
[83] removed: M_D
[84] removed: Piezo4
[85] removed: M_D
[86] removed: Piezo4
[87] removed: M_D
[88] removed: Piezo4
[89] removed: M_J
[90] removed: Piezo1
[91] removed: M_J
[92] removed: Piezo1
[93] removed: M_J
[94] removed: Piezo1

| Cluster [-] | Piezometer [-] | Horizon [-] | Depth [cm] |
|---|---|---|---|
| [..[95] ] | [..[96] ] | B3 | -142 |
| [..[97] ] | [..[98] ] | B3.2 | -150 |
| [..[99] ] | [..[100] ] | B4 | -170 |
| [..[101] ] | Piezo2 | Ah | -9 |
| [..[102] ] | [..[103] ] | B1 | -45 |
| [..[104] ] | [..[105] ] | B2 | -83 |
| [..[106] ] | [..[107] ] | B3 | |
| [..[108] ] | Piezo3 | Ah | -10 |
| [..[109] ] | [..[110] ] | B1 | -41 |
| [..[111] ] | [..[112] ] | B2 | -60 |
| [..[113] ] | [..[114] ] | B3 | |
| [..[115] ] | Piezo4 | B4 | -50 |
| M_K | Piezo1 | Ah | -12 |
| [..[116] ] | [..[117] ] | B1 | -30 |
| [..[118] ] | [..[119] ] | B2 | -50 |
* * *
[95] removed: M_J
[96] removed: Piezo1
[97] removed: M_J
[98] removed: Piezo1
[99] removed: M_J
[100] removed: Piezo1
[101] removed: M_J
[102] removed: M_J
[103] removed: Piezo2
[104] removed: M_J
[105] removed: Piezo2
[106] removed: M_J
[107] removed: Piezo2
[108] removed: M_J
[109] removed: M_J
[110] removed: Piezo3
[111] removed: M_J
[112] removed: Piezo3
[113] removed: M_J
[114] removed: Piezo3
[115] removed: M_J
[116] removed: M_K
[117] removed: Piezo1
[118] removed: M_K
[119] removed: Piezo1

| Cluster | Piezometer | Horizon | Depth |
|---|---|---|---|
| [-] | [-] | [-] | [cm] |
| [..[120] ] | [..[121] ] | B3 | -97 |
| [..[122] ] | Piezo2 | Ah | -15 |
| [..[123] ] | [..[124] ] | B1 | -35 |
| [..[125] ] | [..[126] ] | B2 | -93 |
| [..[127] ] | Piezo3 | Ah | -13 |
| [..[128] ] | [..[129] ] | B1 | -35 |
| [..[130] ] | [..[131] ] | B2 | -91 |
| [..[132] ] | Piezo4 | Ah | -8 |
| [..[133] ] | [..[134] ] | B1 | -45 |
| [..[135] ] | [..[136] ] | B2 | -85 |
| S_J | Piezo1 | Ah | -7 |
| [..[137] ] | [..[138] ] | B | -88 |
| [..[139] ] | [..[140] ] | Cv1 | -110 |
| [..[141] ] | [..[142] ] | Cv2 | >114 |
| [..[143] ] | Piezo2 | Ah | -3 |

[120] removed: M_K
[121] removed: Piezo1
[122] removed: M_K
[123] removed: M_K
[124] removed: Piezo2
[125] removed: M_K
[126] removed: Piezo2
[127] removed: M_K
[128] removed: M_K
[129] removed: Piezo3
[130] removed: M_K
[131] removed: Piezo3
[132] removed: M_K
[133] removed: M_K
[134] removed: Piezo4
[135] removed: M_K
[136] removed: Piezo4
[137] removed: S_J
[138] removed: Piezo1
[139] removed: S_J
[140] removed: Piezo1
[141] removed: S_J
[142] removed: Piezo1
[143] removed: S_J

| Cluster | Piezometer | Horizon | Depth |
|---|---|---|---|
| [-] | [-] | [-] | [cm] |
| [..[144] ] | [..[145] ] | B | -34 |
| [..[146] ] | [..[147] ] | B2 | -59 |
| [..[148] ] | [..[149] ] | Cv | >59 |
| [..[150] ] | Piezo3 | Ah | -9 |
| [..[151] ] | [..[152] ] | B | -35 |
| [..[153] ] | [..[154] ] | B2 | -58 |
| [..[155] ] | [..[156] ] | Cv | >85 |
| [..[157] ] | Piezo4 | Ah | -20 |
| [..[158] ] | [..[159] ] | B | -72 |
| [..[160] ] | [..[161] ] | Cv | -117 |
| [..[162] ] | [..[163] ] | Cv2 | >117 |
| S_V | Piezo1 | Ah | -12 |
| [..[164] ] | [..[165] ] | B1 | -50 |
| [..[166] ] | [..[167] ] | B2 | -80 |
| [..[168] ] | [..[169] ] | B3 | -132 |

[144] removed: S_J
[145] removed: Piezo2
[146] removed: S_J
[147] removed: Piezo2
[148] removed: S_J
[149] removed: Piezo2
[150] removed: S_J
[151] removed: S_J
[152] removed: Piezo3
[153] removed: S_J
[154] removed: Piezo3
[155] removed: S_J
[156] removed: Piezo3
[157] removed: S_J
[158] removed: S_J
[159] removed: Piezo4
[160] removed: S_J
[161] removed: Piezo4
[162] removed: S_J
[163] removed: Piezo4
[164] removed: S_V
[165] removed: Piezo1
[166] removed: S_V
[167] removed: Piezo1
[168] removed: S_V
[169] removed: Piezo1

| Cluster | Piezometer | Horizon | Depth |
|---|---|---|---|
| [-] | [-] | [-] | [cm] |
| [..170 ] | [..171 ] | Cv1 | -160 |
| [..172 ] | [..173 ] | Cv2 | |
| [..174 ] | Piezo2 | Ah | -11 |
| [..175 ] | [..176 ] | B1 | -58 |
| [..177 ] | [..178 ] | Bv | -86 |
| [..179 ] | [..180 ] | B3 | |
| [..181 ] | Piezo3 | Ah | -13 |
| [..182 ] | [..183 ] | B1 | -62 |
| [..184 ] | [..185 ] | B2 | |
| [..186 ] | Piezo4 | Ah | -14 |
| [..187 ] | [..188 ] | Rock | -24 |
| [..189 ] | [..190 ] | B | -81 |
| [..191 ] | [..192 ] | Cv | |

[170]removed: S_V
[171]removed: Piezo1
[172]removed: S_V
[173]removed: Piezo1
[174]removed: S_V
[175]removed: S_V
[176]removed: Piezo2
[177]removed: S_V
[178]removed: Piezo2
[179]removed: S_V
[180]removed: Piezo2
[181]removed: S_V
[182]removed: S_V
[183]removed: Piezo3
[184]removed: S_V
[185]removed: Piezo3
[186]removed: S_V
[187]removed: S_V
[188]removed: Piezo4
[189]removed: S_V
[190]removed: Piezo4
[191]removed: S_V
[192]removed: Piezo4

[revised manuscript text omitted]